# A Diffusive Classification Loss for Learning Energy-based Generative Models

**RuiKang OuYang** [* 1]  **Louis Grenioux** [* 2 3]  **José Miguel Hernández-Lobato** [1]

## Abstract

Score-based generative models have recently achieved remarkable success. While they are usually parameterized by the score, an alternative way is to use a series of time-dependent energy-based models (EBMs), where the score is obtained from the negative input-gradient of the energy. Crucially, EBMs can be leveraged not only for generation, but also for tasks such as compositional sampling or building Boltzmann Generators via Monte Carlo methods. However, training EBMs remains challenging. Direct maximum likelihood is computationally prohibitive due to the need for nested sampling, while score matching, though efficient, suffers from mode blindness. To address these issues, we introduce the *Diffusive Classification* (`DiffCLF`) objective, a simple method that avoids blindness while remaining computationally efficient. `DiffCLF` reframes EBM learning as a supervised classification problem across noise levels, and can be seamlessly combined with standard score-based objectives. We validate the effectiveness of `DiffCLF` by comparing the estimated energies against ground truth in analytical Gaussian mixture cases, and by applying the trained models to tasks such as model composition and Boltzmann Generator sampling. Our results show that `DiffCLF` enables EBMs with higher fidelity and broader applicability than existing approaches. Our code is available at h2o64/diffclf.

*Equal contribution, order assigned randomly. [1]Department of Engineering, University of Cambridge, Cambridge, United Kingdom [2]CMAP, CNRS, École polytechnique, Institut Polytechnique de Paris, Palaiseau, France [3]Center for Computational Mathematics, Flatiron Institute, New York, NY, USA. Correspondence to: RuiKang OuYang <ro352@cam.ac.uk>, Louis Grenioux <lgrenioux@flatironinsitute.org>.

*Proceedings of the 43$^{rd}$ International Conference on Machine Learning*, Seoul, South Korea. PMLR 306, 2026. Copyright 2026 by the author(s).

## 1. Introduction

Probabilistic modeling is a cornerstone of modern machine learning, providing a principled framework to capture complex data distributions and to generate realistic samples. A classical approach is density estimation, often carried out by explicitly modeling it using Energy-Based Models (EBMs), where the density is parametrized as the exponential of the negation of a learnable function, referred to as the energy (LeCun et al., 2006; Kim & Bengio, 2016; Nijkamp et al., 2019; Du & Mordatch, 2019; Grathwohl et al., 2020; Che et al., 2020; Song & Kingma, 2021). While conceptually appealing, EBMs are notoriously hard to train due to the intractable normalizing constant that prevents maximum likelihood estimation, forcing reliance on costly sampling procedures. Despite advances in amortized and efficient sampling (Du & Mordatch, 2019; Du et al., 2021; Grathwohl et al., 2021; Carbone et al., 2023; Senetaire et al., 2026), training EBMs remains computationally challenging.

A popular alternative avoids the difficulty of modeling energies directly by targeting their negative gradient, the score function. Score matching methods (Hyvärinen, 2005) leverage the fact that the intractable normalizing constant disappears upon differentiation, making the score easier to estimate. This approach became especially prominent with the advent of score-based generative models such as Diffusion Models (Sohl-Dickstein et al., 2015; Ho et al., 2020; Song et al., 2021b) and Stochastic Interpolants (Albergo & Vanden-Eijnden, 2023; Albergo et al., 2025), where a forward noising process gradually transforms data into pure noise (or another tractable distribution), and sampling is achieved by reversing this process through a denoising dynamic. Crucially, these denoising dynamics depend on the score of the perturbed data distribution, which can be efficiently learned using Denoising Score Matching (Sohl-Dickstein et al., 2015; Song et al., 2021b).

Beyond generating samples, estimating the underlying energy function, rather than only the score, enables a range of downstream applications. Examples include building Boltzmann Generators from generative models via Monte Carlo methods (Phillips et al., 2024; Zhang et al., 2025) and composing multiple models (Du et al., 2023; Skreta et al., 2025b;a; Thornton et al., 2025; He et al., 2026a), both of which fundamentally require access to energies. While

prior works (Skreta et al., 2025b;a; Zhang et al., 2025; He et al., 2026a) focus on estimating marginal density ratios using only scores, they typically rely on assumptions such as perfectly learned scores or approximations of transition kernels in small time steps. In contrast, direct access to the energy often leads to improved performance.

While several works (Salimans & Ho, 2021; Phillips et al., 2024; Thornton et al., 2025) have explored training energy-based generative models through score-based objectives, yielding approximations of the energies, these methods face significant limitations. Chief among them is mode blindness, where the relative proportions of disjoint high-density regions are misrepresented (Wenliang & Kanagawa, 2021; Zhang et al., 2022; Shi et al., 2024). Recent efforts aim to recover energies of diffusion models directly, but they often demand heavy computation or fragile hyperparameter tuning (Gao et al., 2021; Zhang et al., 2023; Schröder et al., 2023; Zhu et al., 2024).

**Our contributions.** We address the problem of training energy-based generative models in a way that enables direct downstream use of energies. Our main contributions are:

- We introduce the *Diffusive Classification* (`DiffCLF`) objective, which reframes log-density estimation as a **supervised classification problem**. `DiffCLF` is lightweight, flexible, and can be seamlessly combined with classical score-matching objectives.

- We prove that `DiffCLF` consistently recovers the ground-truth distribution at optimality and, unlike score-based methods, is *not mode-blind*.

- We establish connections between `DiffCLF` and prior approaches that exploit temporal correlations in stochastic processes to learn energy-based models.

- We demonstrate the effectiveness of `DiffCLF` across different generative processes and for a range of downstream tasks, including building Boltzmann Generators and compositional generation.

## 2. Preliminary

### 2.1. General framework

This work considers stochastic processes on $\mathbb{R}^d$ for $t \in [0, T]$ of the form

$$Y_t = X_t + \gamma(t)Z , \qquad (1)$$

where $\gamma \in C^2([0, T])$, $T$ denotes the terminal time which may be infinite, $(X_t)_t$ is a stochastic process from which samples can be drawn at any time $t$, and $Z$ is a standard Gaussian noise independent of $(X_t)_t$. Let $q_t$ denote the marginal density of $X_t$ and $p_t$ the marginal density of $Y_t$. This framework serves as a generic setting, with concrete

instances and applications provided later. The objective is to estimate the densities $(p_t)_t$ up to a normalizing constant, given access only to samples from $X_t$. To achieve this, a parametric family of energy-based models is introduced

$$
\begin{aligned}
p_t^\theta(y_t) &= \exp(-\mathrm{U}_t^\theta(y_t))/\mathcal{Z}_t^\theta , \\
\mathcal{Z}_t^\theta &= \exp(\mathrm{F}_t^\theta) = \int \exp(-\mathrm{U}_t^\theta(y_t))\mathrm{d}y_t ,
\end{aligned}
\qquad (2)
$$

where $\mathrm{U}^\theta : [0, T] \times \mathbb{R}^d \to \mathbb{R}$ is the energy function, parameterized by $\theta \in \Theta$ (for instance, neural networks), and $\mathcal{Z}_t^\theta$ is the associated normalizing constant, which is intractable in general. A natural approach to estimating this model is *Maximum Likelihood* (ML) which writes as

$$\mathcal{L}_{\mathrm{ML}}(\theta; t) = -\mathbb{E}[\log p_t^\theta(Y_t)] = \mathbb{E}[\mathrm{U}_t^\theta(Y_t)] + \log \mathcal{Z}_t^\theta , \quad (3)$$

where $Y_t \sim p_t$. Given $t \in [0, T]$, taking gradients of $\mathcal{L}_{\mathrm{ML}}(\cdot; t)$ with respect to $\theta$ yields

$$\nabla_\theta \mathcal{L}^{\mathrm{ML}}(\theta; t) = \mathbb{E}_{p_t}[\nabla_\theta \mathrm{U}_t^\theta(Y_t)] - \mathbb{E}_{p_t^\theta}[\nabla_\theta \mathrm{U}_t^\theta(Y_t)] . \quad (4)$$

The difficulty of this approach lies in the second term, which requires sampling from $p_t^\theta$. Since this distribution can be as complex as the original data distribution $q_t$, sampling typically demands expensive Monte Carlo methods, making ML estimation impractical.

An alternative is to exploit the structure of Equation (1) and apply *Denoising Score Matching* (DSM) (Sohl-Dickstein et al., 2015; Song et al., 2021b) which aims at learning the gradient of the log-density (the score) to recover the energy up to an additive constant as a by-product. Let $p_t(\cdot|x_t)$ be the density of $Y_t$ conditional on $X_t = x_t$. By construction, for all $t \in [0, T]$ and $x_t, y_t \in \mathbb{R}^d$

$$p_t(y_t|x_t) = \mathcal{N}(y_t; x_t, \gamma^2(t)\mathrm{I}_d) . \qquad (5)$$

Using that $p_t(y_t) = \int p_t(y_t|x_t)q_t(x_t)\mathrm{d}x_t$, the score [1] whenever no ambiguity arises. can then be expressed as

$$\nabla \log p_t(y_t) = \mathbb{E}\left[\nabla \log p_t(Y_t|X_t) \mid Y_t = y_t\right] . \quad (6)$$

Equation (6) is often referred to as the Tweedie's formula (Efron, 2011). This characterization shows that the score is a conditional expectation over the posterior distribution of $X_t$ given $Y_t$. Building on this, DSM defines the following mean square objective, whose minimum is attained by Equation (6),

$$\mathcal{L}_{\mathrm{DSM}}(\theta; t) = \mathbb{E}\left[\left\|\nabla \log p_t^\theta(Y_t) - \nabla \log p_t(Y_t|X_t)\right\|^2\right], \quad (7)$$

where $X_t \sim q_t$ and $Y_t \sim p_t(\cdot|X_t)$. Note that various heuristics exist for designing the time distribution see for

---

[1] For clarity, we write $\nabla \log p_t(y)$ (respectively $\nabla \log p_t(y|x)$) as shorthand for $\nabla_y \log p_t(y)$ (respectively $\nabla_y \log p_t(y|x)$)

instance (Song et al., 2021b; Karras et al., 2022; Kingma & Gao, 2023). The DSM objective doesn't require to compute the normalizing constant or to sample from the model.

The framework introduced in Equation (1) underlies many widely used generative models. The core idea is to generate new samples via a Markov process, typically formulated as a *Stochastic Differential Equation* (SDE), whose marginals coincide with those of $Y_t$. Crucially, constructing such processes relies on access to the score function $\nabla \log p_t$. Below, two concrete and widely used instances of this framework are presented.

**Example 1 : Diffusion Models** In *Diffusion Models* (DMs) (Sohl-Dickstein et al., 2015; Ho et al., 2020; Song et al., 2021b), one usually defines $(Y_t)_t$ through a noising (forward) SDE[2] $dY_t = f(t)Y_t dt + g(t)dW_t$ starting from $Y_0 \sim p_0$, where $(W_t)_t$ is a standard Brownian motion. Using the notations of (Karras et al., 2022, Appendix B), the solution of the noising SDE can be written as $Y_t = S(t)Y_0 + S(t)\sigma(t)Z$ with $Z \sim \mathcal{N}(0, I)$, where $S(t) = \exp(\int_0^t f(u)du)$ and $\sigma(t)^2 = \int_0^t (g(u)/S(u))^2 du$. The noising SDE fits Equation (1) with $X_t = S(t)X_0$ where $X_0 \sim p_0$ and $\gamma(t) = S(t)\sigma(t)$. Under mild conditions on $\pi$ (Anderson, 1982), one can derive the corresponding denoising (backward) SDE

$$dY_t = \left[ f(t)Y_t - g^2(t)\nabla \log p_t(Y_t) \right] dt + g(t)d\tilde{W}_t \ , \quad (8)$$

from $Y_T \sim p_T$, where $(\tilde{W}_t)_t$ is a standard Brownian motion. Estimating the score function $\nabla \log p_t$ via DSM (7) allows approximating this SDE which can be solved backward in time to produce samples from $\pi$.

**Example 2 : Stochastic Interpolants** *Stochastic Interpolants* (SIs) (Albergo & Vanden-Eijnden, 2023; Albergo et al., 2025) generalize diffusion models by constructing generative processes that interpolate between any two distributions, not necessarily with a Gaussian endpoint. In this setting, $X_t = I_t(X_0, X_1)$ with $X_0 \sim q_0$, $X_1 \sim q_1$, where datasets are available for both $q_0$ and $q_1$. The interpolation function $I$ is chosen so that $X_0$ and $X_1$ are recovered at times $t = 0$ and $t = 1$, while $\gamma(0) = \gamma(1) = 0$ ensures that $Y_0 \sim q_0$ and $Y_1 \sim q_1$. Under mild assumptions, the dynamics of $(Y_t)_t$ are given by the SDE

$$dY_t = \left[ v_t(Y_t) - \left( \dot{\gamma}(t)\gamma(t) + \frac{g(t)^2}{2} \right) \nabla \log p_t(Y_t) \right] dt$$
$$+ g(t)dW_t \ , \quad \text{from } Y_0 \sim p_0, \quad (9)$$

where $(W_t)_t$ is a standard Brownian motion, $g$ is any positive function and $v_t$ is defined via a conditional expectation

as $v_t(y_t) = \mathbb{E}[\partial_t I_t(X_0, X_1) \mid Y_t = y_t]$. As in DSM, $v_t$ and the score can be learned via simple regression objectives , and once approximated, the SDE can be integrated to generate new samples from $q_1$ [3]. Notably, DMs appear as a special case of this framework. Experiments conducted on both the DMs and SIs settings will be delivered in Section 5.

## 2.2. Why modeling energies ?

As illustrated by DMs and SIs, the most natural quantity to model is often the score $\nabla \log p_t$. However, directly learning the energies themselves brings several important advantages, with only a few highlighted below.

**Boltzmann Generators.** Boltzmann Generators (BGs) (Noé et al., 2019) aim to transform a trained generative model into an effective sampler for a target Boltzmann distribution $\pi$, whose energy is known up to a constant. By exploiting density surrogates learned from a limited set of biased samples, BGs enable unbiased (or asymptotically exact) estimation of expectations under $\pi$ through reweighting and/or Markov corrections. While early BG approaches primarily relied on classical techniques such as importance sampling and MCMC, recent work shows that, in the context of DMs or SIs, one can instead leverage annealed and sequential methods such as AIS (Zhang et al., 2024; 2025), SMC (Phillips et al., 2024) and RE (Zhang et al., 2026). The central idea is to exploit the sequence of intermediate model densities $(p_t^\theta)_t$ to decompose the sampling task into a series of easier transitions bridging a simple base distribution and the target $\pi$.

**Compositionality.** Accurate energy estimation enables training-free compositional operations between generative models. Recent work shows that multiple diffusion models trained on different targets can be combined to form new models of their mixtures or products. Such constructions can be sampled using advanced methods, including AIS (Skreta et al., 2025a;b), annealed Langevin dynamics (Du et al., 2023; Lee et al., 2023; Zhu et al., 2024), SMC (Thornton et al., 2025; He et al., 2026a), and RE (He et al., 2026b).

**Free energy difference estimation.** Another application is estimating free-energy differences between states, a central yet challenging problem in statistical physics (Lelièvre et al., 2010), even with equilibrium samples. Classical methods such as *Thermodynamic Integration* (TI, Kirkwood, 1935) and *Multistate Bennett Acceptance Ratio* (MBAR, Shirts & Chodera, 2008) rely on annealed sequences of intermediate potentials bridging the two states. Recent works (Máté et al., 2025; Du et al., 2025) construct these paths via SIs by training energy-based models to learn the intermedi-

---

[2]Popular choices for the function $f$ and $g$ include the *Variance Preserving* (VP) and *Variance Exploding* (VE), see Appendix A.2 or (Song et al., 2021b) for details.

[3]The transport from $q_1$ to $q_0$ can be obtained by changing the sign of the diffusion coefficient in front of the score in SDE (9), i.e. $g(t)^2/2 \rightarrow -g(t)^2/2$, and integrating from $t = 1$ to $t = 0$ with $Y_1 \sim p_1$.

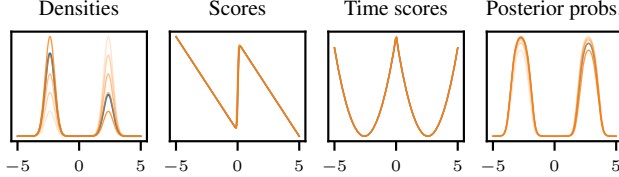

Figure 1. **Densities, scores, time-scores, and classification posterior probabilities of Gaussian mixtures with varying weights.** From left to right : (1) Reference mixture (blue, weights $2/3$–$1/3$) and perturbed mixtures (orange, left mode weight ranging in $[0.2, 0.8]$, with transparency proportional to the value) at $t_1 = 0.1$ under variance-preserving noising (Song et al., 2021b). (2) Scores remain nearly identical across mixtures, (3) Time-scores show the same limitation, while (4) 3-class classification posterior probabilities $p_{t_1}(c = 1 \mid \cdot)$ (10) (with $t_2 = 0.5$, $t_3 = 0.7$) vary with the mixture weights.

ate potentials, providing a data-driven alternative that can outperform hand-crafted designs.

*Remark* 2.1. Strict normalization of the learned energies is not required for the downstream applications considered. SMC and AIS (used in Boltzmann Generators and inference-time control) rely on self-normalized importance weights; Langevin dynamics needs only scores; Metropolis-Hastings accepts unnormalized energies; and TI and MBAR estimate free energy differences without requiring normalization.

### 2.3. On the limitation of score-matching methods

While score-based methods learn energies via their gradients, they suffer from fundamental limitations. Perhaps the most critical is the the "blindness" of score matching (Wenliang & Kanagawa, 2021; Zhang et al., 2022; Shi et al., 2024): divergences that rely solely on scores, such as the Fisher divergence, the Stein discrepancy or any SM-based objective, are not valid when distributions have disjoint supports, since a zero distance does not guarantee equality (Zhang et al., 2022, Theorem 2). Intuitively, the score of a multi-modal distribution captures only local information within each mode, ignoring other high-probability regions. Consequently, distributions with identical modes but different mixture weights produce nearly identical scores. Figure 1 illustrates this: Gaussian mixtures with differing weights ($1^{st}$ panel) yield almost identical scores ($2^{nd}$ panel). This limitation prevents SM objectives from reliably recovering the correct weighting across disconnected regions.

## 3. Learning energies via classification

In this work, the energy is modeled by jointly learning the time-dependent energy function $U_t^\theta$ and the log-normalizing constant $F_t^\theta = -\log \mathcal{Z}_t^\theta$ [4] , not just the score .

---

[4]Note that $F_t^\theta$ is not the true negative log-normalizing constant, but only a learnable parameter.

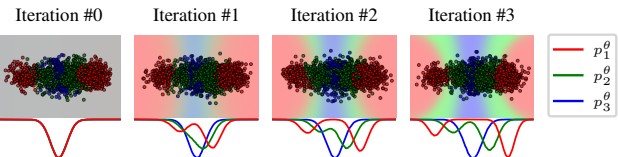

Figure 2. **Classification posterior probabilities and associated EBM during training.** Red, green, and blue dots are samples from $p_{t_1}, p_{t_2}, p_{t_3}$, with 1D marginals of the learned densities shown as curves of the same colors. The background encodes posterior probabilities from the classifier (11) (RGB channels). The target distribution is a mixture of $\mathcal{N}((-1, 0), 0.02I_2)$ with weight 0.3 and $\mathcal{N}((+1, 0), 0.02I_2)$ with weight 0.7, and the intermediate distributions are obtained via a variance-preserving noising scheme. As optimization progresses, class separation improves in the background, enabling accurate recovery of the underlying densities.

Our approach is based on minimizing the DSM objective (7) jointly with the following *Diffusive Classification* (`DiffCLF`) objective

$$\mathcal{L}_{\text{clf}}(\theta; N) = \mathbb{E}_{t_{1:N}}[\mathcal{L}_{\text{clf}}(\theta; t_{1:N})],$$

$$\mathcal{L}_{\text{clf}}(\theta; t_{1:N}) = -\frac{1}{N}\sum_{i=1}^N \mathbb{E}_{p_{t_i}}\left[\log \frac{p_{t_i}^\theta(Y_{t_i})}{\sum_{j=1}^N p_{t_j}^\theta(Y_{t_i})}\right],$$

(10)

where $t_i$ are sampled from $U([0, T])$, and $Y_{t_i}$ are sampled from $p_{t_i}$. It is noticeable that $F_t^\theta$ aims not normalizing $p_t^\theta$ but providing extra degree of freedom to aid training, which follows the same spirit as (Gutmann & Hyvärinen, 2010). In practice, $F_t^\theta$ is implemented as a bias on the last layer of the neural network $U_t^\theta$.

The objective (10) reformulates the task of estimating the EBM as a multi-class classification problem. Consider a sample $y$ associated with a label $c$. If $c = i$, this indicates that $y$ was generated from the marginal distribution at time $t_i$. In multinomial logistic regression, the goal is to estimate the posterior distribution over classes given the data $y$. Here, the class-conditional probability $p(\cdot|c = i)$ is modeled by $p_{t_i}^\theta$. If we further assume that all $N$ labels are equally likely, i.e., $p(c = i) = 1/N$, then the posterior probabilities are

$$p^\theta(c = i \mid y) = p_{t_i}^\theta(y)/\sum_{j=1}^N p_{t_j}^\theta(y) \,.$$

(11)

The categorical cross-entropy of this classifier corresponds exactly to Equation (10). Figure 2 illustrates this in the 3-level setting: given samples from three distinct time steps, the objective solves the classification problem to progressively separate them, as visualized by the posterior probabilities in the background. Because the classifier is constructed as a softmax over the EBMs (shown along the bottom edge of the figure), optimizing this objective also directly learns the marginal distribution of each sample group. As shown

in the rightmost panel of Figure 1, the posterior probabilities (11) reflect changes in the mixture weights, unlike the score. This highlights that, in contrast to DSM, the classification objective doesn't suffer from mode blindness. Proposition 3.1 demonstrates that the true marginals attain the minimum of the DiffCLF objective.

**Proposition 3.1.** *The true marginals* $(p_t)_{t \in [0,T]}$ *are a minimizer of* $\mathcal{L}_{\mathrm{clf}}$ *(Equation (10)).*

However, the minimizer is not unique; e.g., the same minimum is attained by setting $p_t^\theta(y) = c(y)p_t(y)$ for all $t$, where $c$ is any positive function satisfying $\int c(y)p_t(y)dy = 1$. To fix it, Proposition 3.2 shows that by optimizing the joint objective $\mathcal{L}_{\mathrm{DSM}} + \mathcal{L}_{\mathrm{clf}}$, the minimizer is unique.

**Proposition 3.2** (Uniqueness - Informal)**.** *The unique minimizer for the joint objective* $\mathcal{L}_{\mathrm{DSM}} + \mathcal{L}_{\mathrm{clf}}$ *(Equations (7) and (10)) is attained by* $p_t^{\theta^\star} = p_t$*, for every* $t \in [0,T]$*.*

Furthermore, under mild regularity assumptions, the Monte-Carlo estimator of $\mathcal{L}_{\mathrm{DSM}} + \mathcal{L}_{\mathrm{clf}}$ (the quantity actually optimized during training) provably approximates the true objective. In particular, the empirical minimizer is both consistent and asymptotically normal:

**Proposition 3.3** (Consistency - Informal)**.** *Let* $\hat{\theta}^M$ *denote the minimizer of the Monte-Carlo approximation of* $\mathcal{L}_{\mathrm{DSM}} + \mathcal{L}_{\mathrm{clf}}$ *based on* $M$ *samples. Then:*

*(i)* $\hat{\theta}^M \xrightarrow{\mathbb{P}} \theta^\star$*, where* $\theta^\star$ *is the minimizer of* $\mathcal{L}_{\mathrm{DSM}} + \mathcal{L}_{\mathrm{clf}}$*;*

*(ii)* $\sqrt{M}(\hat{\theta}_M - \theta^\star)$ *is asymptotically normal with mean zero and known covariance matrix* $\Sigma_N$*.*

Moreover, Corollary D.9 establishes that the norm of the asymptotic covariance $\Sigma_N$ decreases as the number of classification levels $N$ increases, reflecting the variance-reduction benefits of richer time-discretizations. Formal statements and proofs are provided in Appendix D.

In the simplest case of two times $t, t' \in [0,T]$, the objective reduces to binary classification, yielding

$$\mathcal{L}_{\mathrm{clf}}(\theta; t, t') = -\frac{1}{2}\mathbb{E}_{p_t}\left[\log h_\theta(Y_t; t, t')\right]$$
$$-\frac{1}{2}\mathbb{E}_{p_{t'}}\left[\log(1 - h_\theta(Y_{t'}; t, t'))\right] , \quad (12)$$

where $h_\theta(y; t, t') = \mathrm{sigmoid}(-\mathrm{U}_t^\theta(y) + \mathrm{F}_t^\theta + \mathrm{U}_{t'}^\theta(y) - \mathrm{F}_{t'}^\theta)$ and $\mathrm{sigmoid}(z) = 1/(1 + \exp(-z))$. In Appendix E, we show that these objectives could be further generalized using Bergman divergences to learn the density ratios. We provide pseudo codes for training with DiffCLF with the multi-level and binary-level objectives in Algorithms 1 and 2.

**Computational cost.** While the DSM objective (7) requires exactly two neural network evaluations per time sampled[5], the multi-class classification objective (10) requires $N$. Consequently, minimizing both $\mathcal{L}_{\mathrm{DSM}}$ and $\mathcal{L}_{\mathrm{clf}}$

---

[5]One for the forward pass and one for the backward pass to compute the score with auto-differentiation.

simultaneously requires only $N + 1$ evaluations per time ($N - 1$ more evaluations than DSM). In the binary case, this amounts to just one additional evaluation compared to DSM, making the computational budgets highly comparable.

**Beyond Euclidean spaces.** It is noticeable that DiffCLF remains valid on different processes and manifolds, since it only requires $(p_t^\theta)_t$ to compute the loss. We discuss the case for applying DiffCLF on continuous-time Markov chains (CMTCs) for discrete diffusion in Appendix F.

# 4. Connection to other works

This section discusses connections between the proposed approach and existing methods. It first focuses on approaches that constrain higher-order derivatives of the log-density (Section 4.1), and then considers works that directly operate on the log-density itself (Section 4.2). All the losses introduced, including DiffCLF, operate on the log-densities, without imposing a hard constraint to ensure the learned energies are normalized.

## 4.1. Constraining other derivatives of the log-density

As highlighted in Section 2.3, regressing solely against the score is insufficient to recover the full energy landscape. Our approach tackles this challenge by using a loss that depends directly on the energy values, while alternative methods attempt to mitigate the same limitations through additional constraints on the model's higher-order derivatives.

### 4.1.1. CONNECTION TO REGRESSING TIME DERIVATIVES OF LOG-DENSITIES.

**Time Score Matching.** An intuitive direction is to exploit the temporal structure of the process. This leads to optimizing EBMs so that their time-derivative aligns with that of the true marginals. We refer to this objective as *Time Score Matching* ($t$SM):

$$\mathcal{L}_{t\mathrm{SM}}(\theta; t) = \mathbb{E}\left[\left(\partial_t \log p_t^\theta(Y_t) - \partial_t \log p_t(Y_t)\right)^2\right] , \quad (13)$$

where $Y_t \sim p_t$. Figure 1 shows that, similar to the score, the time-score $\partial_t \log p_t$ also exhibits blindness of mode-weights. A theoretical justification is provided in Appendix C. Moreover, Proposition 4.1 establishes that the binary classification loss (12) converges to the $t$SM objective (13) in the continuous-time limit (see Appendix D.4 for the proof).

**Proposition 4.1.** *Let* $t \in [0,T)$ *and* $\delta > 0$*, we have*

$$\lim_{\delta \to 0^+} \frac{8}{\delta^2}\left(\mathcal{L}_{\mathrm{clf}}(\theta; t, t + \delta) - \log 2\right) = \mathcal{L}_{t\mathrm{SM}}(\theta; t) + C , \quad (14)$$

*where* $C = \mathbb{E}_{p_t}\left[\left(\frac{\partial}{\partial t} \log p_t(Y_t)\right)^2\right]$ *is constant w.r.t* $\theta$*.*

**DRE-$\infty$.** However, as with the score, directly regressing the time-score is generally infeasible as it is usually intractable. (Choi et al., 2022, Proposition 4) extends Proposition 4.1 by showing that the optimal binary classifier associated with objective (12) (using a general classifier) can be used to approximate the time-score. In contrast, the proposed approach embeds the parameterized marginals directly into the classification problem, bypassing the need to approximate the time-score altogether.

**Conditional Time Score Matching.** In parallel, (Yu et al., 2025; Guth et al., 2025) propose an alternative objective, termed *Conditional Time Score Matching* (CtSM), which leverages the tractable conditional time score and enjoys the same gradients as the original formulation. For instance, in the DMs case where $X_t = S(t)X_0$, the time score can be expressed as $\partial \log p_t(y_t) = \mathbb{E}[\partial_t \log p_t(y_t|X_t)|Y_t = y_t]$ where the conditional time-score is

$$\partial_t \log p_t(Y_t|X_t) = -d\frac{\dot{\gamma}(t)}{\gamma(t)} - \frac{\partial_t X_t^\top Z_t}{\gamma(t)} + \|Z_t\|^2 \frac{\dot{\gamma}(t)}{\gamma(t)} , \quad (15)$$

where $Y_t = X_t + \gamma(t)Z_t$, leading to the following objective

$$\mathcal{L}_{\text{CtSM}}(\theta; t) = \mathbb{E}\left[\left(\partial_t \log p_t^\theta(Y_t) - \partial_t \log p_t(Y_t|X_t)\right)^2\right] , \quad (16)$$

with $X_t \sim q_t$ and $Y_t \sim p_t(\cdot|X_t)$. This derivation closely parallels that of DSM (see Section 2.1). Similar to our approach, (Choi et al., 2022) and (Guth et al., 2025) suggest combining this loss with DSM (7) to enhance model learning. Further, the conditional time score for SIs follows exactly the form in Equation (15); however, it remains intractable for general $(X_t)_t$ (see Appendix A for the proof and additional details).

### 4.1.2. LEARNING LOG-DENSITIES WITH SELF-CONSISTENCY.

Another strategy for estimating log-densities is to enforce *self-consistency* relations implied by the dynamics of $(Y_t)_t$. Two main approaches have been explored: one derived from the *Fokker–Planck equation* (FPE) and another from *Bayes' rule*. These methods typically require stronger assumptions on the process and are best understood in structured settings such as DMs or SIs. The detailed introduction and discussion are provided in Appendix A.4.

**Consistency via Fokker–Planck.** When $(Y_t)_t$ admits an SDE representation, its marginals follow the FPE, a partial differential equation governing the log-densities $(\log p_t)_t$. Several works (Sun et al., 2024; Shi et al., 2024) seek to penalize the log-density FPE deviations. While conceptually appealing, this requires backpropagating through time-derivatives, scores, and Laplacians, leading to high computational cost. Variants mitigate these issues by approximating derivatives with finite differences (Plainer et al., 2025). Moreover, Appendix C shows that such objectives remain subject to mode blindness, despite claims to the contrary.

**Consistency via Bayes' Rule.** An alternative derives from the relation between marginals at two times $s$ and $t$, $p_t(y_t)\, p_{s|t}(y_s|y_t) = p_s(y_s)\, p_{t|s}(y_t|y_s)$ for all $y_s, y_t \in \mathbb{R}^d$, where $p_{t|s}$ (resp. $p_{s|t}$) is the conditional distribution of $Y_t$ given $Y_s = y_s$ (resp. $Y_s$ given $Y_t = y_t$). Using approximations of the conditional distributions given by Euler–Maruyama integration of SDEs (8) or (9) (which depend on the score), one can regularize $(\log p_t^\theta)_t$ by enforcing approximate Bayes consistency (He et al., 2026a). However, this method remains valid only when the approximations are sufficiently accurate, which occurs for $s, t$ close together, precisely the regime where the objective is prone to mode blindness. In fact, Proposition A.1 in Appendix A.4 shows that the Bayes and FPE regularizations coincide asymptotically, inheriting the same limitations.

### 4.2. Connection to other training methods

**Maximum likelihood approaches.** While direct ML on a single distribution (3) is notoriously difficult, temporal correlations in $(Y_t)_t$ can alleviate this. (Noble et al., 2025) learn an annealing path $(p_t^\theta)t$, while (Zhang et al., 2023) model the joint time–state distribution and use Gibbs transitions across levels. In the DMs case, (Gao et al., 2021; Zhu et al., 2024) exploit tractable conditionals $p_{t|s}$ to model posteriors as EBM with $p_{s|t}^\theta \propto p_{t|s} \times p_t^\theta$, enabling more efficient ML when $t - s \to 0$. Despite their advantages, these methods still depend on costly sampling loops.

**Noise contrastive estimation.** The binary objective (12) closely resembles the well-known *Noise Contrastive Estimation* (NCE) framework (Gutmann & Hyvärinen, 2010), with the multi-class extension (10) being related to the generalization considered in (Matsuda & Hyvärinen, 2019). Intuitively, our formulation can be interpreted as using the marginal at time $t$ as the "noise distribution" when learning the density at $t'$, and vice versa, but in a fully parametric setting. This connection highlights an additional flexibility of our approach: whenever some marginal densities $p_t$ are known exactly (for example, $p_T$ in DMs, or $p_0$ and $p_1$ in SIs) these can be seamlessly incorporated into the learning framework, potentially improving accuracy.

## 5. Numerical experiments

We begin our numerical study by comparing `DiffCLF` with DSM and CtSM [6] on controlled high-dimensional Gaussian mixtures, and then demonstrate its effectiveness and scalability on complex molecular systems, before turning to the practical applications in Section 2.2.

**DMs and SIs on MOGs.** In the mixture of Gaussian (MOG) setting, the closed-form expression of $p_t$ is available

---

[6]For clarity, `DiffCLF` refers to $\mathcal{L}_{\text{DSM}} + \mathcal{L}_{\text{clf}}$ training and CtSM refers $\mathcal{L}_{\text{DSM}} + \mathcal{L}_{\text{CtSM}}$ training.

*Table 1.* **Comparison on synthetic 40-mode Gaussian mixtures**. A DM with variance preserving noising scheme was trained on MOG-40 using the different objectives. We explore different values of the number of levels $N \in \{2, 4, 8, 16\}$ and ensure equal computational comparison between methods. We report the classification loss (10), Fisher divergence (FD), and Maximum Mean Discrepancy (MMD) ($\times 100$) from the denoising SDE. The classification approach matches DSM in Fisher divergence and MMD, while yielding markedly better consistency in classification loss.

| Dim | $\mathcal{L}_{\text{clf}} + \mathcal{L}_{\text{DSM}}$ (ours) | | | $\mathcal{L}_{\text{C}t\text{SM}} + \mathcal{L}_{\text{DSM}}$ | | | $\mathcal{L}_{\text{DSM}}$ | | |
| --- | --- | --- | --- | --- | --- | --- | --- | --- | --- |
| | $\mathcal{L}_{\text{clf}}$ | FD | MMD | $\mathcal{L}_{\text{clf}}$ | FD | MMD | $\mathcal{L}_{\text{clf}}$ | FD | MMD |
| 8 | $4.41_{\pm 0.12}$ | $2.00_{\pm 1.48}$ | $0.69_{\pm 0.59}$ | $6.80_{\pm 0.86}$ | $5.74_{\pm 2.21}$ | $19.41_{\pm 0.77}$ | $9.19_{\pm 0.33}$ | $4.09_{\pm 3.89}$ | $0.99_{\pm 0.64}$ |
| 16 | $4.19_{\pm 0.14}$ | $2.81_{\pm 1.38}$ | $0.91_{\pm 0.32}$ | $8.33_{\pm 2.36}$ | $7.96_{\pm 2.11}$ | $22.62_{\pm 0.45}$ | $22.36_{\pm 0.76}$ | $5.49_{\pm 5.23}$ | $1.28_{\pm 0.56}$ |
| 32 | $4.04_{\pm 0.23}$ | $3.68_{\pm 1.47}$ | $1.20_{\pm 0.44}$ | $6.13_{\pm 1.45}$ | $10.30_{\pm 1.95}$ | $18.18_{\pm 1.51}$ | $85.07_{\pm 9.53}$ | $3.88_{\pm 3.49}$ | $1.20_{\pm 0.42}$ |
| 64 | $4.01_{\pm 0.46}$ | $4.87_{\pm 1.95}$ | $2.18_{\pm 1.02}$ | $7.78_{\pm 1.64}$ | $9.96_{\pm 1.84}$ | $12.67_{\pm 3.66}$ | $149.45_{\pm 33.76}$ | $3.93_{\pm 3.48}$ | $1.51_{\pm 0.15}$ |
| 128 | $4.40_{\pm 1.00}$ | $6.91_{\pm 2.47}$ | $3.54_{\pm 1.34}$ | $20.86_{\pm 4.93}$ | $9.42_{\pm 1.82}$ | $5.20_{\pm 0.34}$ | $383.53_{\pm 35.99}$ | $6.78_{\pm 5.94}$ | $1.99_{\pm 0.35}$ |

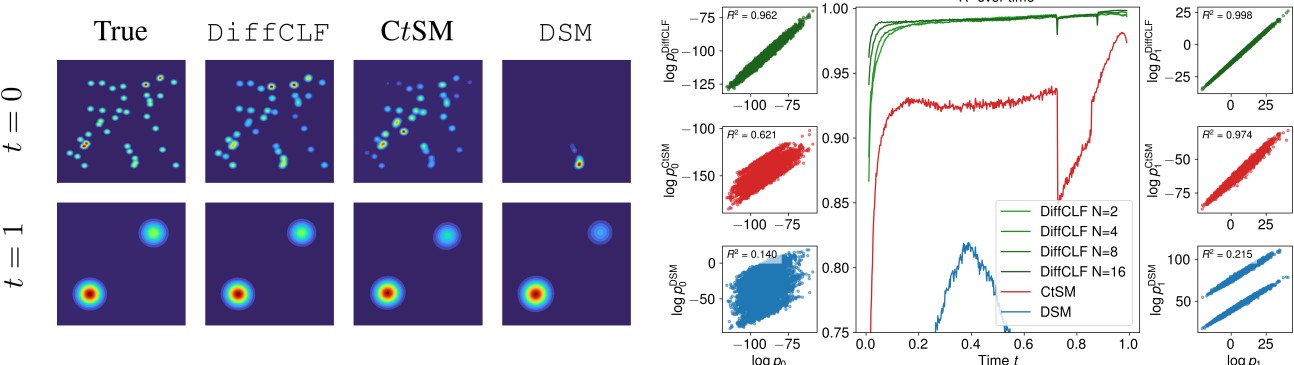

*Figure 3.* **Learned EBMs with SI between a bi-modal and a 40-mode Gaussian mixture.** We use $\mathcal{L}_{\text{DSM}}$, $\mathcal{L}_{\text{DSM}} + \mathcal{L}_{\text{C}t\text{SM}}$, and $\mathcal{L}_{\text{DSM}} + \mathcal{L}_{\text{clf}}$ (DiffCLF, ours). **(Left, $d = 2$):** Learned densities at $t = 0$ (top row) and $t = 1$ (bottom row) for the different methods, showing that DiffCLF best captures the target distributions. **(Right, $d = 128$):** Comparison of learned log-densities $\log p_t^\theta$ versus the exact $\log p_t$ on exact samples from $(Y_t)_t$ across time in terms of scatter plots and $R^2$ statistic. Plots at the left and right edges correspond to $t = 0$ and $t = 1$, respectively; the middle shows the coefficient of determination $R^2$ over $t \in (0, 1)$, indicating that only DiffCLF achieves consistently high agreement with the true log-densities.

(see Appendix H.2), allowing us to quantitatively assess approximation errors of different training objectives. In Table 1, we train diffusion models on the 40-mode mixture (MOG-40) across increasing dimensions. We evaluate the trained models using three metrics: the classification loss (10), which the optimal model should minimize, the Fisher Divergence (FD), measuring accuracy of the learned score, and the Maximum Mean Discrepancy (MMD) (Gretton et al., 2012), reflecting the quality of generated samples. While all objectives achieve comparable FD and MMD, showing that DiffCLF is compatible to the DSM objective that would not degenerate generation quality, our method is the only one that consistently achieves low values of the classification loss, thereby satisfying the self-consistency condition that other approaches fail to capture. We additionally report the ESS which reflects the energy quality, and other metrics for the generation quality in Tables 5 and 6.

In Figure 3, SIs are trained to bridge MOG-40 and a 2-mode mixture (MOG-2) in 2D and 128D respectively. The

figure demonstrates that DiffCLF learns significantly more accurate energies than the baselines. Additional results, including comparison with ground truth marginal densities using the importance sampling effective sample size, and experimental details are provided in Appendix H.3.

**Evaluation on molecular systems.** We further conduct experiments on training energy-based DMs for molecular systems, including Müller-Brown potential, Alanine Dipeptide, and Chignolin. We train DMs with $\mathcal{L}_{\text{DSM}}$ only and $\mathcal{L}_{\text{DSM}} + \mathcal{L}_{\text{clf}}$, then evaluate the learned energy by running molecular dynamics using $\text{U}_{t=0}^\theta$, where we follow the same settings as Plainer et al. (2025). Figures 4 and 9 and Table 11 demonstrate the effectiveness and scalability of DiffCLF. The results demonstrate that DiffCLF aids better learning of energies without degenerating diffusion sampling, achieving comparable performance against the Fokker-Planck regularization proposed by Plainer et al. (2025) while accelerating training by 1.5-2.5$\times$. Details of experiment can be found in Appendix H.7.

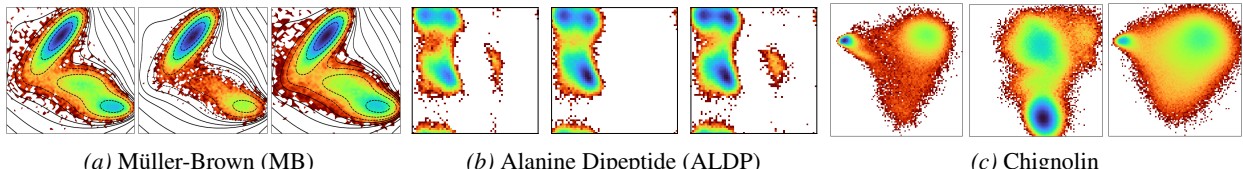

*(a)* Müller-Brown (MB)   *(b)* Alanine Dipeptide (ALDP)   *(c)* Chignolin

*Figure 4.* Samples from Langevin dynamics with $U^\theta_{t=0}$ on three benchmarks. **(Left)**: Reference; **(Middle)**: DSM; **(Right)**: `DiffCLF`. For MB we show the sample histogram; for ALDP, the torsion-angle histogram (x: $\phi$, y: $\psi$); and for Chignolin, the histogram of the first two TIC axes.

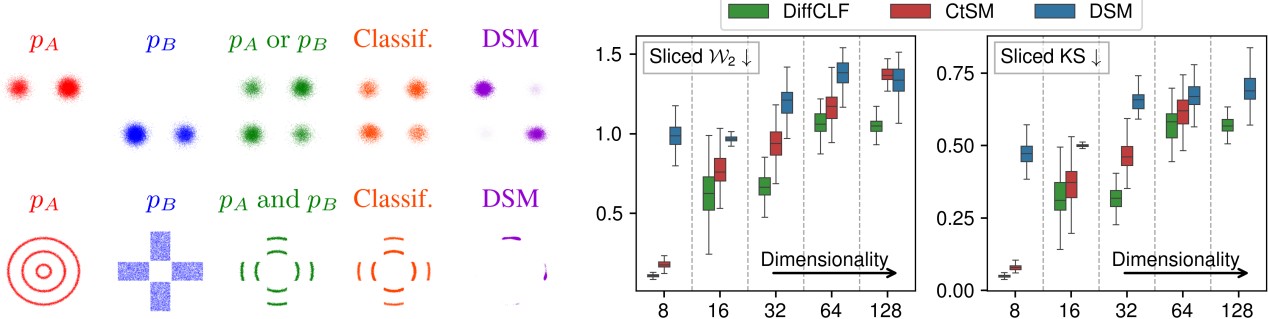

*Figure 5.* **(Left) OR and AND model composition.** *Top:* OR composition, *Bottom:* AND composition. Red/Blue: input distributions, Green: ground truth, Orange: `DiffCLF`, Purple: DSM. Results obtained via 512-step SMC on the product of learned marginals. **(Right) SMC-based BG metrics.** Box plots of Sliced Wasserstein ($\mathcal{W}_2$) and Kolmogorov-Smirnov (KS) distances for a 512-step SMC on the SI between MOG-40 and MOG-2. Optimal scores and velocities are used for kernels, with learned EBMs for marginals. `DiffCLF` consistently outperforms other methods .

**Composition.**   Following (Du et al., 2023; Thornton et al., 2025), we evaluate `DiffCLF` against DSM on two toy composition tasks shown on the left of Figure 5 : an "OR" between two Gaussian mixtures with different weights (top row) and an "AND" between a mixture of rings and uniform rectangles (bottom row). We perform composition using standard SMC (Doucet et al., 2001; Del Moral et al., 2006) with a Metropolized Langevin (MALA) (Roberts & Tweedie, 1996) kernel applied to either the mixture or the product of the learned densities. In this toy setting, this strategy consistently outperformed the annealed Langevin approach of (Du et al., 2023) and the diffusion–SMC approach of (Thornton et al., 2025). As seen in the two right-most columns, models trained with `DiffCLF` produce substantially better results than DSM, particularly in preserving the correct proportions of each region, a direct consequence of avoiding mode blindness. Details are in Appendix H.4.

**Boltzmann Generators.**   To demonstrate the potential of using trained EBMs for building BGs, we consider the energy-based SI between MOG-40 and MOG-2 by embedding the learned energy into the SMC framework of (Phillips et al., 2024), which exploits integration of the SDE (9) to enhance transitions between levels. Specifically, the algorithm is provided with learned energies from either `DiffCLF`, C$t$SM, or DSM, with the last marginal constrained post-training to match the true distribution. To

focus the comparison solely on the energies, we use the analytical velocity and score (see Appendix H.2) to calculate the densities of transitions. As shown in the right part of Figure 5, `DiffCLF` yields substantially more accurate samples than DSM and C$t$SM in all dimensions, highlighting its advantage for building BGs.

**Free energy difference estimation.**   In this experiment, SIs are trained to enhance the accuracy of free-energy difference estimation via TI. The experiments focus on the alanine dipeptide system in implicit solvent (ALDP-imp) and in vacuum (ALDP-vac) at temperature $T = 300K$, with the goal of estimating their free-energy difference through TI. Models are trained using $\mathcal{L}_{\text{base}} + \mathcal{L}_{\text{clf}}$, where $\mathcal{L}_{\text{base}}$ denotes the objective introduced in (Máté et al., 2025). The resulting estimates [7] are reported in Table 2. The estimate produced by `DiffCLF` exhibits improved accuracy.  *

*Table 2.* ALDP solvation free energy estimated with thermodynamic integration.

| Method | Estimation |
|---|---|
| Reference | $29.43_{\pm 0.01}$ |
| TI w/ $\mathcal{L}_{\text{base}}$ (Máté et al., 2025) | $27.30_{\pm 0.45}$ |
| TI w/ $\mathcal{L}_{\text{base}} + \mathcal{L}_{\text{clf}}$ (ours) | $\mathbf{29.02}_{\pm 0.41}$ |

[7]The reference value is from (Du et al., 2025).

# 6. Conclusion

Energy-based generative models provide a compelling perspective on score-based methods, extending their utility well beyond pure sample generation to a wide range of downstream applications. Despite this promise, the existing literature on energy-based training remains limited, with available methods often being computationally demanding, hyperparameter-sensitive, or biased. This work introduced the *Diffusive Classification* (`DiffCLF`) objective, a simple, efficient, and consistent training principle. The method is broadly applicable across different stochastic processes and integrates seamlessly with existing approaches such as DSM, offering both theoretical clarity and practical flexibility. Empirical results demonstrate clear advantages of `DiffCLF`, resulting in more accurate and consistent energy estimates, which in turn improves model composition and model-induced Boltzmann Generators. Nonetheless, our experiments are limited in scale. Exploring applications to large-scale tasks such as image modeling, where SMC-based composition has already shown promise (Thornton et al., 2025), constitutes an exciting direction for future work. Another promising extension lies in the discrete domain: since `DiffCLF` remains valid for general stochastic processes, including Continous-Time Markov Chain, applying it to textual modeling appears especially compelling in light of recent advances at the intersection of EBMs and DMs (Xu et al., 2025).

# Acknowledgement

The authors would like to thank Omar Chehab, Jiajun He, and Hanlin Yu, for the fruitful discussions and feedbacks, and Michael Plainer for explaining details in their codebase for the molecular experiments. RKOY acknowledges the UK Engineering and Physical Sciences Research Council (EPSRC) grant EP/L016516/1 for the University of Cambridge Centre for Doctoral Training, the Cambridge Centre for Analysis. LG acknowledges government funding managed by the French National Research Agency under France 2030, reference ANR-23-IACL-0005. JMHL acknowledges support from EPSRC funding under grant EP/Y028805/1. JMHL also acknowledges support from a Turing AI Fellowship under grant EP/V023756/1. This work was performed using HPC resources from GENCI–IDRIS (AD011015234R1). This project acknowledges the resources provided by the Cambridge Service for Data-Driven Discovery (CSD3) operated by the University of Cambridge Research Computing Service (www.csd3.cam.ac.uk), provided by Dell EMC and Intel using Tier-2 funding from the Engineering and Physical Sciences Research Council (capital grant EP/T022159/1), and DiRAC funding from the Science and Technology Facilities Council (www.dirac.ac.uk).

# Impact Statement

This work advances the methodology for training energy-based models, improving their reliability and applicability in scientific and machine learning settings. By enabling more accurate density estimation and compositional sampling, the proposed approach may support downstream applications such as simulation, probabilistic modeling, and scientific discovery. As with generative modeling techniques in general, misuse in deceptive or biased data generation is a potential risk, but this work does not introduce new risks beyond those already present in the field. We do not foresee immediate negative societal impacts arising specifically from this contribution.

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

# A Diffusive Classification Loss
# for Learning Energy-based Generative Models
# Appendix

## Outline of Appendix

- Appendix A introduces additional backgrounds, where

  - Appendix A.1 introduces Itô SDEs, forward/backward processes, and the Fokker Planck Equation;
  - Appendix A.2 provides details on the parameterization of noising processes in Diffusion models;
  - Appendix A.3 introduces the Time Score and Conditional Time Score Matching for Diffusion models, with an extension for Stochastic Interpolants;
  - Appendix A.4 introduces two related works for learning energies with self-consistency, including one from Fokker-Planck Equation (Appendix A.4.1) , the other one from Bayes' rule (Appendix A.4.2), and their connection (Appendix A.4.3).

- Appendix B provides connections to related works

  - Appendix B.1 connects `DiffCLF` to other methods that train energy-based Diffusion models;
  - Appendix B.2 connects `DiffCLF` to Multi-state Bennett Acceptance Ratio.

- Appendix C illustrates the mode-blindness issue of Score Matching and Time Score Matching.

- Appendix D provides proof of propositions

  - Appendix D.1 provides proof of Proposition 3.1;
  - Appendix D.2 provides proof of Proposition 3.2;
  - Appendix D.3 provides proof of Proposition 3.3;
  - Appendix D.4 provides proof of Proposition 4.1;
  - Appendix D.5 provides proof of Proposition A.1.

- Appendix E extends `DiffCLF` with Bregman divergence.

- Appendix F extends `DiffCLF` to discrete Diffusion models.

- Appendix H elaborates experimental details.

  - Appendix H.2 introduces closed-form expressions for Diffusion models and Stochastic Interpolants for Mixture-of-Gaussians;
  - Appendix H.3 provides details of training energy-based Diffusion models for Mixture-of-Gaussians;
  - Appendix H.4 provides details on experiments of model compositions;
  - Appendix H.5 provides details on experiments of Boltzmann generators;
  - Appendix H.6 provides details on experiments of free energy estimation;
  - Appendix H.7 provides details on experiments of molecular systems.

# A. Additional background

## A.1. Background and Foundations for Itô SDEs

In this section, we will introduce the *Itô SDEs*, its *time reversal*, and the *Fokker Planck Equation*.

**Forward and Backward Processes of Itô SDEs.** Given $f : [0, T] \times \mathbb{R}^d \to \mathbb{R}^d$ and $g : [0, T] \times \mathbb{R}^d \to \mathbb{R}^d$ with regularity, the Itô SDE is defined as

$$\mathrm{d}X_t = f(t, X_t)\mathrm{d}t + g(t, X_t)\mathrm{d}W_t, \quad t \in [0, T] , \tag{17}$$

where $(W_t)_{t \in [0,T]}$ is the standard Wiener process. With additional regularity on $f$ and $g$ (see (Anderson, 1982)), its *time reversal* is given as

$$\mathrm{d}X_t = \left[ f(t, X_t) - g(t, X_t)g(t, X_t)^\top \nabla \log p_t(X_t) \right] \mathrm{d}t + g(t, X_t)\mathrm{d}\tilde{W}_t , \tag{18}$$

where $(\tilde{W}_t)_{t \in [0,T]}$ is the time-reversed Wiener process. We denote $(p_t)_{t \in [0,T]}$ as the marginal distributions admitted by the SDE, starting either from $(t = 0, X_0 \sim p_0)$ or $(t = T, X_T \sim p_T)$. For simplicity, in the rest of context, we consider the *Additive-noise Itô SDEs*, *i.e.* with $g(t, x) = g(t)$, which gives

$$\text{(forward SDE)} \quad \mathrm{d}X_t = f(t, X_t)\mathrm{d}t + g(t)\mathrm{d}W_t , \tag{19}$$

$$\text{(backward SDE)} \quad \mathrm{d}X_t = \left[ f(t, X_t) - g(t)^2 \nabla \log p_t(X_t) \right] \mathrm{d}t + g(t)\mathrm{d}\tilde{W}_t . \tag{20}$$

**Fokker Planck Equation.** The *Fokker Planck Equation* (FPE), which is also known as the *Forward Kolmogorov Equation*, describes the density change along the forward SDE as follows

$$\partial_t p_t(x) = -\nabla \cdot (f(t, x)p_t(x)) + \frac{g(t)^2}{2}\Delta \cdot p_t(x) , \tag{21}$$

where $\nabla$ is the divergence operator and $\Delta$ is the Laplacian operator both with respect to the variable $x$. Its log-density version, *i.e. log-density FPE* is given by

$$\partial_t \log p_t(x) = -\nabla \cdot f(x, t) - f(x, t)^\top \nabla \log p_t(x) + \frac{g(t)^2}{2} \left[ \nabla \cdot \nabla \log p_t(x) + \|\nabla \log p_t(x)\|^2 \right] . \tag{22}$$

## A.2. On the practical parameterization of noising processes of DMs

In this part, we will introduce the parameterization of noising processes in DMs, which is introduced by (Karras et al., 2022). Recall the noising SDE defined as

$$\mathrm{d}Y_t = f(t)Y_t\mathrm{d}t + g(t)\mathrm{d}W_t ,$$

where $(W_t)_t$ is a standard Brownian motion. The noising kernel, $p_{t|0}(y_t|y_0)$, is a Gaussian distribution with closed-form expressions of the mean and covariance due to the linearity of drift (Song et al., 2021b; Karras et al., 2022):

$$p_{t|0}(y_t|y_0) = \mathcal{N}(y_t; S(t)y_0, \Sigma(t)) ,$$

where

$$S(t) = \exp\left( \int_0^t f(u)\mathrm{d}u \right), \quad \Sigma(t) = \left[ S(t)^2 \int_0^t \frac{g(u)^2}{S(u)^2}\mathrm{d}u \right] \mathrm{I}_d . \tag{23}$$

Let $\sigma(t)^2 = \int_0^t (g(u)^2/S(u)^2)\mathrm{d}u$, we have

$$\dot{S}(t) = \frac{\mathrm{d}}{\mathrm{d}t}S(t) = \frac{\mathrm{d}}{\mathrm{d}t}\exp\left( \int_0^t f(u)\mathrm{d}u \right) = S(t)f(t) \implies f(t) = \frac{\dot{S}(t)}{S(t)} , \tag{24}$$

$$\dot{\sigma}(t) = \frac{\mathrm{d}}{\mathrm{d}t}\sigma(t) = \frac{\mathrm{d}}{\mathrm{d}t}\sqrt{\int_0^t \frac{g(u)^2}{S(u)^2}\mathrm{d}u} = \frac{1}{2\sigma(t)}\frac{g(t)^2}{S(t)^2} \implies g(t) = S(t)\sqrt{2\dot{\sigma}(t)\sigma(t)} . \tag{25}$$

**Variance Preserving.** The *Variance Preserving* (VP) SDE (Song et al., 2021b) is defined as

$$f(t) = -\frac{1}{2}\beta(t), \quad g(t) = \sqrt{\beta(t)}, \quad \text{and } \beta(t) = \beta_{\min} + t(\beta_{\max} - \beta_{\min}) \, ,$$

that is,

$$S(t) = \exp\left(\int_0^t -\frac{1}{2}\beta(u)\mathrm{d}u\right) = \exp\left(-\frac{1}{2}\left(\frac{\beta_{\max} - \beta_{\min}}{2}t^2 + \beta_{\min}t\right)\right) \, ,$$

$$\sigma(t)^2 = \int_0^t \frac{\beta_{\min} + (\beta_{\max} - \beta_{\min})u}{\exp\left(-\frac{\beta_{\max}-\beta_{\min}}{2}u^2 - \beta_{\min}u\right)}\mathrm{d}u = \exp\left(\frac{1}{2}\left(\frac{\beta_{\max} - \beta_{\min}}{2}t^2 + \beta_{\min}t\right)\right) - 1 \, .$$

**Variance Exploding.** The *variance Exploding* (VE) SDE (Song et al., 2021b) is defined as

$$f(t) \equiv 0, \quad g(t) = \sigma_{\min}\sqrt{2\log\sigma_\mathrm{d}}\sigma_\mathrm{d}^t, \quad \text{and } \sigma_\mathrm{d} = \sigma_{\max}/\sigma_{\min} \, ,$$

that is,

$$S(t) = \exp\left(\int_0^1 0\mathrm{d}u\right) = 1 \, ,$$

$$\sigma(t)^2 = \int_0^1 \frac{2\sigma_{\min}^2\sigma_\mathrm{d}^{2u}\log\sigma_\mathrm{d}}{1}\mathrm{d}u = \sigma_{\min}^2\left(\sigma_\mathrm{d}^{2t} - 1\right) \, .$$

### A.3. Conditional Time Score

This section recaps and extends the analyses of (Guth et al., 2025) and (Yu et al., 2025), providing a unified presentation of the conditional time score framework. Using the stochastic process introduced in Equation (1), the log-density evolution can be written as

$$\partial_t \log p_t(y) = \partial_t \log \int q_t(x)p_t(y|x)\mathrm{d}x \tag{26}$$

$$= \frac{1}{p_t(y)}\left(\int p_t(y|x)\partial_t q_t(x) + q_t(x)\partial_t p_t(y|x)\mathrm{d}x\right)$$

$$= \int \frac{q_t(x)p_t(y|x)}{p_t(y)}(\partial_t \log q_t(x) + \partial_t \log p_t(y|x))\mathrm{d}x$$

$$= \mathbb{E}\left[\partial_t \log q_t(X_t) + \partial_t \log p_t(Y_t|X_t)|Y_t = y\right] \, ,$$

where

$$\partial_t \log p_t(y|x) = \partial_t \log \mathcal{N}(y; x, \gamma(t)^2 I)$$

$$= \partial_t\left(-\frac{d}{2}\log 2\pi - d\log\gamma(t) - \frac{\|y - x\|^2}{2\gamma(t)^2}\right)$$

$$= -d\frac{\dot\gamma(t)}{\gamma(t)} + \frac{\|y - x\|^2}{\gamma(t)^2}\frac{\dot\gamma(t)}{\gamma(t)} \, ,$$

while $\partial_t \log q_t(X_t)$ doesn't have a general form and therefore should be treated case-by-case.

In this work, we mainly discuss special cases of Equation (1), where $(X_t)_{t\in[0,T]}$ is deterministic once the source(s), *i.e.* $X_0$ for DMs and $(X_0, X_1)$ for SIs, are given. For simplicity, we define $\xi \sim \mu$ as the source and $X_t = T_t(\xi)$ is obtained by a deterministic map $T_t$ (see examples bellow). In such case, the marginal of $X_t$ can be defined by a Dirac delta

$$q_t(x) = \int \mu(\xi)\delta(x - T_t(\xi))\mathrm{d}\xi \Rightarrow p_t(y) = \int \mu(\xi)p_t(y|T_t(\xi))\mathrm{d}\xi \, .$$

Therefore, Equation (26) can be written as

$$
\begin{aligned}
\partial_t \log p_t(y) &= \partial_t \log \int \mu(\xi) p_t(y|T_t(\xi)) \mathrm{d}\xi \\
&= \frac{1}{p_t(y)} \int \mu(\xi) p_t(y|T_t(\xi)) \partial_t \log p_t(y|T_t(\xi)) \mathrm{d}\xi \\
&= \int p(\xi|y) \partial_t \log \mathcal{N}(y; T_t(\xi), \gamma(t)^2 I) \mathrm{d}\xi \\
&= \mathbb{E}\left[ -d\frac{\dot{\gamma}(t)}{\gamma(t)} - \frac{(Y_t - T_t(\xi))^\top \partial_t T_t(\xi)}{\gamma(t)^2} + \frac{\|Y_t - T_t(\xi)\|^2}{\gamma(t)^2} \frac{\dot{\gamma}(t)}{\gamma(t)} \Big| Y_t = y \right] .
\end{aligned}
$$

**Example 1: Diffusion Models** As introduced in Section 2.1, $(X_t)_{t \in [0,T]}$ in DMs are defined as $X_t = S(t)X_0$ with $X_0 \sim \pi$ and $S : \mathbb{R} \mapsto \mathbb{R}$. Therefore, $\xi = X_0$, $\mu = \pi$, and $T_t(\xi) = S(t)\xi$:

$$
\partial_t \log p_t(y) = \mathbb{E}\left[ -d\frac{\dot{\gamma}(t)}{\gamma(t)} - \frac{\dot{S}(t)X_0^\top(y - S(t)X_0)}{\gamma(t)^2} + \frac{\|y - S(t)X_0\|^2}{\gamma(t)^2} \frac{\dot{\gamma}(t)}{\gamma(t)} \right]. \tag{27}
$$

**Example 2: Stochastic Interpolants** As introduced in Section 2.1, SIs are defined by two ends, *i.e.* $X_t = I_t(X_0, X_1)$ given $(X_0, X_1)$. Therefore, $\xi = (X_0, X_1)$, $\mu$ any coupling of $p_0, p_1$ with marginals $p_0, p_1$, and $T_t(\xi_0, \xi_1) = I_t(\xi_0, \xi_1)$.

$$
\partial_t \log p_t(y) = \mathbb{E}\left[ -d\frac{\dot{\gamma}(t)}{\gamma(t)} - \frac{\partial_t I_t(X_0, X_1)^\top(y - I_t(X_0, X_1))}{\gamma(t)^2} + \frac{\|y - I_t(X_0, X_1)\|^2}{\gamma(t)^2} \frac{\dot{\gamma}(t)}{\gamma(t)} \right]. \tag{28}
$$

### A.4. Learning Log-densities with self-consistency

In this section, we will introduce two strategies to learn log-densities via enforcing their self-consistency relations: the Fokker-Planck regularization (Sun et al., 2024; Shi et al., 2024; Plainer et al., 2025) and the Bayes (or RNE) regularization (He et al., 2026a). Such relations naturally arise from the underlying dynamics of the process and can be exploited to design training objectives. These methods typically rely on stronger assumptions about the generative process $Y_t$, which is why we primarily focus on the well-structured settings of DMs and SIs.

#### A.4.1. CONSISTENCY FROM FOKKER–PLANCK EQUATION

Assume that the dynamic of the process $(Y_t)_t$ can be described by an (additive-noise) Itô SDE

$$
dY_t = \alpha_t(Y_t)\mathrm{d}t + \beta_t \mathrm{d}W_t, \tag{29}
$$

where $\alpha_t : [0,T] \times \mathbb{R}^d \to \mathbb{R}^d$ and $\beta_t : [0,T] \to \mathbb{R}_+$. Then the evolution of densities induced by this process is governed by the *Fokker-Planck Equation* (FPE), (see Appendix A.1 for more details) given by

$$
\partial_t p_t + \nabla \cdot (\alpha_t p_t) - \frac{\beta_t^2}{2} \Delta p_t = 0 , \tag{30}
$$

where $\nabla\cdot$ is the divergence operator and $\Delta$ is the Laplacian operator both with respect to $y$. This formulation arises when the stochastic process admits an SDE representation. For example, in Diffusion Models we have $\alpha_t(y) = f(t)y$ and $\beta_t = g(t)$, while in Stochastic Interpolants $\alpha_t(y) = \mathbb{E}[\partial_t I_t(Y_0, Y_1) + \dot{\gamma}(t)Z|Y_t = y] + \frac{1}{2}\beta_t^2 \nabla \log p_t(y)$ and $\beta_t = g(t)$ [8]. Although the Fokker–Planck equation (30) is formulated in terms of the density $p_t$, it can equivalently be rewritten in terms of the log-density $\log p_t$ as

$$
\partial_t \log p_t - \mathcal{F}_t(p_t) = 0, \quad \mathcal{F}_t(p_t) = \frac{\beta_t^2}{2}\left[ \Delta \log p_t(y) + \|\nabla p_t(y)\|^2 \right] - \alpha_t \cdot \nabla \log p_t - \nabla \cdot \alpha_t .
$$

---

[8]Note that $\alpha_t = v_t$ and $\beta_t = \sqrt{2\dot{\gamma}(t)\gamma(t)}$ is also a valid choice but we keep the previous decomposition to make the following proofs easier.

To enhance the accuracy of $(p_t^\theta)_t$, recent works (Sun et al., 2024; Shi et al., 2024; Plainer et al., 2025) propose enforcing its self-consistency by optimizing the following objective

$$\mathcal{L}_{\mathrm{FPE}}(\theta) = \mathbb{E}_t[\mathcal{L}_{\mathrm{FPE}}(\theta; t)], \quad \mathcal{L}_{\mathrm{FPE}}(\theta; t) = \mathbb{E}_{p_t}\left[\left(\partial_t \log p_t^\theta(Y_t) - \mathcal{F}_t(p_t^\theta)(Y_t)\right)^2\right] . \tag{31}$$

Although (Shi et al., 2024) claim that this approach overcomes the blindness of score matching, we demonstrate in Appendix C that it remains susceptible to the same issue. The objective can also be combined with DSM, and a pretrained score estimator may be directly incorporated into $\mathcal{F}$ without further optimization. However, the method is computationally demanding, as training requires back-propagating through high-order derivatives of $\mathrm{U}^\theta$ (specifically, the time derivative, the score, and the Laplacian) resulting in a substantial increase in cost. To mitigate this, (Plainer et al., 2025) propose approximating the time-score using finite differences and estimating the residual term $\mathcal{F}_t$ with an unbiased estimator.

### A.4.2. CONSISTENCY FROM BAYES RULE

A complementary perspective arises by considering the correlation between the distribution at two consecutive times $0 < s < t < T$. In this case, the marginal densities of $Y_s$ and $Y_t$ are connected through Bayes' rule

$$p_t(y_t)p_{s|t}(y_s|y_t) = p_s(y_s)p_{t|s}(y_t|y_s), \quad \text{for all } y_s, y_t \in \mathbb{R}^d , \tag{32}$$

where $p_{t|s}$ and $p_{s|t}$ denote the conditional distributions of $Y_t$ given $Y_s$ and $Y_s$ given $Y_t$, respectively. Although these conditional distributions are generally intractable, they admit tractable approximations when $(Y_t)_t$ is defined as the solution of an SDE, as in DMs or SIs. We should first note that the time-reversal, which generally exists for DMs or SIs, of Equation (29) is written as

$$\mathrm{d}Y_t = [\alpha_t(Y_t) - \beta_t^2 \nabla \log p_t(Y_t)]\mathrm{d}t + \beta_t \mathrm{d}\tilde{W}_t , \tag{33}$$

where $(\tilde{W}_t)_t$ is a standard Brownian motion in reversed time. By letting $\delta = t - s$, one can approximate the transition kernels via an Euler–Maruyama (EM) discretization of the corresponding SDEs (8) and (9), yielding

$$p_{t|s}(\cdot|y_s) \approx \mathcal{N}\left(y_s + \delta\alpha_s(y_s), \ \delta\beta_s^2 \mathrm{I}_d\right) , \tag{34}$$

$$p_{s|t}(\cdot|y_t) \approx \mathcal{N}\left(y_t - \delta\left[\alpha_t(y_t) - \beta_t^2\nabla \log p_t(y_t)\right], \ \delta\beta_t^2 \mathrm{I}_d\right) , \tag{35}$$

where, letting $\tilde{p}_{t|s}(\cdot|y_s)$ and $\tilde{p}_{s|t}(\cdot|y_t)$ denote the Gaussian approximation kernels on the right-hand sides of equation 34 and equation 35 respectively, the local weak approximation error can be quantified as

$$\mathbb{E}_{Y_t \sim p_{t|s}(\cdot|y_s)}[h(Y_t)] = \mathbb{E}_{\tilde{Y}_t \sim \tilde{p}_{t|s}(\cdot|y_s)}[h(\tilde{Y}_t)] + \mathcal{O}(\delta^2) , \tag{36}$$

$$\mathbb{E}_{Y_s \sim p_{s|t}(\cdot|y_t)}[h(Y_s)] = \mathbb{E}_{\tilde{Y}_s \sim \tilde{p}_{s|t}(\cdot|y_t)}[h(\tilde{Y}_s)] + \mathcal{O}(\delta^2) , \tag{37}$$

for any sufficiently smooth test function $h$ with polynomial growth, provided that the drift $\alpha$, diffusion $\beta$, and score function $\nabla \log p$ satisfy standard Lipschitz and regularity conditions. In (He et al., 2026a), the authors propose to regularize the sequence $(\log p_t^\theta)_t$ by minimizing the squared discrepancy between the logarithm of the two sides of the Bayes rule (32)

$$\mathcal{L}_{\mathrm{Bayes}}(\theta) = \mathbb{E}_{s,t}[\mathcal{L}_{\mathrm{Bayes}}(\theta; s, t)] ,$$

with

$$\mathcal{L}_{\mathrm{Bayes}}(\theta; s, t) = \mathbb{E}_{p_s, p_t}\left[\left(\log p_s^\theta(Y_s) - \log p_t^\theta(Y_t) + \log p_{t|s}^\theta(Y_t|Y_s) - \log p_{s|t}^\theta(Y_s|Y_t)\right)^2\right] ,$$

where $p_{t|s}^\theta$ and $p_{s|t}^\theta$ denote approximations of $p_{t|s}$ (34) and $p_{s|t}$ (35), obtained by replacing $\nabla \log p_t$ with its approximation $\nabla \log p_t^\theta$.

As highlighted in (He et al., 2026a), for DMs, the specific choices of $f$ and $g$ allow a closed-form expression for the forward-time kernel $p_{t|s}$. This yields a more accurate approximation of the time-forward kernel than the EM scheme (34) and avoids reliance on intractable quantities such as the score. However, even in this favorable case, the time-backward kernel (35) remains approximate, introducing bias that breaks self-consistency. Exact self-consistency is only recovered in the small-step limit ($\delta \to 0$), where the EM scheme becomes accurate.

*Table 3.* Comparison of properties between `DiffCLF` (ours) and related methods. For *Number of Function Evaluation* (NFE), we report the actual evaluations when jointly training with $\mathcal{L}_{\text{DSM}}$, where we treat auto-differentiation costing roughly twice a network forward pass. For DRE-$\infty$, though (Choi et al., 2022) proposes to directly parameterize the score, we choose to treat it a way for energy-based training and therefore report the NFE for energy-parameterization, which doubles the original calculation. For FPE-reg, we follow the approximations made in (Plainer et al., 2025) and count their NFE.

| Method | NFE | No approx. | Prior knowledge of $(p_t)_t$ | mode-weight aware |
|---|---|---|---|---|
| DRE-$\infty$ (Choi et al., 2022) | 4 | ✗ | – | ✗ |
| CtSM (Yu et al., 2025; Guth et al., 2025) | 2 | ✓ | $\partial_t X_t$ | ✗ |
| FPE-reg (Plainer et al., 2025) | 14 | ✗ | Induced from known FPE | ✗ |
| Bayes-reg (He et al., 2026a) | 3 | ✗ | $p_{t|s}$ & $p_{s|t}$ | ✗ |
| $N$-`DiffCLF` (ours) | $N+1$ | ✓ | – | ✓ |

### A.4.3. BAYES AND FOKKER-PLANK REGULARIZATIONS: DISCRETE V.S. CONTINUOUS

Now, we are going to show that $\mathcal{L}_{\text{Bayes}}$ and $\mathcal{L}_{\text{FPE}}$ are asymptotically related as follows:

**Proposition A.1.** *Let $\delta > 0$. In the small step-size regime, the Bayes objective $\mathcal{L}_{\text{Bayes}}$ recovers the Fokker–Planck regularization $\mathcal{L}_{\text{FPE}}$, i.e.,*

$$\lim_{\delta \to 0} \frac{1}{\delta} \mathcal{L}_{\text{Bayes}}(\theta; t, t+\delta) = \mathcal{L}_{\text{FPE}}(\theta; t) . \tag{38}$$

The proof is provided at appendix D.5. This result is also closely related to the derivations in (Skreta et al., 2025b, Appendix D.2). Building on this observation, Proposition A.1 implies that $\mathcal{L}_{\text{Bayes}}$ is either in the small step-size regime, where it remains mode blind, or in the non-small step-size regime, where it becomes biased.

## B. Additional connection to related works

### B.1. Connection to related works in training energy-based generative models

Table 3 summurizes the computational overhead, assumptions/requirements, and mode-blindness issue for those methods.

DRE-$\infty$ (Choi et al., 2022) and CtSM (Yu et al., 2025; Guth et al., 2025), and FPE-reg (Plainer et al., 2025) train models such that their time derivatives match the ground-truth ones, which is termed Time Score Matching (tSM) (Choi et al., 2022). Jointly training with DSM, the optimality yields $\nabla \log p_t^\theta = \nabla \log p_t$ and $\partial_t \log p_t^\theta = \partial_t \log p_t$. However, it is not sufficient to reach the optimality of log density, i.e. $\log p_t^\theta = \log p_t$, especially when the modes are disconnected (see Appendix C for more discussions).

Bayes-reg (or RNE He et al., 2026a) leverages the transition kernels for energy-based training, which is guaranteed to reach optimal $(p_t^\theta)_t$ if training with arbitrary time pairs $(t, t')$. However, in practical cases such as DMs or SIs, the transition kernels can only be approximated when $t$ and $t'$ close enough, through a Euler–Maruyama discretization of the dynamic's SDE. Proposition A.1 shows that when $t$ and $t'$ are close enough, Bayes-reg recovers FPE-reg, and therefore, it might share the same issues in tSM.

`DiffCLF` treats the energy training as classification tasks, which, in the 2-class case, recovers tSM in the continuous-time limit (see Proposition 4.1). In fact, when $(t, t')$ in Equation (12) are close enough, one could approximate

$$\log p_{t'}^\theta(x) = \log p_t^\theta(x) + \partial_t \log p_t^\theta(x)(t - t') + \mathcal{O}(|t - t'|^2) .$$

Therefore, the first term in Equation (12) can be approximated as [9]

$$\mathbb{E}_{p_t}\left[\log \frac{p_t^\theta(X)}{p_t^\theta(X) + p_{t'}^\theta(X)}\right] = \mathbb{E}_{p_t}\left[\log \frac{1}{1 + \exp(\partial_t \log p_t^\theta(X)(t - t') + \mathcal{O}(|t - t'|^2))}\right] , \tag{39}$$

$$= \mathbb{E}_{p_t}\left[\log\left(\frac{1}{2} + \frac{1}{4}\partial_t \log p_t^\theta(X)(t - t')\right)\right] + \mathcal{O}(|t - t'|^2). \tag{40}$$

---

[9] This is induced by the first order approximation of the log sigmoid function, i.e. $\log \frac{1}{1+e^{-z}} = -\log 2 + \frac{z}{2} + \mathcal{O}(z^2)$.

Analogously, the second term in Equation (12) can be approximated as

$$\mathbb{E}_{p_{t'}}\left[\log\frac{p_{t'}^\theta(X)}{p_t^\theta(X) + p_{t'}^\theta(X)}\right] = \mathbb{E}_{p_{t'}}\left[\log\left(\frac{1}{2} - \frac{1}{4}\partial_t\log p_t^\theta(X)(t - t')\right)\right] + \mathcal{O}(|t - t'|^2). \tag{41}$$

DRE-$\infty$ (Choi et al., 2022) proposes to parameterize a time-score network $s_\theta$ and optimize Equation (12) with approximations given by Equations (40) and (41):

$$\mathcal{L}_{\text{DRE}-\infty}(\theta) = \mathbb{E}_t[\mathcal{L}_{\text{DRE}-\infty}(\theta; t, \delta)] \tag{42}$$

$$\text{with } \mathcal{L}_{\text{DRE}-\infty}(\theta; t, \delta) = -\frac{1}{2}\mathbb{E}_{p_t}[\log(1 - h_\theta(Y, t, \delta))] - \frac{1}{2}\mathbb{E}_{p_{t+\delta}}[\log h_\theta(Y, t, \delta)], \tag{43}$$

$$h_\theta(Y, t, \delta) = \frac{1}{2} + \frac{1}{4}s_\theta(x, t)\delta. \tag{44}$$

## B.2. Connection to MBAR

Computing free-energy differences, differences in log-normalizing constants between two potentials, is a central task in statistical physics and molecular dynamics (Lelièvre et al., 2010). In this section, we first introduce the free-energy estimation problem and the golden standard method MBAR (Shirts & Chodera, 2008) and build connection between with `DiffCLF`, which shows that their difference is the choice of model-parameterization.

**Free-energy estimation and FEP.**  Given a distribution

$$p(y) = \frac{1}{\mathcal{Z}}\exp(-\mathrm{U}(y)), \quad \text{with} \quad \mathcal{Z} = \int_\Omega \exp(-\mathrm{U}(y))\mathrm{d}y,$$

where $\Omega \subseteq \mathbb{R}^d$ is the support and $\mathrm{U} : \Omega \to \mathbb{R}$ is the energy function, the *free energy* is defined as the negative log-partition function, i.e.

$$\mathrm{F} = -\log\mathcal{Z} = \int_\Omega \exp(-U(y))\mathrm{d}y.$$

While direct estimation of F is difficult, one typically estimates the *free-energy difference* between two states $A$ and $B$ with supports $\Omega_A, \Omega_B$ and energy functions $\mathrm{U}_A, \mathrm{U}_B$ respectively,

$$\Delta\mathrm{F}_{AB} = \mathrm{F}_A - \mathrm{F}_B = \log\frac{Z_B}{Z_A}.$$

The easiest way to estimate $\Delta F_{AB}$ is through Free Energy Perturbation (Zwanzig, 1954, FEP) , which reformulates the free energy difference estimation task as an importance sampling problem:

$$\Delta F_{AB} = \log\mathbb{E}_{p_A}[\exp(U_A(x) - U_B(x))] = -\log\mathbb{E}_{p_B}[\exp(U_B(x) - U_A(x))]. \tag{45}$$

**MBAR.**  The golden-standard method for estimating this free-energy difference is *Multi-state Bennett Acceptance Ratio* (MBAR) (Shirts & Chodera, 2008), which generalizes *Bennett Acceptance Ratio* (BAR) (Bennett, 1976) to multi-states by introducing (i) a sequence of intermediate distributions with tractable energy functions bridging the two states $\{\mathrm{U}_i\}_{n=1}^N$ and (ii) associated samples with these distributions $\{y_n^{(k)}\}_{n=1,k=1}^{N,K_n}$ such that

$$\mathrm{U}_0 = \mathrm{U}_A, \quad \mathrm{U}_N = \mathrm{U}_B, \quad p_n(y) = \frac{\exp(-\mathrm{U}_n(y))}{\mathcal{Z}_n}, \quad y_n^k \sim p_n,$$

where $Z = \mathcal{Z}_n = \int_{\Omega_n}\exp(-\mathrm{U}_n(y))\mathrm{d}y$. Let $\mathrm{F}_n = -\log\mathcal{Z}_n$, by defining

$$p(c = n|y) = \frac{\exp(-\mathrm{U}_n(y) + \mathrm{F}_n)}{\sum_{m=1}^N \exp(-\mathrm{U}_m(y) + \mathrm{F}_m)} \quad \text{and} \quad p(y|n) = p_n(y).$$

MBAR treats the free energy estimation problem as a multiclass classification problem with maximum likelihood

$$
\begin{aligned}
\mathrm{F}^*_{1:N} &= \arg\max_{\mathrm{F}_{1:N}} \mathbb{E}_{p(n)}\mathbb{E}_{p_n}[\log p(c = n|y)] \\
&= \arg\max_{\mathrm{F}_{1:N}} \mathbb{E}_{p(n)}\mathbb{E}_{p_n}\left[\frac{\exp(-\mathrm{U}_n(y) + \mathrm{F}_n)}{\sum_{m=1}^{N}\exp(-\mathrm{U}_m(y) + \mathrm{F}_m)}\right] \\
&\approx \arg\max_{\mathrm{F}_{1:N}} \frac{1}{N}\sum_{n=1}^{N}\frac{1}{K_n}\sum_{k=1}^{K_n}\log\frac{\exp(-\mathrm{U}_n(y_n^{(k)}) + \mathrm{F}_n)}{\sum_{m=1}^{N}\exp(-\mathrm{U}_m(y_n^{(k)}) + \mathrm{F}_m)} \ ,
\end{aligned}
\tag{46}
$$

which solved using a fixed point iteration (see (Shirts & Chodera, 2008, Equation 3)) or using the Newton-Raphson algorithm (see (Shirts & Chodera, 2008, Equation 6)). Although the optimization problem is easy to solve, it requires equilibrium samples $\{y_n^{(k)}\}_k$ from each intermediate distribution $p_n$, where the intermediate energy functions are usually defined by tempering e.g.

$$
\mathrm{U}_n(y) = (1 - \beta_n)\mathrm{U}_A(y) + \beta_n\mathrm{U}_B(y) \quad , \text{with } \beta_1 = 0 \text{ and } \beta_N = 1 \ .
\tag{47}
$$

To get equilibrium samples from each intermediate distributions, one typically used annealed Markov Chain Monte Carlo samplers which are expensive.

**Connection between MBAR and `DiffCLF`.** By comparison with the `DiffCLF` objective (46) and the MBAR objective (10), one could observe that the difference between MBAR and `DiffCLF` is the *model-parameterization*.

- In MBAR, the energy functions $(\mathrm{U}_t)_t$ [10] are assumed known and the learnable objects are only the free energies $(\mathrm{F}_t)_t$, i.e.

$$
p_t^\theta(y) = \exp(-U_t(y) + \mathrm{F}_t^\theta) \ .
$$

- In `DiffCLF`, the EBMs are fully parameterized (see Equation (2)).

Besides, the accuracy of MBAR critically depends on the choice of the path, e.g. the anealling temperatures. Similarly to the BG case, learned energies could directly provides a data-driven approach to construct more effective paths, potentially surpassing hand-crafted designs.

## C. Blindness of Score Matching and Time Score Matching

In this section, we revisit the blindness of score matching, first analyzed in (Wenliang & Kanagawa, 2021; Zhang et al., 2022), and show that the problem persists even when matching higher-order derivatives or the time derivative of the trajectory $(\log p_t)_t$.

Let a set of time-dependent distributions with differentiable densities $\{g_t^1, \ldots, g_t^K\}$ having mutual disjoint (disconnected) support sets $\{\mathcal{X}_t^1, \ldots, \mathcal{X}_t^K\}$, where $\mathcal{X}_t^i \cap \mathcal{X}_t^j = \emptyset$ for any $i \neq j$ [11] For all $t \in [0, T]$, we define two mixture distributions $p_t = \sum_{k=1}^{K}\alpha_t^k g_t^k$ and $q_t = \sum_{k=1}^{K}\beta_t^k g_t^k$, where $\sum_{k=1}^{K}\alpha_t^k = 1$ and $\sum_{k=1}^{K}\beta_t^k = 1$.

**Score Matching is blind.** Score Matching is minimizing the *Fisher Divergence* (FD) between $p_t$ and $q_t$. Let $t \in [0, T]$, following the (Zhang et al., 2022, Proposition 1) we have that

$$
\begin{aligned}
\mathrm{FD}(p_t, q_t) &:= \mathbb{E}_{p_t}\left[\|\nabla\log p_t(X_t) - \nabla\log q_t(X_t)\|\,|^2\right] \ , \\
&= \sum_{k=1}^{K}\int_{\mathcal{X}_t^k}\left\|\frac{\nabla(\sum_{j=1}^{K}\alpha_t^j g_t^j(x))}{p_t(x)} - \frac{\nabla(\sum_{j=1}^{K}\beta_t^j g_t^j(x))}{q_t(x)}\right\|^2\alpha_t^k g_t^k(x)\mathrm{d}x \ , \\
&= \sum_{k=1}^{K}\int_{\mathcal{X}_t^k}\left\|\frac{\cancel{\alpha_t^k}\nabla g_t^k(x)}{\cancel{\alpha_t^k}g_t^k(x)} - \frac{\cancel{\beta_t^k}\nabla g_t^k(x)}{\cancel{\beta_t^k}g_t^k(x)}\right\|^2\alpha_t^k g_t^k(x)\mathrm{d}x = 0,
\end{aligned}
$$

---

[10]For clarity, we change the index of $n$ to a set of discretized time $t$ as in `DiffCLF`.

[11]Note that multi-modality could be modified depending on $t$ by simply setting two components and respective weights equal.

using $g_t^i(x_j) = 0$ and $\nabla g_t^i(x_j) = 0$ for $\forall x_j \in \mathcal{X}_j$ when $i \neq j$. Therefore, SM (or FD) is ill-defined on disconnected sets and has the blindness problem.

*Remark* C.1. The marginal $p_t$ needs not be supported on disjoint sets for all $t \in [0, T]$. For example, in DMs the terminal distribution $p_T$ is typically Gaussian and thus fully connected. In such cases, mixture proportions may be correctly estimated. However, at intermediate times where the support is disconnected, the proportions can be misestimated. Since the objective averages losses across time, the blindness issue may persist overall.

**Higher-order Score Matching is blind.** As suggested in (Lu et al., 2022), one could minimize the divergence between the Hessian or Laplacian of log-densities, i.e.

$$\mathbb{E}_{p_t}\left[\left\|\nabla^2 \log p_t(X_t) - \nabla^2 \log q_t(X_t)\right\|_F^2\right] \quad \text{or} \quad \mathbb{E}_{p_t}\left[(\Delta \log p_t(X_t) - \Delta \log q_t(X_t))^2\right] ,$$

where $\|\cdot\|_F$ is the Frobenius norm. Similarly as the previous paragraph, for all $t \in [0, T]$ and $x \in \mathcal{X}_t^k$,

$$\nabla^2 \log p_t(x) = \frac{p_t(x) \sum_{j=1}^K \alpha_t^j \nabla^2 g_t^j(x) - \left(\sum_{j=1}^K \alpha_t^j \nabla g_t^j(x)\right) \nabla p_t(x)^\top}{p_t(x)^2} ,$$

$$= \frac{\cancel{\alpha_t^k} g_t^k(x) \cancel{\alpha_t^k} \nabla^2 g_t^k(x) - \cancel{\alpha_t^k} \nabla g_t^k(x) \cancel{\alpha_t^k} \nabla g_t^k(x)^\top}{(\cancel{\alpha_t^k} g_t^k(x))^2} ,$$

which doesn't depend on the weights $\alpha$. The cases for Laplacian or other higher-order (w.r.t. $x$) regression are analogous.

**Time Score Matching can be blind.** The Time Score Matching (see Appendix A.3 for details) objective can be written for any $t \in [0, T]$ as

$$\mathbb{E}_{p_t}\left[(\partial_t \log p_t(X_t) - \partial_t \log q_t(X_t))^2\right] =$$

$$\sum_{k=1}^K \int_{\mathcal{X}_t^k} \left(\frac{\partial_t(\sum_{j=1}^K \alpha_t^j g_t^j(x))}{p_t(x)} - \frac{\partial_t(\sum_{j=1}^K \beta_t^j g_t^j(x))}{q_t(x)}\right)^2 \alpha_t^k g_t^k(x)\mathrm{d}x , \quad (48)$$

where $\partial_t(\sum_{j=1}^K \alpha_t^j g_t^j(x))$ can be expanded by leveraging $g_t^i(x_j) = 0$ for any $x_j \in \mathcal{X}_j$ and $i \neq j$

$$\partial_t\left(\sum_{j=1}^K \alpha_t^j g_t^j(x)\right) = (\partial_t \alpha_t^k)g_t^k(x) + \sum_{j=1}^K \alpha_t^j \partial_t g_t^j(x), \quad \forall x \in \mathcal{X}_t^k , \quad (49)$$

hence,

$$\mathbb{E}_{p_t}\left[(\partial_t \log p_t(X_t) - \partial_t \log q_t(X_t))^2\right]$$

$$= \sum_{k=1}^K \int_{\mathcal{X}_t^k} \left(\frac{(\partial_t \alpha_t^k)g_t^k(x) + \sum_{j=1}^K \alpha_t^j \partial_t g_t^j(x)}{\alpha_t^k g_t^k(x)} - \frac{(\partial_t \beta_t^k)g_t^k(x) + \sum_{j=1}^K \beta_t^j \partial_t g_t^j(x)}{\beta_t^k g_t^k(x)}\right)^2 \alpha_t^k g_t^k(x)\mathrm{d}x ,$$

$$= \sum_{k=1}^K \int_{\mathcal{X}_t^k} \left(\frac{\partial_t \alpha_t^k}{\alpha_t^k} - \frac{\partial_t \beta_t^k}{\beta_t^k} + \frac{\sum_{j\neq k} \alpha_t^j \partial_t g_t^j(x)}{\alpha_t^k g_t^k(x)} - \frac{\sum_{j\neq k} \beta_t^j \partial_t g_t^j(x)}{\beta_t^k g_t^k(x)}\right)^2 \alpha_t^k g_t^k(x)\mathrm{d}x \geq 0 ,$$

which can be 0 in some cases. For example, if the mixture weights $\alpha, \beta$ are time-independent (as in DMs) and the supports of $g_t$ vary only slowly over time (also typical in DMs), then the loss becomes mode-blind.

**The Fokker-Planck regularization is mode blind.** When $p_t, q_t$ are generated from the same Itô SDE, say $dX_t = f(t, X_t)\mathrm{d}t + g(t, X_t)\mathrm{d}W_t$, starting from $p_0, q_0$ respectively, the Fokker-Planck Equation (see Appendix A.1) tells us

$$\partial_t \log p_t(x) = -\nabla \cdot f(t, x) - f(t, x)^\top \nabla \log p_t(x) + \frac{1}{2}g(t, x)^\top g(t, x)\left(\Delta \log p_t(x) + \|\nabla \log p_t\|^2\right) ,$$

$$\partial_t \log q_t(x) = -\nabla \cdot f(t, x) - f(t, x)^\top \nabla \log q_t(x) + \frac{1}{2}g(t, x)^\top g(t, x)\left(\Delta \log q_t(x) + \|\nabla \log q_t\|^2\right) .$$

Therefore, the FPE regularization is equivalent to regressing a combination of (i) the score and its squared norm, and (ii) the Laplacian, which are all mode-blind. Therefore, FPE regularization in such cases is blind in this case.

**Alleviating the blindness of score matching.** Several approaches have been proposed to address mode blindness. (Zhang et al., 2022) introduce an auxiliary noise distribution $m$ and minimize the Fisher divergence between mixtures of $(\pi, m)$ and $(p^\theta, m)$, which indirectly enforces proximity between $\pi$ and $p^\theta$. Similarly, (Schröder et al., 2023) propose the energy discrepancy, a contrastive objective between $\pi$ and its noisy counterpart that provably avoids blindness (see (Schröder et al., 2023, Figure 1)). However, both methods hinge on the careful design and tuning of the auxiliary noise distribution or kernel, limiting their scalability in practice.

# D. Proofs of propositions

## D.1. Proof of Proposition 3.1

**Proposition 3.1.** *Let $N \in \mathbb{N}^*$ and $t_{1:N} \in [0, T]^N$. The Diffusive Classification loss is*

$$\mathcal{L}_{\mathrm{clf}}(\theta; N) = \mathbb{E}_{t_{1:N}}[\mathcal{L}_{\mathrm{clf}}(\theta; t_{1:N})], \quad \mathcal{L}_{\mathrm{clf}}(\theta; t_{1:N}) = -\frac{1}{N} \sum_{i=1}^{N} \mathbb{E}_{p_{t_i}} \left[ \log \frac{p_{t_i}^\theta(Y_i)}{\sum_{j=1}^{N} p_{t_j}^\theta(Y_i)} \right],$$

*where $p_t^\theta(y_t) = \exp(-\mathrm{U}_t^\theta(y_t) + \mathrm{F}_t^\theta)$. A minimizer of $\mathcal{L}_{\mathrm{clf}}(\theta)$ is attained by $p_t^{\theta^*} = p_t$ for all $t \in [0, T]$.*

*Proof.* We first show that, given a batch of times $t_{1:N} \in [0, T]^N$, $p_{t_i}^{\theta*} = p_{t_i}$ for any $i \in [\![1, N]\!]$ obtains the optimality. For clarity, for all $i \in [\![1, N]\!]$, set $p_i = p_{t_i}$ and $p_i^\theta = p_{t_i}^\theta$. Let $\Omega_i = \mathrm{supp}(p_i)$ and $\Omega = \bigcup_{i=1}^N \Omega_i$. Extend each $p_i$ by 0 on $\Omega \setminus \Omega_i$ so that all integrals are over $\Omega$. Then

$$\sum_{i=1}^{N} \mathbb{E}_{p_i} \left[ \log \frac{p_i^\theta(Y_{t_i})}{\sum_{j=1}^{N} p_j^\theta(Y_{t_i})} \right] = \int_\Omega \sum_{i=1}^{N} p_i(y) \log \frac{p_i^\theta(y)}{\sum_{j=1}^{N} p_j^\theta(y)} \, \mathrm{d}y \ . \tag{50}$$

For $y \in \Omega$, define

$$s(y) := \sum_{k=1}^{N} p_k(y), \qquad w_i(y) := \frac{p_i(y)}{s(y)}, \qquad w_i^\theta(y) := \frac{p_i^\theta(y)}{\sum_{k=1}^{N} p_k^\theta(y)} \ ,$$

interpreting the ratios arbitrarily if $s(y) = 0$ (those $y$ do not affect the integral). Then the integrand rewrites as

$$\sum_{i=1}^{N} p_i(y) \log w_i^\theta(y) = s(y) \sum_{i=1}^{N} w_i(y) \log w_i^\theta(y) \ .$$

Since $\sum_i w_i(y) = \sum_i w_i^\theta(y) = 1$, Gibbs' inequality gives, for all $y$ with $s(y) > 0$,

$$\sum_{i=1}^{N} w_i(y) \log w_i^\theta(y) \le \sum_{i=1}^{N} w_i(y) \log w_i(y) \ ,$$

with equality iff $w_i^\theta(y) = w_i(y)$ for all $i$. Multiplying by $s(y)$ and integrating over $\Omega$ yields

$$\sum_{i=1}^{N} \mathbb{E}_{p_i} \left[ \log \frac{p_i^\theta(Y_i)}{\sum_{j=1}^{N} p_j^\theta(Y_i)} \right] \le \int_\Omega \sum_{i=1}^{N} p_i(y) \log \frac{p_i(y)}{\sum_{j=1}^{N} p_j(y)} \, \mathrm{d}y \ ,$$

where equality holds whenever

$$\frac{p_i^\theta(y)}{\sum_{j=1}^{N} p_j^\theta(y)} = \frac{p_i(y)}{\sum_{j=1}^{N} p_j(y)} \quad \text{for } \bar{p}\text{-a.e. } y \in \Omega \ ,$$

with $\bar{p} = \frac{1}{N} \sum_i p_i$. In particular, choosing $p_i^\theta = p_i$ for all $i$ satisfies the condition and attains the minimum. Therefore, choosing $p_t^\theta = p_t$ for all $t \in [0, T]$ attains the minimum of $\mathcal{L}_{\mathrm{clf}}(\theta)$. $\qquad \square$

### D.2. Proof of Proposition 3.2

It is noticeable that optimizing the `DiffCLF` objective solely can achieve different minimums. For example, one could easily verify that $p_t^\theta(x) = c(x)p_t(x)$ for a non-negative function $c$ is also a minimum. In the following proposition, we will show that jointly optimizing with the *Denoising Score Matching* (DSM) objective (7) can fix this non-uniqueness issue.

To prove, we will first present the properties of minimizers for $\mathcal{L}_{\text{clf}}$ and $\mathcal{L}_{\text{DSM}}$ respectively, and then combine them to finish the proof.

**Proposition 3.2** (Formal). *Let $t \in [0, T]$, if conditions (A1) and (A2) are fulfilled:*

*(A1) For any $t$, $p_t$ and $p_t^\theta$ are absolutely continuous w.r.t. the Lebesgue measure on $\Omega_t := \text{supp}(p_t)$, with $\nabla \log p_t, \nabla \log p_t^\theta \in L^2(p_t)$,*

*(A2) The union of supports $\Omega := \bigcup_{t \in [0,T]} \Omega_t$ is (path-)connected,*

*then $p_t^{\theta^\star} = p_t$ for all $t \in [0, T]$ is the unique minimizer of the joint objective $\mathcal{L}_{\text{DSM}} + \mathcal{L}_{\text{clf}}$ (Equations (7) and (10)).*

*Proof.* We first show that, given any batch of times $t_{1:N}$, by optimizing $\sum_{i=1}^{N} \mathcal{L}_{\text{DSM}}(\theta; t_i) + \mathcal{L}_{\text{clf}}(\theta; t_{1:N})$, then $p_{t_i}^\theta = p_{t_i}$ for all $i \in [\![1, N]\!]$ is the unique minimizer of the optimization problem. For clarity, let $p_i = p_{t_i}$ and $p_i^\theta = p_{t_i}^\theta$.

**Step 1 (structure of $\mathcal{L}_{\text{clf}}$ minimizers).** By Proposition 3.1, any minimizer of $\mathcal{L}_{\text{clf}}(\theta; t_{1:N})$ satisfies, for $\bar{p}$-a.e. $y \in \Omega$ and any $i \in [\![1, N]\!]$,

$$\frac{p_i^\theta(y)}{\sum_{j=1}^{N} p_j^\theta(y)} = \frac{p_i(y)}{\sum_{j=1}^{N} p_j(y)} \, ,$$

where $\bar{p} := \frac{1}{N} \sum_{i=1}^{N} p_i$. Equivalently, there exists a positive measurable function $c : \Omega \to (0, \infty)$, independent of $i$ (i.e. time-independent), such that

$$p_i^\theta(y) = c(y) \, p_i(y) \quad \text{for all } i, \ \bar{p}\text{-a.e. } y \in \Omega \, , \tag{51}$$

together with the per-class normalization constraints

$$\int_\Omega c(y) \, p_i(y) \, \mathrm{d}y = 1 \quad \text{for each } i = 1, \ldots, N \, . \tag{52}$$

**Step 2 (structure of $\mathcal{L}_{\text{DSM}}$ minimizers).** By standard DSM identifiability (under (A1)), any minimizer of $\sum_{i=1}^{N} \mathcal{L}_{\text{DSM}}(\theta; t_i)$ satisfies

$$\nabla \log p_i^\theta(y) = \nabla \log p_i(y) \quad \text{for } p_i\text{-a.e. } y \in \Omega_i, \ \forall i \in [\![1, N]\!] \, . \tag{53}$$

Using Equation (51) in Equation (53) gives, for each $i$,

$$\nabla \log \big(c(y) \, p_i(y)\big) = \nabla \log p_i(y) \quad \Rightarrow \quad \nabla \log c(y) = 0 \quad \text{for } p_i\text{-a.e. } y \in \Omega_i \, .$$

Therefore, $c(y) \equiv C$ is constant $\bar{p}$-a.e. on $\Omega$. Recalling the normalization constraint where $\int c(y)p_i(y)dy = 1$, we have $C = 1$ and hence $p_i^\theta = p_i$ a.e. for all $i$. Since the normalization function $c(y) \equiv 1$ is unique, $p_i^\theta = p_i$ is the unique minimizer. Hence, the joint objective $\mathcal{L}_{\text{DSM}}(\theta) + \mathcal{L}_{\text{clf}}(\theta)$ has an unique minimum, that is $p_t^\theta = p_t$ for all $t \in [0, T]$. $\square$

### D.3. Proof of Proposition 3.3

In this part, we deliver the properties of estimator for the joint objective $\mathcal{L}_{\text{DSM}} + \mathcal{L}_{\text{clf}}$ (Equations (7) and (10)). We first consider the joint loss given a fixed set of times, $t_{1:N} \in [0, T]^N$. Then further extend it with consideration of the randomness of times to finish the proof.

D.3.1. LEMMATA

For clarity, let $q_{t_i} = q_i$, $p_{t_i} = p_i$, $p_{t_i}^\theta = p_i^\theta$, and $\mathrm{supp}(p_i) = \Omega_i$ for all $i \in [\![1, N]\!]$, and define

$$\mathcal{L}_{t_{1:N}}^M(\theta) = \frac{1}{M} \sum_{m=1}^M l_{t_{1:N}}(\theta; m), \quad l_{t_{1:N}}(\theta; m) = -\frac{1}{N} \sum_{i=1}^N \log \frac{p_i^\theta(Y_i^{(m)})}{\sum_{j=1}^N p_j^\theta(Y_i^{(m)})} ,$$

$$\mathcal{J}_{t_{1:N}}^M(\theta) = \frac{1}{NM} \sum_{m=1}^M \sum_{i=1}^N j_{t_i}(\theta; m), \quad j_{t_i}(\theta; m) = \left\| \nabla \log p_i^\theta(\tilde{Y}_i^{(m)}) - \nabla \log p_i(\tilde{Y}_i^{(m)} | \tilde{X}_i^{(m)}) \right\|^2 ,$$

where

- $\mathcal{L}_{t_{1:N}}^M$ is the empirical `DiffCLF` loss;

- $\mathcal{J}_{t_{1:N}}^M$ is the empirical DSM loss;

- $Y_i^{(m)} \overset{i.i.d.}{\sim} p_i$ for all $i \in [\![1, N]\!]$;

- $\tilde{X}_i^{(m)} \overset{i.i.d.}{\sim} q_i$ for all $i \in [\![1, N]\!]$;

- $\tilde{Y}_i^{(m)} \sim p_i(\cdot | \tilde{X}_i^{(m)})$ for all $i \in [\![1, N]\!]$;

- $Y_i^{(m)}, Y_i^{(m)}, \tilde{Y}_i^{(m)}$ are independent for all $i \in [\![1, N]\!]$.

Let $\Omega = \cup_{i=1}^N \Omega_i$ as the extended support, where $p_i(y) = 0$ for any $x \in \Omega / \Omega_i$ for all $i$.

Recall from Proposition 3.1, when jointly optimizing the DSM and `DiffCLF` losses, the optimal parameter $\theta^\star$ is unique and recovers the true marginal densities $(p_t)_t$ (or $(p_i)_i$ in the fixed time-batch setting). Therefore in the following, we have $p_t^{\theta^\star} = p_t$.

Let $\hat{\theta}_{t_{1:N}}^M$ optimizing the joint empirical loss $\mathcal{L}_{t_{1:N}}^M + \mathcal{J}_{t_{1:N}}^M$ and $\theta_{t_{1:N}}^\star$ optimizing the joint population loss $\mathcal{L}_{t_{1:N}} + \mathcal{J}_{t_{1:N}}$. For simplicity, we use $\theta^\star$ and $\theta_{t_{1:N}}^*$ interchangeably whenever no ambiguity arises. By leveraging the uniqueness of the joint optimum, we proceed in three steps:

(1) We generalize Lemmas 12–14 of (Gutmann & Hyvärinen, 2010) from the binary case to the multi-class setting with a fully parameterized model, obtaining Lemmas D.4 to D.6.

(2) We impose standard regularity conditions on the DSM objective. In particular, Assumptions D.1 and D.2 are classical assumptions in $L_2$-risk minimization and $M$-estimation, ensuring convergence in probability of the empirical Hessian and vanishing population gradient at $\theta^\star$. We additionally posit Assumption D.3, which provides a finite-sample covariance structure for the DSM gradient. We do not attempt to verify Assumption D.3 for specific architectures, as our main focus is the diffusive classification objective.

(3) Finally, combining Assumptions D.1 to D.3 with Lemmas D.4 to D.6, we show that $\hat{\theta}_{t_{1:N}}$ is a consistent estimator of $\theta^\star$, and we obtain its asymptotic covariance matrix together with an $\mathcal{O}(1/M)$ error bound around $\theta^\star$, in Lemma D.7.

**Assumption D.1.**

(i) $\nabla_\theta^2 \left[ M^{-1} \sum_m j_t(\theta; m) \Big|_{\theta=\theta^\star} \right]$ converges in probability to a positive-definite matrix $\mathcal{P}_t$ as the sample size $M$ tends to infinity.

(ii) $\nabla_\theta^2 \mathcal{J}_{t_{1:N}}^M$ converges in probability to $\mathcal{P}_{t_{1:N}} := N^{-1} \sum_{i=1}^N \mathcal{P}_{t_i}$ as the sample size $M$ tends to infinity. By (i), $\mathcal{P}_{t_{1:N}}$ is a positive-definite matrix.

**Assumption D.2.** $\mathbb{E}\left[ \nabla_\theta \left( M^{-1} \sum_m j_t(\theta; m) \right) \Big|_{\theta=\theta^\star} \right] = 0$ and $\mathbb{E}\left[ \nabla_\theta \mathcal{J}_{t_{1:N}}^M(\theta) \Big|_{\theta=\theta^\star} \right] = 0$.

**Assumption D.3.**

$$\text{Cov}\left[\nabla_\theta\left(M^{-1}\sum_m j_t(\theta;m)\right)\Big|_{\theta=\theta^\star}\right] = \mathcal{C}_t/M \ ,$$

and

$$\text{Cov}\left[\nabla_\theta\mathcal{J}^M_{t_{1:N}}(\theta)\Big|_{\theta=\theta^\star}\right] = \frac{1}{MN}\sum_{i=1}^N \mathcal{C}_{t_i} \ .$$

**Lemma D.4.** $\nabla^2_\theta\mathcal{L}^M_{t_{1:N}}(\theta)\Big|_{\theta=\theta^\star}$ *converges in probability to* $\mathcal{I}_{t_{1:N}}$ *as the sample size* $M$ *tends to infinity, where*

$$\mathcal{I}_{t_{1:N}} = \frac{1}{N}\sum_{i=1}^N \mathbb{E}_{p_i}\left[D_i(Y_i;t_{1:N})D_i(Y_i;t_{1:N})^\top\right] \ ,$$

*where*

$$D_i(y_i;t_{1:N}) = \nabla_\theta\log p^\theta_i(y_i)\Big|_{\theta=\theta^\star} - \bar{g}^{\theta^\star}_{t_{1:N}}(y_i) \quad and \quad \bar{g}^\theta_{t_{1:N}}(y) = \frac{\sum_{j=1}^N p^\theta_j(y)\nabla_\theta\log p^\theta_j(y)}{\sum_{j=1}^N p^\theta_j(y)} \ .$$

*Proof.*

$$\begin{aligned}
\nabla^2_\theta\mathcal{L}^M_{t_{1:N}}(\theta)\Big|_{\theta=\theta^\star} &= -\frac{1}{M}\sum_{m=1}^M\frac{1}{N}\sum_{i=1}^N\left[\nabla^2_\theta\log\frac{p^\theta_i(y^{(m)}_i)}{\sum_{j=1}^N p^\theta_j(y^{(m)}_i)}\Big|_{\theta=\theta^\star}\right] \\
&= -\frac{1}{N}\sum_{i=1}^N\frac{1}{M}\sum_{m=1}^M\left[\nabla_\theta\left(\frac{\nabla_\theta p^\theta_i(y^{(m)}_i)}{p^\theta_i(y^{(m)}_i)} - \frac{\sum_j\nabla_\theta p^\theta_j(y^{(m)}_i)}{\sum_j p^\theta_j(y^{(m)}_i)}\right)\Big|_{\theta=\theta^\star}\right] \\
&= -\frac{1}{N}\sum_{i=1}^N\frac{1}{M}\sum_{m=1}^M\left[R(y^{(m)}_i) - K(y^{(m)}_i)\right] \ ,
\end{aligned}$$
(54)

where

$$R(y) = \frac{p^\theta_i(y)\sum_j p^\theta_j(y)\nabla_\theta(\sum_j p^\theta_j(y)\nabla_\theta p^\theta_i(y) - p^\theta_i(y)\sum_j\nabla_\theta p^\theta_j(y))}{(p^\theta_i(y)\sum_j p^\theta_j(y))^2}\Big|_{\theta=\theta^\star}$$

$$K(y) = \frac{(\nabla_\theta(p^\theta_i(y)\sum_j p^\theta_j(y)))(\sum_j p^\theta_j(y)\nabla_\theta p^\theta_i(y) - p^\theta_i(y)\sum_j\nabla_\theta p^\theta_j(y))^\top}{(p^\theta_i(y)\sum_j p^\theta_j(y))^2}\Big|_{\theta=\theta^\star}$$

Firstly, as $M\to\infty$, the summation converges to the expectation in probability, i.e.

$$\lim_{M\to\infty}\frac{1}{M}\sum_{m=1}^M R(y^{(m)}_i)\xrightarrow{P}\mathbb{E}_{p_i}[R(Y_i)] \quad and \quad \lim_{M\to\infty}\frac{1}{M}\sum_{m=1}^M K(y^{(m)}_i)\xrightarrow{P}\mathbb{E}_{p_i}[K(Y_i)] \ ,$$

and we have

$$
\begin{aligned}
\sum_{i=1}^{N} \mathbb{E}_{p_i}\left[R(Y_i)\right] &= \sum_{i=1}^{N} \int_{\Omega_i} p_i(y_i) \frac{\nabla_\theta\left(\sum_j p_j^\theta(y_i)\nabla_\theta p_i^\theta(y_i) - p_i^\theta(y_i)\sum_j \nabla_\theta p_j^\theta(y_i)\right)\big|_{\theta=\theta^\star}}{p_i^{\theta^\star}(y_i)\sum_j p_j^{\theta^\star}(y_i)} \mathrm{d}y_i \\
&\stackrel{(i)}{=} \sum_{i=1}^{N} \int_{\Omega_i} \cancel{p_i(y_i)} \frac{\nabla_\theta\left(\sum_j p_j^\theta(y_i)\nabla_\theta p_i^\theta(y_i) - p_i^\theta(y_i)\sum_j \nabla_\theta p_j^\theta(y_i)\right)\big|_{\theta=\theta^\star}}{\cancel{p_i(y_i)}\sum_j p_j(y_i)} \mathrm{d}y_i \\
&= \int_{\Omega} \sum_{i=1}^{N} \frac{\nabla_\theta\left(\sum_j p_j^\theta(y_i)\nabla_\theta p_i^\theta(y_i) - p_i^\theta(y_i)\sum_j \nabla_\theta p_j^\theta(y_i)\right)\big|_{\theta=\theta^\star}}{\sum_j p_j(y_i)} \mathrm{d}y_i \\
&\stackrel{(ii)}{=} \nabla_\theta \int_{\Omega} \sum_{i=1}^{N} \frac{\sum_j p_j^\theta(y_i)\nabla_\theta p_i^\theta(y_i) - p_i^\theta(y_i)\sum_j \nabla_\theta p_j^\theta(y_i)}{\sum_j p_j(y_i)} \mathrm{d}y_i \Big|_{\theta=\theta^\star} \\
&= \nabla_\theta \int_{\Omega} \frac{\cancel{\sum_i\sum_j p_j^\theta(y_i)\nabla_\theta p_i^\theta(y_i)} - \cancel{\sum_i p_i^\theta(y_i)\sum_j \nabla_\theta p_j^\theta(y_i)}}{\sum_j p_j(y_i)} \mathrm{d}y_i \Big|_{\theta=\theta^\star} \\
&= 0\,,
\end{aligned}
$$

where $(i)$ bases on $p_i^{\theta^\star} = p_i$ and $(ii)$ swaps the order of differentiation and integration with mild conditions. Therefore, as $M \to \infty$, Equation (54) converges as follows in probability:

$$
\lim_{M\to\infty} \nabla_\theta^2 \mathcal{L}_{t_{1:N}}^M(\theta)\Big|_{\theta=\theta^\star} \xrightarrow{P} -\frac{1}{N}\sum_{i=1}^{N} \mathbb{E}_{p_i}\left[-K(Y_i)\right] = \frac{1}{N}\sum_{i=1}^{N} \mathbb{E}_{p_i}\left[K(Y_i)\right]\,.
$$

Notice that

$$
\nabla_\theta\left(p_i^\theta(y)\sum_j p_j^\theta(y)\right) = \underbrace{\sum_j p_j^\theta(y)\nabla_\theta p_i^\theta(y)}_{:=a_i^\theta(y)} + \underbrace{p_i^\theta(y)\sum_j \nabla_\theta p_j^\theta(y)}_{b_i^\theta(y)},
$$

we could simplify

$$
\begin{aligned}
\frac{1}{N}\sum_{i=1}^{N} \mathbb{E}_{p_i}\left[K(Y_i)\right] &= \frac{1}{N}\sum_{i=1}^{N} \int_{\Omega_i} p_i(y_i) \frac{(a_i^{\theta^\star}(y_i)+b_i^{\theta^\star}(y_i))(a_i^{\theta^\star}(y_i)-b_i^{\theta^\star}(y_i))^\top}{(p_i^\theta(y_i)\sum_j p_j^{\theta^\star}(y_i))^2} \mathrm{d}y_i \\
&\stackrel{(i)}{=} \frac{1}{N} \int_{\Omega} \sum_{i=1}^{N} p_i(y_i) \frac{a_i^{\theta^\star}(y_i)a_i^{\theta^\star}(y_i)^\top - b_i^{\theta^\star}(y_i)b_i^{\theta^\star}(y_i)^\top}{p_i(y_i)^2 (\sum_j p_j(y_i))^2} \mathrm{d}y_i\,,
\end{aligned}
\tag{55}
$$

where $(i)$ is again by $p_i^{\theta^\star} = p_i$ and

$$
a_i^{\theta^\star}(y)a_i^{\theta^\star}(y)^\top = \left(\sum_j p_j(y)\nabla_\theta p_i^\theta(y)\Big|_{\theta=\theta^\star}\right)\left(\sum_j p_j(y)\nabla_\theta p_i^\theta(y)\Big|_{\theta=\theta^\star}\right)^\top
\tag{56}
$$

$$
= p_i(y)^2 \sum_k \sum_j p_k(y)p_j(y)\nabla_\theta \log p_i^\theta(y)\Big|_{\theta=\theta^\star}\left[\nabla_\theta \log p_i^\theta(y)\Big|_{\theta=\theta^\star}\right]^\top,
\tag{57}
$$

$$
b_i^{\theta^\star}(y)b_i^{\theta^\star}(y)^\top = \left(p_i(y)\sum_j \nabla_\theta p_j^\theta(y)\Big|_{\theta=\theta^\star}\right)\left(p_i(y)\sum_j \nabla_\theta p_j^\theta(y)\Big|_{\theta=\theta^\star}\right)^\top
$$

$$
= p_i(y)^2 \sum_k \sum_j p_k(y)p_j(y)\nabla_\theta \log p_k^\theta(y)\Big|_{\theta=\theta^\star}\left[\nabla_\theta \log p_j^\theta(y)\Big|_{\theta=\theta^\star}\right]^\top.
\tag{58}
$$

By plugging Equations (57) and (58) into Equation (55), we have

$$\frac{1}{N}\sum_{i=1}^{N}\mathbb{E}_{p_i}\left[K(Y_i)\right] = \frac{1}{N}\int_{\Omega}\sum_{i=1}^{N}p_i(y_i)\frac{\sum_k\sum_j p_k(y_i)p_j(y_i)\nabla_\theta\log p_i^\theta(y_i)\big|_{\theta=\theta^\star}\left[\nabla_\theta\log p_i^\theta(y_i)\big|_{\theta=\theta^\star}\right]^\top}{(\sum_j p_j(y_i))^2}$$

$$-\sum_{i=1}^{N}p_i(y_i)\frac{\sum_k\sum_j p_k(y_i)p_j(y_i)\nabla_\theta\log p_k^\theta(y_i)\big|_{\theta=\theta^\star}\left[\nabla_\theta\log p_j^{\theta^\star}(y_i)\big|_{\theta=\theta^\star}\right]^\top}{(\sum_j p_j(y_i))^2}\mathrm{d}y_i$$

$$= \frac{1}{N}\int_{\Omega}\sum_{i=1}^{N}p_i(y_i)\underbrace{\left(\nabla_\theta\log p_i^\theta(y_i)\big|_{\theta=\theta^\star}-\bar{g}_{t_{1:N}}^{\theta^\star}(y_i)\right)}_{:=D_i(y;t_{1:N})}\left(\nabla_\theta\log p_i^\theta(y_i)\big|_{\theta=\theta^\star}-\bar{g}_{t_{1:N}}^{\theta^\star}(y_i)\right)^\top\mathrm{d}y_i$$

$$= \frac{1}{N}\sum_{i=1}^{N}\mathbb{E}_{p_i}\left[D_i(Y_i;t_{1:N})D_i(Y_i;t_{1:N})^\top\right] := \mathcal{I}_{t_{1:N}} , \tag{59}$$

where

$$\bar{g}_{t_{1:N}}^{\theta^\star}(y) = \frac{\sum_{j=1}^{N}p_j^\theta(y)\nabla_\theta\log p_j^\theta(y)}{\sum_{j=1}^{N}p_j^\theta(y)} \quad\text{and}\quad D_i(y;t_{1:N}) = \nabla_\theta\log p_i^\theta(y)\big|_{\theta=\theta^\star}-\bar{g}_{t_{1:N}}^{\theta^\star}(y) .$$

Hence,

$$\lim_{M\to\infty}\nabla_\theta^2\mathcal{L}_{t_{1:N}}^M(\theta)\big|_{\theta=\theta^\star} \xrightarrow{P} \mathcal{I}_{t_{1:N}} .$$

$\square$

**Lemma D.5.** *We have* $\mathbb{E}\left[\nabla_\theta\mathcal{L}_{t_{1:N}}^M(\theta)\big|_{\theta=\theta^\star}\right] = 0.$

*Proof.* First notice that

$$\mathbb{E}\left[\nabla_\theta\mathcal{L}_{t_{1:N}}^M(\theta)\big|_{\theta=\theta^\star}\right] = \frac{1}{M}\sum_{m=1}^{M}\mathbb{E}\left[\nabla_\theta l_{t_{1:N}}(\theta;m)\big|_{\theta=\theta^\star}\right] \overset{i.i.d.}{=} \mathbb{E}\left[\nabla_\theta l_{t_{1:N}}(\theta;m)\big|_{\theta=\theta^\star}\right] ,$$

where the expectation is over $\otimes_{i=1}^{N}p_i$. We then have

$$\mathbb{E}\left[\nabla_\theta l_{t_{1:N}}(\theta;m)\big|_{\theta=\theta^\star}\right] = -\frac{1}{N}\sum_{i=1}^{N}\mathbb{E}_{p_i}\left[\nabla_\theta\log\frac{p_i^\theta(Y_i)}{\sum_{j=1}^{N}p_j^\theta(Y_i)}\Big|_{\theta=\theta^\star}\right]$$

$$= -\frac{1}{N}\sum_{i=1}^{N}\mathbb{E}_{p_i}\left[\frac{\nabla_\theta p_i^\theta(Y_i)\big|_{\theta=\theta^\star}}{p_i^{\theta^\star}(Y_i)} - \frac{\sum_j\nabla_\theta p_j^\theta(Y_i)\big|_{\theta=\theta^\star}}{\sum_j p_j^{\theta^\star}(Y_i)}\right]$$

$$= -\frac{1}{N}\sum_{i=1}^{N}\left[\int_{\Omega_i}p_i(y_i)\frac{\sum_j p_j^{\theta^\star}(y_i)\nabla_\theta p_i^\theta(y_i)\big|_{\theta=\theta^\star} - p_i^{\theta^\star}(y_i)\sum_j\nabla_\theta p_j^\theta(y_i)\big|_{\theta=\theta^\star}}{p_i^{\theta^\star}(y_i)\sum_j p_j^{\theta^\star}(y_i)}\mathrm{d}y_i\right] .$$

By using the extended support $\Omega$ and the optimality $p_i^{\theta^\star} = p_i$, we have

$$\mathbb{E}\left[\nabla_\theta l_{t_{1:N}}(\theta;m)\big|_{\theta=\theta^\star}\right] = -\frac{1}{N}\int_{\Omega}\sum_{i=1}^{N}p_i(y_i)\frac{\sum_j p_j(y_i)\nabla_\theta p_i^\theta(y_i)\big|_{\theta=\theta^\star} - p_i(y_i)\sum_j\nabla_\theta p_j^\theta(y_i)\big|_{\theta=\theta^\star}}{p_i(y_i)\sum_j p_j(y_i)}\mathrm{d}y_i$$

$$= -\frac{1}{N}\int_{\Omega}\frac{\sum_i\sum_j p_j(y_i)\nabla_\theta p_i^\theta(y_i)\big|_{\theta=\theta^\star} - \sum_i p_i(y_i)\sum_j\nabla_\theta p_j^\theta(y_i)\big|_{\theta=\theta^\star}}{\sum_j p_j(y_i)}\mathrm{d}y_i$$

$$= 0 .$$

$\square$

**Lemma D.6.** *The covariance* $\mathrm{Cov}\left[\nabla_\theta \mathcal{L}_{t_{1:N}}^M (\theta^\star)\Big|_{\theta=\theta^\star}\right]$ *is*

$$\frac{1}{M}\left(\frac{1}{N}\mathcal{I}_{t_{1:N}} - \frac{1}{N^2}\sum_i \mathbb{E}_{p_i}\left[D_i(Y_i)\right] \mathbb{E}_{p_i}\left[D_i(Y_i)\right]^\top\right) ,$$

*where* $D_i(y_i) = \nabla_\theta \log p_i^\theta(y_i)\Big|_{\theta=\theta^\star} - \bar{g}_{\theta^\star}(y_i)$, $\bar{g}_{\theta^\star}(y_i)$ *is defined in Lemma D.4, and the expectations are taken over* $p_i$.

*Proof.* As $\mathbb{E}\left[\nabla_\theta \mathcal{L}_{t_{1:N}}^M (\theta)\Big|_{\theta=\theta^\star}\right] = 0$, we have

$$\mathrm{Cov}\left[\nabla_\theta \mathcal{L}_{t_{1:N}}^M (\theta)\Big|_{\theta=\theta^\star}\right] = \mathbb{E}\left[\nabla_\theta \mathcal{L}_{t_{1:N}}^M (\theta)\Big|_{\theta=\theta^\star} \left(\nabla_\theta \mathcal{L}_{t_{1:N}}^M (\theta^\star)\Big|_{\theta=\theta}\right)^\top\right]$$

$$\overset{i.i.d.}{=} \frac{1}{M}\mathbb{E}\left[\nabla_\theta l_{t_{1:N}} (\theta;m)\Big|_{\theta=\theta^\star} \left(\nabla_\theta l_{t_{1:N}} (\theta;m)\Big|_{\theta=\theta^\star}\right)^\top\right] ,$$

where

$$\nabla_\theta l_{t_{1:N}}(\theta^\star;m) = -\frac{1}{N}\sum_{i=1}^N \left(\underbrace{\nabla_\theta \log p_i^\theta(Y_i)\Big|_{\theta=\theta^\star} - \bar{g}_{\theta^\star}(Y_i)}_{D_i(y_i)}\right) .$$

Hence,

$$\mathbb{E}\left[\nabla_\theta l_{t_{1:N}} (\theta;m)\Big|_{\theta=\theta^\star} \left(\nabla_\theta l_{t_{1:N}} (\theta;m)\Big|_{\theta=\theta^\star}\right)^\top\right]$$

$$= \frac{1}{N^2}\mathbb{E}\left[\left(\sum_i D_i(Y_i)\right)\left(\sum_i D_i(Y_i)\right)^\top\right]$$

$$\overset{(i)}{=} \frac{1}{N^2}\left(\sum_i \mathbb{E}_{p_i}\left[D_i(Y_i)D_i(Y_i)^\top\right] + \sum_{i\neq j} \mathbb{E}_{p_i}\left[D_i(Y_i)\right]\mathbb{E}_{p_j}\left[D_j(Y_j)\right]^T\right) ,$$

where $(i)$ bases on the i.i.d. assumption and

$$\sum_i \mathbb{E}_{p_i}\left[D_i(Y_i)D_i(Y_i)^\top\right] = N\mathcal{I}_{t_{1:N}} .$$

To explore the second term, first notice that

$$\sum_i \mathbb{E}_{p_i}\left[D_i(Y_i)\right] = \int_\Omega \sum_i \left(\nabla_\theta p_i^\theta(y_i)\Big|_{\theta=\theta^\star} - p_i(y_i)\frac{\sum_j \nabla_\theta p_j^\theta(y_i)\Big|_{\theta=\theta^\star}}{\sum_j p_j(y_i)}\right) dy_i = 0 .$$

Hence,

$$0 = \left(\sum_i \mathbb{E}_{p_i}\left[D_i(Y_i)\right]\right)\left(\sum_i \mathbb{E}_{p_i}\left[D_i(Y_i)\right]\right)^\top$$

$$= \sum_i \mathbb{E}_{p_i}\left[D_i(Y_i)\right]\mathbb{E}_{p_i}\left[D_i(Y_i)\right]^\top + \sum_{i\neq j} \mathbb{E}_{p_i}\left[D_i(Y_i)\right]\mathbb{E}_{p_j}\left[D_j(X_j)\right]^\top ,$$

that means

$$\sum_{i\neq j} \mathbb{E}_{p_i}\left[D_i(Y_i)\right]\mathbb{E}_{p_j}\left[D_j(X_j)\right]^\top = -\sum_i \mathbb{E}_{p_i}\left[D_i(Y_i)\right]\mathbb{E}_{p_i}\left[D_i(Y_i)\right]^\top ,$$

and therefore

$$\text{Cov}\left[\nabla_\theta \mathcal{L}^M_{t_{1:N}}(\theta)\Big|_{\theta=\theta^\star}\right] = \frac{1}{M}\mathbb{E}\left[\nabla_\theta l_{t_{1:N}}(\theta;m)\Big|_{\theta=\theta^\star}\left(\nabla_\theta l_{t_{1:N}}(\theta;m)\Big|_{\theta=\theta^\star}\right)^\top\right]$$

$$= \frac{1}{M}\frac{1}{N^2}\left(N\mathcal{I}_{t_{1:N}} - \sum_i \mathbb{E}_{p_i}\left[D_i(Y_i)\right]\mathbb{E}_{p_i}\left[D_i(Y_i)\right]^\top\right).$$

$\square$

**Lemma D.7** (Asymptotic normality). $\sqrt{M}(\hat{\theta}^M_{t_{1:N}} - \theta^\star_{t_{1:N}})$ *is asymptotically normal with zero mean and covariance matrix* $\Sigma_{t_{1:N}}$

$$\Sigma_{t_{1:N}} = \frac{1}{N}(\mathcal{I}_{t_{1:N}} + \mathcal{P}_{t_{1:N}})^{-1}\left(\mathcal{I}_{t_{1:N}} - \frac{1}{N}\sum_i \mathbb{E}_{p_i}\left[D_i(Y_i)\right]\mathbb{E}_{p_i}\left[D_i(Y_i)\right]^\top + \mathcal{C}_{t_{1:N}}\right)(\mathcal{I}_{t_{1:N}} + \mathcal{P}_{t_{1:N}})^{-1},$$

*if* $\mathcal{I}_{t_{1:N}}$ *is full rank and* $\mathcal{P}_{t_{1:N}}$ *is positive-definite.*

*Proof.* We first Taylor expand $\nabla_\theta\left(\mathcal{L}^M_{t_{1:N}}(\theta) + \mathcal{J}^M_{t_{1:N}}(\theta)\right)$ at $\hat{\theta}^M_{t_{1:N}}$ around $\theta^\star$:

$$0 = \nabla_\theta\left(\mathcal{L}^M_{t_{1:N}}(\theta) + \mathcal{J}^M_{t_{1:N}}(\theta)\right)\Big|_{\theta=\hat{\theta}^M_{t_{1:N}}} \approx \left(\nabla_\theta\mathcal{L}^M_{t_{1:N}}(\theta) + \nabla_\theta\mathcal{J}^M_{t_{1:N}}(\theta)\right)\Big|_{\theta=\theta^\star}$$

$$+ \left(\nabla^2_\theta\mathcal{L}^M_{t_{1:N}}(\theta) + \nabla^2_\theta\mathcal{J}^M_{t_{1:N}}(\theta)\right)\Big|_{\theta=\theta^\star}(\hat{\theta}^M_{1:N} - \theta^\star). \quad (60)$$

Therefore, we have

$$\sqrt{M}(\hat{\theta}^M_{t_{1:N}} - \theta^\star) \approx -\left[\left(\nabla^2_\theta\mathcal{L}^M_{t_{1:N}}(\theta) + \nabla^2_\theta\mathcal{J}^M_{t_{1:N}}(\theta)\right)\Big|_{\theta=\theta^\star}\right]^{-1}\sqrt{M}\left(\nabla_\theta\mathcal{L}^M_{t_{1:N}}(\theta) + \nabla_\theta\mathcal{J}^M_{t_{1:N}}(\theta)\right)\Big|_{\theta=\theta^\star}.$$

By assumption D.1 and Lemma D.4, $\nabla^2_\theta\mathcal{L}^M_{t_{1:N}}(\theta)\Big|_{\theta=\theta^\star} \xrightarrow{P} \mathcal{I}_{t_{1:N}}$ and $\nabla^2_\theta\mathcal{J}^M_{t_{1:N}}(\theta)\Big|_{\theta=\theta^\star} \xrightarrow{P} \mathcal{P}_{t_{1:N}}$ for large sample size $M$. Using assumptions D.2 and D.3 and Lemmas D.5 and D.6, we see that

$$\sqrt{M}\left(\nabla_\theta\mathcal{L}^M_{t_{1:N}}(\theta) + \nabla_\theta\mathcal{J}^M_{t_{1:N}}(\theta)\right)\Big|_{\theta=\theta^\star},$$

converges in distribution to a Gaussian distribution with mean zero and covariance

$$M\left(\text{Cov}\left[\nabla_\theta\mathcal{L}_{t_{1:N}}(\theta)\Big|_{\theta=\theta^\star}\right] + \text{Cov}\left[\nabla_\theta\mathcal{J}_{t_{1:N}}(\theta)\Big|_{\theta=\theta^\star}\right]\right)$$

$$= \frac{1}{N}\mathcal{I}_{t_{1:N}} - \frac{1}{N^2}\sum_i \mathbb{E}_{p_i}\left[D_i(Y_i)\right]\mathbb{E}_{p_i}\left[D_i(Y_i)\right]^\top + \frac{1}{N}\sum_i \mathcal{C}_{t_i}.$$

In meanwhile, since $\mathcal{I}_{t_{1:N}}$ is positive-semidefinite, by using the assumption that it has full rank, we have $\mathcal{I}_{t_{1:N}}$ is a positive-definite matrix. By using the assumption that $\mathcal{P}_{t_{1:N}}$ is positive-definite, we have $\mathcal{I}_{t_{1:N}} + \mathcal{P}_{t_{1:N}}$ is positive-definite and therefore invertible.

Combing the above results, we could show that $\sqrt{M}(\hat{\theta}^M_{t_{1:N}} - \theta^\star)$ converges in distribution to a Gaussian distribution of zero mean and covariance $\Sigma_{t_{1:N}}$,

$$\Sigma_{t_{1:N}} = \frac{1}{N}(\mathcal{I}_{t_{1:N}} + \mathcal{P}_{t_{1:N}})^{-1}\left(\mathcal{I}_{t_{1:N}} - \frac{1}{N}\sum_i \mathbb{E}_{p_i}\left[D_i(Y_i)\right]\mathbb{E}_{p_i}\left[D_i(Y_i)\right]^\top + \sum_i \mathcal{C}_{t_i}\right)(\mathcal{I}_{t_{1:N}} + \mathcal{P}_{t_{1:N}})^{-1}.$$

$\square$

D.3.2. PROOF OF PROPOSITION 3.3

For clarity, we present the general `DiffCLF` loss defined in Equation (10) and the DSM loss defined in Equation (7) as references,

$$\mathcal{L}_{\text{clf}}(\theta; N) = -\mathbb{E}_{t_{1:N}} \frac{1}{N} \sum_{i=1}^{N} \mathbb{E}_{p_i} \left[ \log \frac{p_i^{\theta}(Y_i)}{\sum_j p_j^{\theta}(Y_i)} \right] \ ,$$

$$\mathcal{L}_{\text{DSM}}(\theta) = \mathbb{E}_t \mathbb{E}_{q_t} \mathbb{E}_{p_t(\cdot|\tilde{X}_t)} \left[ \left\| \nabla \log p_t^{\theta}(\tilde{Y}_t) - \nabla \log p_t(\tilde{Y}_t | \tilde{X}_t) \right\|^2 \right] \ ,$$

and the empirical losses are

$$\mathcal{L}_{\text{clf}}^{M}(\theta; N) = \frac{1}{M} \sum_{m=1}^{M} l(\theta; m), \quad l(\theta; m) = \frac{1}{N} \sum_{i=1}^{N} \log \frac{p_{t_i^{(m)}}^{\theta}(Y_i^{(m)})}{\sum_j p_{t_j^{(m)}}^{\theta}(Y_i^{(m)})} \ , \tag{61}$$

$$\mathcal{L}_{\text{DSM}}^{M}(\theta) = \frac{1}{M} \sum_{m=1}^{M} j(\theta; m), \quad j(\theta; m) = \left\| \nabla \log p_{\tilde{t}^{(m)}}^{\theta}(\tilde{Y}_{\tilde{t}^{(m)}}) - \nabla \log p_t(\tilde{Y}_{\tilde{t}^{(m)}} | \tilde{X}_{\tilde{t}^{(m)}}) \right\|^2 \ , \tag{62}$$

where $t_{1:N}^{(m)} \sim U([0,T])^N$, $Y_i^{(m)} \sim p_{t_i^{(m)}}$, $\tilde{t}^{(m)} \sim U([0,T])$, $\tilde{X}_{\tilde{t}^{(m)}} \sim q_{\tilde{t}^{(m)}}$, $\tilde{Y}_{\tilde{t}^{(m)}} \sim p_{\tilde{t}^{(m)}}(\cdot|\tilde{X}_{\tilde{t}^{(m)}})$. All samples are independent to each other, except for $\tilde{X}_{\tilde{t}^{(m)}}$ and $\tilde{Y}_{\tilde{t}^{(m)}}$.

Define the population joint loss and empirical joint loss as

$$\mathcal{L}_{\text{joint}}(\theta; N) = \mathcal{L}_{\text{DSM}}(\theta) + \mathcal{L}_{\text{clf}}(\theta; N) \ , \tag{63}$$

$$\mathcal{L}_{\text{joint}}^{M}(\theta; N) = \mathcal{L}_{\text{DSM}}^{M}(\theta) + \mathcal{L}_{\text{clf}}^{M}(\theta; N) \ . \tag{64}$$

**Proposition 3.3** (Formal). *Let $\hat{\theta}^M$ be the optimal parameter learned with the empirical joint loss (Equation (64)). If conditions (a) to (c) are fulfilled, i.e.*

*(a)* $\sup_\theta |\mathcal{L}_{\text{joint}}^{M}(\theta) - \mathcal{L}_{\text{joint}}(\theta)| \xrightarrow{P} 0$ *as $M \to \infty$,*

*(b) The matrix $\mathcal{I} = \mathbb{E}_{t_{1:N}} \left[ \frac{1}{N} \sum_{i=1}^{N} \mathbb{E}_{p_i} \left[ D_i(Y_i; t_{1:N}) D_i(Y_i; t_{1:N})^\top \right] \right]$ has full rank, where*

$$D_i(y_i; t_{1:N}) = \nabla_\theta \log p_i^\theta(y_i) \Big|_{\theta=\theta^\star} - \bar{g}_{t_{1:N}}^{\theta^\star}(y_i) \quad \text{and} \quad \bar{g}_{t_{1:N}}^{\theta^\star}(y) = \frac{\sum_{j=1}^{N} p_j(y) \nabla_\theta \log p_j^\theta(y) \Big|_{\theta=\theta^\star}}{\sum_{j=1}^{N} p_j(y)} \ .$$

*(c) The matrices $(\mathcal{P}_t)_t$ are positive-definite, where*

$$\lim_{M \to \infty} \nabla_\theta^2 \left( \frac{1}{M} \sum_m \| \nabla \log p_t^\theta(\tilde{Y}_t^{(m)}) - \nabla \log p_{t^{(m)}}(\tilde{Y}_t^{(m)} | \tilde{X}_t^{(m)}) \|^2 \right) \xrightarrow{P} \mathcal{P}_t \ ,$$

*with $\tilde{X}_t^{(m)} \overset{i.i.d.}{\sim} q_t$ and $\tilde{Y}_t^{(m)} \overset{i.i.d.}{\sim} p_t(\cdot|\tilde{X}_t^{(m)})$.*

*Then $\hat{\theta}^M$ has the following properties*

*(1) **(Consistency)** $\hat{\theta}^M$ is a consistent estimator of $\theta^\star$*

*(2) **(Asymptotically Normality)** $\sqrt{M}(\hat{\theta}^M - \theta^\star)$ is asymptotically normal with zero mean and covariance matrix $\Sigma$,*

$$\Sigma = (\mathcal{I} + \mathcal{P})^{-1} \left( \underbrace{\frac{1}{N} \mathcal{I} - \frac{1}{N^2} \mathbb{E}_{t_{1:N}} \left[ \sum_i \mathbb{E}_{p_{t_i}} [D_i(Y_i; t_{1:N})] \mathbb{E}_{p_{t_i}} [D_i(Y_i; t_{1:N})]^\top \right]}_{= \frac{1}{N^2} \mathbb{E}_{t_{1:N}} \sum_i \text{Cov}_{p_{t_i}} (D_i(Y_i; t_{1:N}))} + \mathcal{C} \right) (\mathcal{I} + \mathcal{P})^{-1} \ , \tag{65}$$

*where $D_i(y_i; t_{1:N})$ is defined in condition (b), $\mathcal{P} = \mathbb{E}_t[\mathcal{P}_t]$, and $\mathcal{C} = \mathbb{E}_t[\mathcal{C}_t]$ with*

$$\mathcal{C}_t = \mathrm{Cov}\left[\nabla_\theta\left(\left\|\nabla\log p_t^\theta(\tilde{Y}_t) - \nabla\log p_t(\tilde{Y}_t|\tilde{X}_t)\right\|^2\right)\right],$$

*with $\tilde{X}_t \sim q_t$ and $\tilde{Y}_t \sim p_t(\cdot|\tilde{X}_t)$.*

*(3) (**Error Bound**) For large sample sizes $M$, $\mathbb{E}[\|\hat{\theta}^M - \theta^\star\|^2] \approx \mathrm{tr}(\Sigma)/M$.*

*Proof.* Firstly, recall that Proposition 3.1 shows that $\theta^\star$ is a unique minimizer. Following standard procedure in $M$-estimators, and with condition (a), the consistency is trivial to prove.

To show the asymptotical normality and the error bound, we require the following quantities evaluated at the optimality $\theta^\star$:

(i) the asymptotic Hessian of loss,

$$\lim_{M\to\infty}\nabla_\theta^2\mathcal{L}_{\mathrm{joint}}^M(\theta)\Big|_{\theta=\theta^\star} = \lim_{M\to\infty}\nabla_\theta^2\mathcal{L}_{\mathrm{DSM}}^M(\theta)\Big|_{\theta=\theta^\star} + \lim_{M\to\infty}\nabla_\theta^2\mathcal{L}_{\mathrm{clf}}^M(\theta)\Big|_{\theta=\theta^\star},$$

(ii) the mean of gradient of loss,

$$\mathbb{E}\left[\nabla_\theta\mathcal{L}_{\mathrm{joint}}^M(\theta)\Big|_{\theta=\theta^\star}\right] = \mathbb{E}\left[\nabla_\theta\mathcal{L}_{\mathrm{DSM}}^M(\theta)\Big|_{\theta=\theta^\star}\right] + \mathbb{E}\left[\nabla_\theta\mathcal{L}_{\mathrm{clf}}^M(\theta)\Big|_{\theta=\theta^\star}\right],$$

(iii) the covariance of gradient of loss,

$$\mathrm{Cov}\left[\nabla_\theta\mathcal{L}_{\mathrm{joint}}^M(\theta)\Big|_{\theta=\theta^\star}\right] = \mathrm{Cov}\left[\nabla_\theta\mathcal{L}_{\mathrm{DSM}}^M(\theta)\Big|_{\theta=\theta^\star}\right] + \mathrm{Cov}\left[\nabla_\theta\mathcal{L}_{\mathrm{clf}}^M(\theta)\Big|_{\theta=\theta^\star}\right],$$

**(i)** Following the proof of Lemma D.7, we have

$$\lim_{M\to\infty}\nabla_\theta^2\mathcal{L}_{\mathrm{clf}}^M(\theta)\Big|_{\theta=\theta^\star} \xrightarrow{P} \mathbb{E}_{t_{1:N}}[\mathcal{I}_{t_{1:N}}] := \mathcal{I} \quad \text{and} \quad \lim_{M\to\infty}\nabla_\theta^2\mathcal{L}_{\mathrm{DSM}}^M(\theta)\Big|_{\theta=\theta^\star} \xrightarrow{P} \mathbb{E}_t[\mathcal{P}_t] := \mathcal{P}.$$

By the condition (b), the semi-positive definite matrix $\mathcal{I}$ has full rank, meaning that it is positive-definite. By the condition (c), $\mathcal{P}$ is a weighted sum of positive-definite matrices and therefore it is positive-definite as well. Combing these, $\mathcal{I} + \mathcal{P}$ is positive-definite, and therefore is invertible.

**(ii)** By leveraging the i.i.d. condition for Monte Carlo samples, we have

$$\mathbb{E}\left[\nabla_\theta\mathcal{L}_{\mathrm{clf}}^M(\theta)\Big|_{\theta=\theta^\star}\right] = \mathbb{E}\left[\nabla_\theta l(\theta; m)\Big|_{\theta=\theta^\star}\right] = \mathbb{E}_{t_{1:N}}\mathbb{E}_{Y_{1:N}^{(m)}|t_{1:N}}\left[\nabla_\theta l(\theta; m)\Big|_{\theta=\theta^\star}\right],$$

$$\mathbb{E}\left[\nabla_\theta\mathcal{L}_{\mathrm{DSM}}^M(\theta)\Big|_{\theta=\theta^\star}\right] = \mathbb{E}\left[\nabla_\theta j(\theta; m)\Big|_{\theta=\theta^\star}\right] = \mathbb{E}_t\mathbb{E}_{\tilde{X}_t}\mathbb{E}_{\tilde{Y}|\tilde{X}}\left[\nabla_\theta j(\theta; m)\Big|_{\theta=\theta^\star}\right],$$

where $\mathbb{E}_{Y_{1:N}^{(m)}|t_{1:N}}\left[\nabla_\theta l(\theta; m)\Big|_{\theta=\theta^\star}\right] = \mathbb{E}_{\tilde{X}_t|t}\mathbb{E}_{\tilde{Y}_t|\tilde{X}_t}\left[\nabla_\theta j(\theta; m)\Big|_{\theta=\theta^\star}\right] = 0$ is given by Lemma D.5 and assumption D.2. Hence, $\mathbb{E}\left[\nabla_\theta\mathcal{L}_{\mathrm{joint}}^M(\theta)\Big|_{\theta=\theta^\star}\right] = 0$.

**(iii)** Again, we apply the i.i.d. condition,

$$\mathrm{Cov}\left[\nabla_\theta\mathcal{L}_{\mathrm{clf}}^M(\theta)\Big|_{\theta=\theta^\star}\right] = \frac{1}{M}\mathrm{Cov}\left[\nabla_\theta l(\theta; m)\Big|_{\theta=\theta^\star}\right] = \frac{1}{M}\mathbb{E}\left[\nabla_\theta l(\theta; m)\Big|_{\theta=\theta^\star}\left(\nabla_\theta l(\theta; m)\Big|_{\theta=\theta^\star}\right)^\top\right],$$

$$\mathrm{Cov}\left[\nabla_\theta\mathcal{L}_{\mathrm{DSM}}^M(\theta)\Big|_{\theta=\theta^\star}\right] = \frac{1}{M}\mathrm{Cov}\left[\nabla_\theta j(\theta; m)\Big|_{\theta=\theta^\star}\right] = \frac{1}{M}\mathbb{E}\left[\nabla_\theta j(\theta; m)\Big|_{\theta=\theta^\star}\left(\nabla_\theta j(\theta; m)\Big|_{\theta=\theta^\star}\right)^\top\right],$$

where we could split the expectation again [12]

$$\frac{1}{M}\mathbb{E}\left[\nabla_\theta l(\theta;m)\Big|_{\theta=\theta^\star}\left(\nabla_\theta l(\theta;m)\Big|_{\theta=\theta^\star}\right)^\top\right] = \frac{1}{M}\mathbb{E}_{t_{1:N}}\mathbb{E}_{Y_{1:N}^{(m)}|t_{1:N}}\left[\nabla_\theta l(\theta;m)\Big|_{\theta=\theta^\star}\left(\nabla_\theta l(\theta;m)\Big|_{\theta=\theta^\star}\right)^\top\right]$$

$$\stackrel{(a)}{=} \frac{1}{M}\mathbb{E}_{t_{1:N}}\left[\frac{1}{N^2}\left(N\mathcal{I}_{t_{1:N}} - \sum_i \mathbb{E}_{p_{t_i}}[D_i(Y_i)]\,\mathbb{E}_{p_{t_i}}[D_i(Y_i)]^\top\right)\right],$$

$$\frac{1}{M}\mathbb{E}\left[\nabla_\theta j(\theta;m)\Big|_{\theta=\theta^\star}\left(\nabla_\theta j(\theta;m)\Big|_{\theta=\theta^\star}\right)^\top\right] = \frac{1}{M}\mathbb{E}_t\mathbb{E}_{\tilde{X}_t^{(m)}|t}\mathbb{E}_{\tilde{Y}_t^{(m)}|\tilde{X}_t^{(m)}}\left[\nabla_\theta j(\theta;m)\Big|_{\theta=\theta^\star}\left(\nabla_\theta j(\theta;m)\Big|_{\theta=\theta^\star}\right)^\top\right]$$

$$\stackrel{(b)}{=} \frac{1}{M}\mathbb{E}_t\left[\mathcal{C}_t\right] := \frac{1}{M}\mathcal{C},$$

where $(a)$ is given by Lemma D.6 and $(b)$ is by Assumption D.3. Hence,

$$\mathrm{Cov}\left[\sqrt{M}\nabla_\theta \mathcal{L}_{\mathrm{joint}}^M(\theta^\star)\right] = M\,\mathrm{Cov}\left[\nabla_\theta\mathcal{L}_{\mathrm{clf}}^M(\theta)\Big|_{\theta=\theta^\star}\right] + M\,\mathrm{Cov}\left[\nabla_\theta\mathcal{L}_{\mathrm{DSM}}^M(\theta)\Big|_{\theta=\theta^\star}\right]$$

$$= \frac{1}{N}\mathcal{I} - \frac{1}{N^2}\mathbb{E}_{t_{1:N}}\left[\sum_i \mathbb{E}_{p_{t_i}}[D_i(Y_i)]\,\mathbb{E}_{p_{t_i}}[D_i(Y_i)]^\top\right] + \mathcal{C}.$$

Combining (i)-(iii) and according to Taylor expansion which gives

$$\sqrt{M}(\hat{\theta}^M - \theta^\star) \approx -\left[\nabla_\theta^2\mathcal{L}_{\mathrm{joint}}^M(\theta)\Big|_{\theta=\theta^\star}\right]^{-1}\left(\sqrt{M}\nabla_\theta\mathcal{L}_{\mathrm{joint}}^M(\theta)\Big|_{\theta=\theta^\star}\right), \tag{66}$$

we have

$$\mathbb{E}\left[\sqrt{M}(\hat{\theta}^M - \theta^\star)\right] = 0 \quad\text{and}\quad \lim_{M\to\infty}\mathrm{Cov}\left[\sqrt{M}(\hat{\theta}^M - \theta^\star)\right] \stackrel{P}{\to} \Sigma,$$

where

$$\Sigma := (\mathcal{I} + \mathcal{P})^{-1}\left(\frac{1}{N}\mathcal{I} - \frac{1}{N^2}\mathbb{E}_{t_{1:N}}\left[\sum_i \mathbb{E}_{p_{t_i}}[D_i(Y_i)]\,\mathbb{E}_{p_{t_i}}[D_i(Y_i)]^\top\right] + \mathcal{C}\right)(\mathcal{I} + \mathcal{P})^{-1},$$

and notice that

$$\frac{1}{N}\mathcal{I} - \frac{1}{N^2}\mathbb{E}_{t_{1:N}}\left[\sum_i \mathbb{E}_{p_{t_i}}[D_i(Y_i)]\,\mathbb{E}_{p_{t_i}}[D_i(Y_i)]^\top\right] = \frac{1}{N^2}\mathbb{E}_{t_{1:N}}\left[\sum_i \mathrm{Cov}_{p_{t_i}}[D_i(Y_i)]\right].$$

That is, $\sqrt{M}(\hat{\theta}^M - \theta^\star)$ in probability converges to a Gaussian distribution with zero mean and covariance $\Sigma$. $\qquad\square$

**Proposition D.8.** *Let's treat the information matrix $\mathcal{I}$ and the asymptotic covariance $\Sigma$ defined in Proposition 3.3 as functions depending on $N$, i.e.*

$$\mathcal{I}(N) = \mathbb{E}_{t_{1:N}}\left[\frac{1}{N}\sum_{i=1}^N \mathbb{E}_{p_{t_i}}\left[D_i(Y_i; t_{1:N})D_i(Y_i; t_{1:N})^\top\right]\right],$$

$$\Sigma(N) = (\mathcal{I}(N) + \mathcal{P})^{-1}\left(\frac{1}{N}\mathcal{I}(N) - \frac{1}{N^2}\mathbb{E}_{t_{1:N}}\left[\sum_i \mathbb{E}_{p_{t_i}}[D_i(Y_i; t_{1:N})]\,\mathbb{E}_{p_{t_i}}[D_i(Y_i; t_{1:N})]^\top\right] + \mathcal{C}\right)(\mathcal{I}(N) + \mathcal{P})^{-1}.$$

*By assuming that $\int_0^T \mathbb{E}_{p_t}\left[\|\nabla_\theta \log p_t^{\theta^\star}(X)\|^2\right]\,dt < \infty$, then*

$$\lim_{N\to\infty}\mathcal{I}(N) = \mathcal{I}_\infty := \int_0^T \mathbb{E}_{p_t}\left[D_t(y)D_t(y)^\top\right]\,dt,$$

---

[12]For clarity, we write $D_i(y; t_{1:N})$ as $D_i(y)$ in the proof.

*where $D_t(y) = \nabla_\theta \log p_t^\theta(y)\big|_{\theta=\theta^\star} - c(y)$ and $c(y) = \dfrac{\int_0^T p_t(y)\nabla_\theta \log p_t^\theta(y)\big|_{\theta=\theta^\star} \mathrm{d}t}{\int_0^T p_t(y)\mathrm{d}t}$. By further assuming $\mathcal{I}_\infty$ is finite and non-singular, then*

$$\lim_{N\to\infty} N\left(\Sigma(N) - (\mathcal{I}_\infty - \mathcal{P})^{-1} \mathcal{C} (\mathcal{I}_\infty - \mathcal{P})^{-1}\right)$$

$$= (\mathcal{I}_\infty - \mathcal{P})^{-1}\left(\int_0^T \left(\mathbb{E}_{p_t}\left[D_t(Y_t)\right]\right)\left(\mathbb{E}_{p_t}\left[D_t(Y_t)\right]\right)^\top \mathrm{d}t\right)(\mathcal{I}_\infty - \mathcal{P})^{-1}.$$

*Proof.* First note that, for each fixed $x$, by the strong law of large numbers applied to the i.i.d. sequence $\{t_j\}_{j\ge 1}$ with $t_j \sim \mathrm{Unif}[0, T]$,

$$\frac{1}{N}\sum_{j=1}^N p_j(y)\nabla_\theta \log p_j^\theta(y)\Big|_{\theta=\theta^\star} \xrightarrow[N\to\infty]{a.s.} \mathbb{E}_t\left[p_t(y)\nabla_\theta \log p_t^\theta(y)\Big|_{\theta=\theta^\star}\right] = \frac{1}{T}\int_0^T p_t(y)\nabla_\theta \log p_t^\theta(y)\Big|_{\theta=\theta^\star} \mathrm{d}t,$$

and

$$\frac{1}{N}\sum_{j=1}^N p_j(y) \xrightarrow[N\to\infty]{a.s.} \mathbb{E}_t[p_t(y)] = \frac{1}{T}\int_0^T p_t(y)\,\mathrm{d}t,$$

so that, whenever the denominator is non-zero,

$$\bar{g}_{t_{1:N}}^{\theta^\star}(y) = \frac{\sum_{j=1}^N p_j(y)\nabla_\theta \log p_j^{\theta^\star}(y)\big|_{\theta=\theta^\star}}{\sum_{j=1}^N p_j(y)} \xrightarrow[N\to\infty]{a.s.} \frac{\int_0^T p_t(y)\nabla_\theta \log p_t^\theta(y)\big|_{\theta=\theta^\star} \mathrm{d}t}{\int_0^T p_t(y)\,\mathrm{d}t} = c(y).$$

Consequently,

$$D_i(y; t_{1:N}) = \nabla_\theta \log p_i^{\theta^\star}(y)\Big|_{\theta=\theta^\star} - \bar{g}_{t_{1:N}}^{\theta^\star}(y) \xrightarrow[N\to\infty]{a.s.} D_{t_i}(y) := \nabla_\theta \log p_{t_i}^{\theta^\star}(y)\Big|_{\theta=\theta^\star} - c(y).$$

By the assumed square integrability of $\nabla_\theta \log p_t^{\theta^\star}\big|_{\theta=\theta^\star}$ and the definition of $c(y)$, the random matrices

$$D_i(Y; t_{1:N})D_i(Y; t_{1:N})^\top$$

are dominated in $L^1$ by an integrable function that does not depend on $N$. Hence, by dominated convergence,

$$\lim_{N\to\infty} \mathbb{E}_{p_i}\left[D_i(Y_i; t_{1:N})D_i(Y_i; t_{1:N})^\top\right] = \mathbb{E}_{p_{t_i}}\left[D_{t_i}(Y_i)D_{t_i}(Y_i)^\top\right] \quad \text{a.s. in } t_{1:N}.$$

Taking the average over $i$ and then expectation with respect to $t_{1:N}$, and applying again the law of large numbers in $i$ together with dominated convergence, yields

$$\lim_{N\to\infty} \mathcal{I}(N) = \mathbb{E}_t\left[\mathbb{E}_{p_t}\left[D_t(Y_t)D_t(Y_t)^\top\right]\right] = \frac{1}{T}\int_0^T \mathbb{E}_{p_t}\left[D_t(Y_t)D_t(Y_t)^\top\right]\mathrm{d}t = \mathcal{I}_\infty.$$

The same argument applies to the means $m_i$: since

$$m_i = \mathbb{E}_{p_i}[D_i(Y_i; t_{1:N})] \xrightarrow[N\to\infty]{} \mathbb{E}_{p_{t_i}}[D_{t_i}(Y_i)],$$

and the latter are square integrable in $t$ by assumption, we obtain

$$\lim_{N\to\infty} \mathbb{E}_{t_{1:N}}\left[\frac{1}{N}\sum_{i=1}^N m_i m_i^\top\right] = \mathbb{E}_t\left[\left(\mathbb{E}_{p_t}[D_t(Y_t)]\right)\left(\mathbb{E}_{p_t}[D_t(Y_t)]\right)^\top\right]$$

$$= \frac{1}{T}\int_0^T \left(\mathbb{E}_{p_t}[D_t(Y_t)]\right)\left(\mathbb{E}_{p_t}[D_t(Y_t)]\right)^\top \mathrm{d}t$$

$$= \mathcal{Q}_\infty.$$

Using the definition of $\Sigma(N)$, we can write

$$N \left( \Sigma(N) - (\mathcal{I}_\infty - \mathcal{P})^{-1} \mathcal{C} (\mathcal{I}_\infty - \mathcal{P})^{-1} \right)$$

$$= (\mathcal{I}(N) + \mathcal{P})^{-1} \left( \mathcal{I}(N) - \frac{1}{N} \mathbb{E}_{t_{1:N}} \left[ \sum_i m_i m_i^\top \right] + N\mathcal{C} \right) (\mathcal{I}(N) + \mathcal{P})^{-1} - N (\mathcal{I}_\infty - \mathcal{P})^{-1} \mathcal{C} (\mathcal{I}_\infty - \mathcal{P})^{-1} .$$

By the first part of the proof, $\mathcal{I}(N) \to \mathcal{I}_\infty$ and $\frac{1}{N} \mathbb{E}_{t_{1:N}} \left[ \sum_i m_i m_i^\top \right] \to \mathcal{Q}_\infty$. Since $\mathcal{I}_\infty$ is non-singular, the matrix inversion map is continuous at $\mathcal{I}_\infty$, and thus $\mathcal{I}(N)^{-1} \to \mathcal{I}_\infty^{-1}$. Together with the convergence of the mean outer products to $\mathcal{J}_\infty$, we obtain

$$\lim_{N \to \infty} N \left( \Sigma(N) - (\mathcal{I}_\infty - \mathcal{P})^{-1} \mathcal{C} (\mathcal{I}_\infty - \mathcal{P})^{-1} \right) = (\mathcal{I}_\infty - \mathcal{P})^{-1} \mathcal{Q}_\infty (\mathcal{I}_\infty - \mathcal{P})^{-1} ,$$

which completes the proof. $\qquad\square$

**Corollary D.9.** *Under the assumptions of Proposition D.8, for any matrix norm $\| \cdot \|$ there exists a constant $C > 0$ and $N_0 \in \mathbb{N}$ such that*

$$\| \Sigma(N) - (\mathcal{I}_\infty - \mathcal{P})^{-1} \mathcal{C} (\mathcal{I}_\infty - \mathcal{P})^{-1} \| \leq \frac{C}{N} \quad \text{for all } N \geq N_0.$$

*In other words, for large time-batch size $N$, the covariance matrix $\Sigma(N)$ decays at rate $1/N$ as $N$ increases.*

*Proof.* For clarity, let

$$f(N) = \Sigma(N) - (\mathcal{I}_\infty - \mathcal{P})^{-1} \mathcal{C} (\mathcal{I}_\infty - \mathcal{P})^{-1} .$$

By Proposition D.8 we have

$$\lim_{N \to \infty} N f(N) = (\mathcal{I}_\infty - \mathcal{P})^{-1} \mathcal{Q}_\infty (\mathcal{I}_\infty - \mathcal{P})^{-1} := \Sigma_\infty.$$

Let $\| \cdot \|$ be any matrix norm. The map $A \mapsto \|A\|$ is continuous, so

$$\lim_{N \to \infty} \left\| N\Sigma(N) - N (\mathcal{I}_\infty - \mathcal{P})^{-1} \mathcal{C} (\mathcal{I}_\infty - \mathcal{P})^{-1} - \Sigma_\infty \right\| = 0.$$

Hence there exists $N_0 \in \mathbb{N}$ such that, for all $N \geq N_0$,

$$\| N f(N) - \Sigma_\infty \| \leq 1.$$

For such $N$ we obtain

$$\| N f(N) \| \leq \| N f(N) - \Sigma_\infty \| + \| \Sigma_\infty \| \leq 1 + \| \Sigma_\infty \| \implies \| f(N) \| \leq \frac{1 + \| \Sigma_\infty \|}{N} \quad \text{for all } N \geq N_0.$$

Setting $C := 1 + \| \Sigma_\infty \|$ completes the proof. $\qquad\square$

## D.4. Proof of Proposition 4.1

For clarity of the proof, we first present two lemmata as important stepping stones.

### D.4.1. LEMMATA

**Lemma D.10** (Kolmogorov semigroup expansion)**.** *Let $X_t$ solves the Itô SDE $\mathrm{d}X_t = f(t, X_t)\mathrm{d}t + g(t, X_t)\mathrm{d}W_t$ and $\phi \in \mathbb{C}_b^4$. Then with a small time increment $\delta$,*

$$\mathbb{E}[\phi(X_{t+\delta})|X_t = x] = \phi(x) + \delta \mathcal{L}_t \phi(x) + \frac{\delta^2}{2} \mathcal{L}_t^2 \phi(x) + \mathcal{O}(\delta^2) , \tag{67}$$

*or equivalently*

$$\mathbb{E}_{p_{t+\delta}}[\phi(X)] = \mathbb{E}_{p_t}[\phi(X)] + \delta \mathbb{E}_{p_t}[\mathcal{L}_t \phi(X)] + \frac{\delta^2}{2} \mathbb{E}_{p_t}[\mathcal{L}_t^2 \phi(X)] + \mathcal{O}(\delta^2) , \tag{68}$$

*where $\mathcal{L}_t \phi := f(t, \cdot)^\top \nabla \phi + \frac{1}{2} g(t, \cdot) g(t, \cdot)^\top \Delta \phi$ is the time-dependent backward generator.*

**Lemma D.11** (Generator-Adjoint Identity)**.** *Given the Itô SDE, $\phi$, and $\mathcal{L}_t$ defined in Lemma D.10, we have*

$$\mathbb{E}_{p_t}[\mathcal{L}_t \phi(X)] = \mathbb{E}_{p_t} \left[ \phi(X) \frac{\partial}{\partial t} \log p_t(X) \right] . \tag{69}$$

D.4.2. PROOF OF PROPOSITION

**Proposition D.12.** *Let $t \in [0, T)$ and $\delta > 0$, we have*

$$\lim_{\delta \to 0^+} \frac{8}{\delta^2} \left( \mathcal{L}_{\mathrm{clf}}(\theta; t, t + \delta) - \log 2 \right) = \mathcal{L}_{tSM}(\theta; t) + C ,  \tag{70}$$

*where $C = \mathbb{E}_{p_t} \left[ \left( \frac{\partial}{\partial t} \log p_t(Y_t) \right)^2 \right]$ is a constant with respect to $\theta$.*

*Proof.* In the binary case, we first rewrite the classification loss as follows

$$\mathcal{L}_{\mathrm{clf}}(\theta; t, t + \delta) = -\frac{1}{2} \left\{ \mathbb{E}_{p_t} \left[ \log \frac{p_t^\theta(x)}{p_t^\theta(x) + p_{t+\delta}^\theta(x)} \right] + \mathbb{E}_{p_{t+\delta}} \left[ \log \frac{p_{t+\delta}^\theta(x)}{p_t^\theta(x) + p_{t+\delta}^\theta(x)} \right] \right\}$$

$$= -\frac{1}{2} \left\{ \underbrace{\mathbb{E}_{p_t} \left[ \log \sigma(\log p_t^\theta(x) - \log p_{t+\delta}^\theta(x)) \right]}_{A} + \underbrace{\mathbb{E}_{p_{t+\delta}} \left[ \log \sigma(\log p_{t+\delta}^\theta(x) - \log p_t^\theta(x)) \right]}_{B} \right\} ,$$

where $\sigma(z) = 1/(1 + e^{-z})$ is the sigmoid function. By Taylor expansion,

$$\log p_t^\theta(x) - \log p_{t+\delta}^\theta(x) = -\delta \frac{\partial}{\partial t} \log p_t^\theta(x) + \mathcal{O}(\delta^2),$$

$$\log \sigma(z) = -\log 2 + \frac{z}{2} - \frac{z^2}{8} + \mathcal{O}(z^4).$$

Therefore, $A$ and $B$ can be simplified as

$$A = -\log 2 + \mathbb{E}_{p_t} \left[ -\frac{\delta}{2} \frac{\partial}{\partial t} \log p_t^\theta(x) - \frac{\delta^2}{8} \left( \frac{\partial}{\partial t} \log p_t^\theta(x) \right)^2 \right] + \mathcal{O}(\delta^3),$$

$$B = -\log 2 + \mathbb{E}_{p_{t+\delta}} \left[ \frac{\delta}{2} \frac{\partial}{\partial t} \log p_t^\theta(x) - \frac{\delta^2}{8} \left( \frac{\partial}{\partial t} \log p_t^\theta(x) \right)^2 \right] + \mathcal{O}(\delta^3).$$

Next, we use Lemma D.10 with $\phi_1(x) = \frac{\partial}{\partial t} \log p_t^\theta(x)$ and $\phi_2(x) = \left( \frac{\partial}{\partial t} \log p_t^\theta(x) \right)^2$ as follows

$$\mathbb{E}_{p_{t+\delta}} [\phi_1(x)] = \mathbb{E}_{p_t} \left[ \frac{\partial}{\partial t} \log p_t^\theta(x) \right] + \delta \mathbb{E}_{p_t} \left[ \mathcal{L}_t \frac{\partial}{\partial t} \log p_t^\theta(x) \right] + \mathcal{O}(\delta^2),$$

$$\mathbb{E}_{p_{t+\delta}} [\phi_2(x)] = \mathbb{E}_{p_t} \left[ \left( \frac{\partial}{\partial t} \log p_t^\theta(x) \right)^2 \right] + \mathcal{O}(\delta).$$

Plugging $\mathbb{E}_{p_{t+\delta}} [\phi_1(x)]$ and $\mathbb{E}_{p_{t+\delta}} [\phi_2(x)]$, $B$ can be written as

$$B = -\log 2 + \frac{\delta}{2} \mathbb{E}_{p_t} \left[ \frac{\partial}{\partial t} \log p_t^\theta(x) \right] + \frac{\delta^2}{2} \mathbb{E}_{p_t} \left[ \mathcal{L}_t \frac{\partial}{\partial t} \log p_t^\theta(x) \right] - \frac{\delta^2}{8} \mathbb{E}_{p_t} \left[ \left( \frac{\partial}{\partial t} \log p_t^\theta(x) \right)^2 \right] + \mathcal{O}(\delta^3). \tag{71}$$

Therefore,

$$A + B = -2 \log 2 + \frac{\delta^2}{2} \mathbb{E}_{p_t} \left[ \mathcal{L}_t \frac{\partial}{\partial t} \log p_t^\theta(x) \right] - \frac{\delta^2}{4} \mathbb{E}_{p_t} \left[ \left( \frac{\partial}{\partial t} \log p_t^\theta(x) \right)^2 \right] + \mathcal{O}(\delta^3).$$

By Lemma D.11, the $\mathbb{E}_{p_t} \left[ \mathcal{L}_t \frac{\partial}{\partial t} \log p_t^\theta(x) \right]$ term could be written as

$$\mathbb{E}_{p_t} \left[ \mathcal{L}_t \frac{\partial}{\partial t} \log p_t^\theta(x) \right] = \mathbb{E}_{p_t} \left[ \frac{\partial}{\partial t} \log p_t^\theta(x) \frac{\partial}{\partial t} \log p_t(x) \right] , \tag{72}$$

which results in

$$A + B + 2\log 2$$

$$= -\frac{\delta^2}{4}\left(\mathbb{E}_{p_t}\left[\left(\frac{\partial}{\partial t}\log p_t^\theta(x)\right)^2\right] - 2\mathbb{E}_{p_t}\left[\frac{\partial}{\partial t}\log p_t^\theta(x)\frac{\partial}{\partial t}\log p_t(x)\right]\right) + \mathcal{O}(\delta^3)$$

$$= -\frac{\delta^2}{4}\left(\mathbb{E}_{p_t}\left[\left(\frac{\partial}{\partial t}\log p_t^\theta(x) - \frac{\partial}{\partial t}\log p_t(x)\right)^2\right] + \mathbb{E}_{p_t}\left[\left(\frac{\partial}{\partial t}\log p_t(x)\right)^2\right]\right) + \mathcal{O}(\delta^3).$$

Therefore, given a small time-increment $\delta$, the binary-classification loss can be written as

$$\mathcal{L}_{\text{clf}}(\theta; t, t+\delta) = \frac{\delta^2}{8}\mathbb{E}_{p_t}\left[\left(\frac{\partial}{\partial t}\log p_t^\theta(x) - \frac{\partial}{\partial t}\log p_t(x)\right)^2\right] + \frac{\delta^2}{8}\mathbb{E}_{p_t}\left[\left(\frac{\partial}{\partial t}\log p_t(x)\right)^2\right] + \log 2 + \mathcal{O}(\delta^3). \quad (73)$$

Since $C = \frac{\delta^2}{8}\mathbb{E}_{p_t}\left[\left(\frac{\partial}{\partial t}\log p_t(x)\right)^2\right] + \log 2$ is constant *w.r.t.* $\theta$, it recovers the tSM regularization. $\qquad\square$

### D.5. Proof of Proposition A.1

**Proposition A.1.** *Let $\delta > 0$. In the small step-size regime, the Bayes objective $\mathcal{L}_{\text{Bayes}}$ recovers the Fokker–Planck regularization $\mathcal{L}_{\text{FPE}}$, i.e.,*

$$\lim_{\delta\to 0}\frac{1}{\delta}\mathcal{L}_{\text{Bayes}}(\theta; t, t+\delta) = \mathcal{L}_{\text{FPE}}(\theta; t) . \quad (74)$$

*Proof.* From Equation (31), we write the full learning objective of the FPE regularzation for reference:

$$\mathcal{L}_{\text{FPE}}(\theta) = \int_0^1 \mathbb{E}_{p_t}\left[\left(\partial_t \log p_t^\theta(y) + \nabla\cdot\alpha_t(y) + \alpha_t(y)^\top\nabla\log p_t^\theta(y)\right.\right.$$

$$\left.\left. -\frac{1}{2}\beta_t^2\nabla\cdot\nabla\log p_t^\theta(y) - \frac{1}{2}\beta_t^2\|\nabla\log p_t^\theta(y)\|^2\right)^2\right]dt . \quad (75)$$

Now, let's look at the Bayes regularization. Given the EM discretizations for both the forward (34) and backward (35) kernels and assume that the $(y_{t-\delta}, y_t)$ are sampled forwardly via the Euler-Maruyama discretization, *i.e.*, $y_t = y_{t-\delta} + \alpha_{t-\delta}(y_{t-\delta})\delta + \beta_{t-\delta}\sqrt{\delta}z$, where $z \sim \mathcal{N}(0, I)$ and $\delta > 0$. Because the Bayes objective evaluates the continuous flow of a probability density, the forward transition kernel must account for the spatial volume deformation induced by the deterministic drift mapping $y \mapsto y + \alpha_{t-\delta}(y)\delta$. By the change of variables formula, the density scales by the inverse of the Jacobian determinant $|\det(I + \delta\nabla\alpha_{t-\delta}(y_{t-\delta}))|$. Using the standard matrix identity $\det(I + \delta A) = 1 + \delta\text{tr}(A) + \mathcal{O}(\delta^2)$, we can express the volume-corrected forward kernel as:

$$p_{t|t-\delta}(y_t|y_{t-\delta}) = \frac{1}{1 + \delta\nabla\cdot\alpha_{t-\delta}(y_{t-\delta}) + \mathcal{O}(\delta^2)}\mathcal{N}(y_t; y_{t-\delta} + \alpha_{t-\delta}(y_{t-\delta})\delta, \beta_{t-\delta}^2\delta I) . \quad (76)$$

Let $b_t(y) = \alpha_t(y) - \beta_t^2\nabla\log p_t(y)$. Because the forward Euler-Maruyama step gives

$$y_{t-\delta} - y_t = -\alpha_{t-\delta}(y_{t-\delta})\delta - \beta_{t-\delta}\sqrt{\delta}z = \mathcal{O}(\sqrt{\delta}), \quad (77)$$

we can Taylor expand the drift and diffusion coefficients around $(t, y_t)$. Specifically,

$$\alpha_{t-\delta}(y_{t-\delta})\delta = \alpha_t(y_t)\delta + \mathcal{O}(\delta^{3/2}) \quad\text{and}\quad \beta_{t-\delta}\sqrt{\delta} = \beta_t\sqrt{\delta} + \mathcal{O}(\delta^{3/2}). \quad (78)$$

Then the log forward and backward kernels can be expanded as:

$$\log p_{t|t-\delta}(y_t|y_{t-\delta}) = C_t - \frac{1}{2}\|z\|^2 - \log\left(1 + \delta\nabla\cdot\alpha_{t-\delta}(y_{t-\delta}) + \mathcal{O}(\delta^2)\right)$$

$$= C_t - \frac{1}{2}\|z\|^2 - \delta\nabla\cdot\alpha_t(y_t) + \mathcal{O}(\delta^{3/2}) , \tag{79}$$

$$\log p_{t-\delta|t}(y_{t-\delta}|y_t) = C_t - \frac{1}{2\beta_t^2\delta}\|\alpha_{t-\delta}(y_{t-\delta})\delta + \beta_{t-\delta}\sqrt{\delta}z - b_t(y_t)\delta\|^2$$

$$= C_t - \frac{1}{2\beta_t^2\delta}\|\alpha_t(y_t)\delta + \beta_t\sqrt{\delta}z - b_t(y_t)\delta + \mathcal{O}(\delta^{3/2})\|^2$$

$$= C_t - \frac{1}{2}\|z\|^2 - \frac{\delta}{2\beta_t^2}\|\alpha_t(y_t) - b_t(y_t)\|^2 - \frac{\sqrt{\delta}}{\beta_t}z^\top(\alpha_t(y_t) - b_t(y_t)) + \mathcal{O}(\delta) , \tag{80}$$

where $C_t = -\frac{d}{2}\log(2\pi\beta_t^2\delta)$. Plugging back $\alpha_t(y_t) - b_t(y_t) = \beta_t^2\nabla\log p_t(y_t)$, the log Radon–Nikodym derivative can be approximated as

$$\log\frac{p_{t|t-\delta}(y_t|y_{t-\delta})}{p_{t-\delta|t}(y_{t-\delta}|y_t)} = \frac{\beta_t^2\delta}{2}\|\nabla\log p_t(y_t)\|^2 + \beta_t\sqrt{\delta}z^\top\nabla\log p_t(y_t) - \delta\nabla\cdot\alpha_t(y_t) + \mathcal{O}(\delta) . \tag{81}$$

In meanwhile, one could Taylor expand the log marginal density $\log p_{t-\delta}(y_{t-\delta})$ around $(t, y_t)$:

$$\log p_{t-\delta}(y_{t-\delta}) = \log p_t(y_t) - \frac{\partial}{\partial t}\log p_t(y_t)\delta + \nabla\log p_t(y_t)^\top(y_{t-\delta} - y_t)$$

$$+ \frac{1}{2}(y_{t-\delta} - y_t)^\top\nabla^2\log p_t(y_t)(y_{t-\delta} - y_t) + \mathcal{O}(\delta^{3/2}) . \tag{82}$$

By plugging $y_t - y_{t-\delta} = \alpha_t(y_t)\delta + \beta_t\sqrt{\delta}z + \mathcal{O}(\delta^{3/2})$, we have

$$\log p_{t-\delta}(y_{t-\delta}) = \log p_t(y_t) - \frac{\partial}{\partial t}\log p_t(y_t)\delta - \nabla\log p_t(y_t)^\top\alpha_t(y_t)\delta$$

$$- \beta_t\sqrt{\delta}z^\top\nabla\log p_t(y_t) + \frac{\beta_t^2\delta}{2}z^\top\nabla^2\log p_t(y_t)z + \mathcal{O}(\delta^{3/2}) . \tag{83}$$

Now, we could plug all the approximations together, and take the expectation

$$\frac{1}{\delta}\mathbb{E}_z\left[\log p_{t-\delta}(y_{t-\delta}) - \log p_t(y_t) + \log\frac{p_{t|t-\delta}(y_t|y_{t-\delta})}{p_{t-\delta|t}(y_{t-\delta}|x_t)}\right]$$

$$= \left(-\frac{\partial}{\partial t}\log p_t(y_t) - \nabla\cdot\alpha_t(y_t) - \nabla\log p_t(y_t)^\top f(t, x_t) + \frac{\beta_t^2}{2}(\text{tr}(\nabla^2\log p_t(y_t)) + \|\nabla\log p_t(y_t)\|^2)\right) , \tag{84}$$

which recovers the Fokker Plank residual when $\delta \to 0$, and therefore RNE regularization recovers the Fokker Plank regularization in the limit. $\square$

# E. Extend DiffCLF with Bregman Divergence

The effective training objective of `DiffCLF` is to estimate the density ratios between noise levels, i.e. $p_t^\theta/p_s^\theta$ for $s, t \in [0, T]$. Bregman divergence (Bregman, 1967; Gutmann & Hirayama, 2011) could be utilized to generalize this density-ratio estimation framework.

## E.1. Bregman Divergence

The Bregman divergence (Bregman, 1967) between two functionals $f : \mathbb{R}^d \to \mathbb{R}^m$ and $g : \mathbb{R}^d \to \mathbb{R}^m$ within an underlying measure $\mu$ is defined as

$$D_\phi^\mu(f, g) = \mathbb{E}_\mu\left[\phi(f) - \phi(g) - \nabla\phi(g)\cdot(f - g)\right] , \tag{85}$$

where $\phi : \mathbb{R}^m \to \mathbb{R}$ is a strightly convex *generator* and $\nabla\phi(g)$ refers to $\nabla_g\phi(g)$. In the following context, we stick using $f$ as some known quantities while $g$ is the learnable components. Therefore, $\mathbb{E}_\mu\phi(f)$ is a constant with respect to the learnable parameters, and hence we could equivalently minimize the following loss,

$$\mathcal{L}_\phi^\mu(f, g) = \mathbb{E}_\mu \left[-\phi(g) + \nabla\phi(g) \cdot g\right] - \mathbb{E}_\mu \left[\nabla\phi(g) \cdot f\right] . \tag{86}$$

### E.2. Binary case

In the binary case, `DiffCLF` aims to match the $p_t^\theta/p_s^\theta$ with the ground truth density ratio $p_t/p_s$, for any $t, s \in [0, T]$. That is,

$$f = \frac{p_t}{p_s} \quad \& \quad g = \frac{p_t^\theta}{p_s^\theta} := r_{st}^\theta .$$

Let $\mu = p_s$,

$$\mathcal{L}_\phi^\mu(f, g) = \mathbb{E}_{p_s} \left[-\phi\left(r_{st}^\theta(x_s)\right) + \phi'\left(r_{st}^\theta(x_s)\right) r_{st}^\theta(x_s)\right] - \mathbb{E}_{p_s} \left[\phi'\left(r_{st}^\theta(x_s)\right) \frac{p_t(x_s)}{p_s(x_s)}\right]$$

$$= \mathbb{E}_{p_s} \left[-\phi\left(r_{st}^\theta(x_s)\right) + \phi'\left(r_{st}^\theta(x_s)\right) r_{st}^\theta(x_s)\right] - \mathbb{E}_{p_t} \left[\phi'\left(r_{st}^\theta(x_t)\right)\right] , \tag{87}$$

where we use $\phi'$ to denote the derivative of $\phi$ for clarity as $\phi$ is now a scalar function.

Let $\phi(r) = r \log r - (1 + r) \log(1 + r)$, where $\phi'(r) = \log r - \log(1 + r)$, the above loss recovers the binary `DiffCLF` loss (Equation (12)):

$$\mathbb{E}_{p_s} \left[ - \left( \cancel{r_{st}^\theta(x_s) \log r_{st}^\theta(x_s)} - (1 + \cancel{r_{st}^\theta(x_s)}) \log(1 + r_{st}^\theta(x_s)) \right) \right.$$

$$\left. + \cancel{r_{st}^\theta(x_s) \log \frac{r_{st}^\theta(x_s)}{(1 + r_{st}^\theta(x_s))}} \right] - \mathbb{E}_{p_t} \left[ \log \frac{r_{st}^\theta(x_s)}{(1 + r_{st}^\theta(x_s))} \right] ,$$

which can be simplified as

$$\mathbb{E}_{p_s} \left[ \log \left(1 + r_{st}^\theta(x_s)\right) \right] - \mathbb{E}_{p_t} \left[ \log \frac{r_{st}^\theta(x_t)}{1 + r_{st}^\theta(x_t)} \right] \tag{88}$$

$$= - \mathbb{E}_{p_s} \left[ \log \left( \frac{p_s^\theta(x_s)}{p_s^\theta(x_s) + p_t^\theta(x_s)} \right) \right] - \mathbb{E}_{p_t} \left[ \log \frac{p_t^\theta(x_t)}{p_s^\theta(x_t) + p_t^\theta(x_t)} \right] \tag{89}$$

$$= 2\mathcal{L}_{\text{clf}}(\theta; s, t) . \tag{90}$$

Another possible choices of $\phi(r)$ include $r \log r - r$, $- \log r$, and $\frac{1}{2}r^2$. However, these alternative choices arise singularity on either $r = 0$ or $r = \infty$. Therefore, we argue that $\phi(r) = r \log r - (1 + r) \log(1 + r)$ is the optimal option.

### E.3. Multiclass case

In the multiclass case, `DiffCLF` aims to match all the density ratios $p_{t_i}^\theta/p_{t_j}^\theta$, given $i, j \in [\![1, N]\!]$. For clarity, we write $p_i$ as $p_{t_i}$. The functionals $f$ and $g$ are vector-valued, defined as:

$$f = \left[ \frac{p_1}{\sum_{j=1}^N p_j}, ...., \frac{p_N}{\sum_{j=1}^N p_j} \right] \quad \text{and} \quad g = \left[ \frac{p_1^\theta}{\sum_{j=1}^N p_j^\theta}, ...., \frac{p_N^\theta}{\sum_{j=1}^N p_j^\theta} \right] := r_{1:N}^\theta ,$$

where $r_j^\theta = p_j^\theta / \sum_i p_i^\theta$. Let $\mu$ be the simple mixture of $\{p_j\}_{j=1}^N$, i.e. $\mu = \bar{p} := \frac{1}{N} \sum_{j=1}^N p_j$ ,

$$\mathcal{L}_\phi^\mu(f, g) = \mathbb{E}_{\bar{p}} \left[ -\phi\left(r_{1:N}^\theta(x)\right) + \nabla\phi\left(r_{1:N}^\theta(x)\right) \cdot r_{1:N}^\theta(x) \right] - \mathbb{E}_{\bar{p}} \left[ \nabla\phi\left(r_{1:N}^\theta(x)\right) \cdot \frac{\mathbf{p}(x)}{\sum_{j=1}^N p_j(x)} \right]$$

$$= \mathbb{E}_{\bar{p}} \left[ -\phi\left(r_{1:N}^\theta(x)\right) + \nabla\phi\left(r_{1:N}^\theta(x)\right) \cdot r_{1:N}^\theta(x) \right] - \frac{1}{N} \sum_{j=1}^N \mathbb{E}_{p_j} \left[ \left(\nabla\phi\left(r_{1:N}^\theta(x)\right)\right)_j \right] , \tag{91}$$

where $\mathbf{p} = [p_1, ..., p_N]$.

Let $\phi(r) = \sum_i r_i \log r_i$, where $\nabla \phi(r) = [1 + \log r_1, ..., 1 + \log r_N]$, then

$$
\begin{aligned}
\mathcal{L}_\phi^\mu(f, g) &= \mathbb{E}_{\bar{p}} \left[ -\sum_{j=1}^N \cancel{r_j^\theta(x) \log r_j^\theta(x)} + \sum_{j=1}^N (1 + \cancel{\log r_j^\theta(x)}) r_j^\theta(x) \right] - \frac{1}{N} \sum_{j=1}^N \mathbb{E}_{p_j} \left[ 1 + \log r_j^\theta(x_j) \right] \\
&= \mathbb{E}_{\bar{p}} \left[ \sum_{j=1}^N r_j^\theta(x) \right] - 1 - \frac{1}{N} \sum_{j=1}^N \mathbb{E}_{p_j} \left[ \log r_j^\theta(x_j) \right] \\
&\overset{(i)}{=} -\frac{1}{N} \sum_{j=1}^N \mathbb{E}_{p_j} \left[ \log \frac{p_j^\theta(x_j)}{\sum_i p_i^\theta(x_j)} \right] ,
\end{aligned}
\tag{92}
$$

recovers the multiclass `DiffCLF` loss defined in Equation (10), where $(i)$ is based on $\sum_i r_i^\theta \equiv 1$.

Another possible choices of $\phi(r)$ include $\frac{1}{2} \sum_i r_i^2$ and $\sum_i 1/r_i$. In particular, one could also choose $\phi(r) = \sum_i r_i \log r_i - (1 + r_i) \log(1 + r_i)$ as the binary case, which leads to

$$
\begin{aligned}
\mathcal{L}_\phi^\mu(f, g) &= \mathbb{E}_{\bar{p}} \left[ -\sum_{j=1}^N \left( \cancel{r_j^\theta(x) \log r_j^\theta(x)} - (1 + \cancel{r_j^\theta(x)}) \log(1 + r_j^\theta(x)) \right) \right. \\
&\quad \left. + \sum_{j=1}^N r_j^\theta (\cancel{\log r_j^\theta(x)} - \log(1 + r_j^\theta(x))) \right] - \frac{1}{N} \sum_{j=1}^N \mathbb{E}_{p_j} \left[ \log r_j^\theta(x_j) - \log(1 + r_j^\theta(x_j)) \right] \\
&= \frac{1}{N} \sum_{j=1}^N \mathbb{E}_{p_j} \left[ 2 \log(1 + r_j^\theta(x_j)) - \log r_j^\theta(x_j) \right] \\
&= \frac{1}{N} \sum_{j=1}^N \mathbb{E}_{p_j} \left[ 2 \log \left( 1 + \frac{p_j^\theta(x_j)}{\sum_i p_i^\theta(x_j)} \right) - \log \frac{p_j^\theta(x_j)}{\sum_i p_i^\theta(x_j)} \right] .
\end{aligned}
\tag{93}
$$

## F. Extension to Discrete Diffusions

Recent works (Campbell et al., 2022; Meng et al., 2022; Lou et al., 2024; Campbell et al., 2024; Gat et al., 2024; Shaul et al., 2025) extend diffusion models to discrete state spaces $\mathbb{Y}$ by formulating the forward noising process as a continuous-time Markov chain (CTMC). A CTMC is specified by a family of (possibly time-dependent) transition rate matrices $(Q_t)_t$, a.k.a. *generators*, where for each $t$: (i) $Q_t = (Q_t(y, y'))_{y, y' \in \mathbb{Y}}$, (ii) $Q_t(y, y) = -\sum_{y' \neq y} Q_t(y, y')$ for any $y \in \mathbb{Y}$, and (iii) $Q_t(y, y') \geq 0$ for all $y \neq y' \in \mathbb{Y}$. We write

$$
Y_t \sim \mathrm{CTMC}(Q_t) ,
$$

to mean that $Y_0 \sim p_0$ followed by infinitesimal transitions governed by $Q_t$. As a stochastic process starting from a target distribution $p_0$, $(Q_t)_t$ induces a sequence of intermediate marginals $(p_t)_t$, which satisfy the *Kolmogorov forward equation* (also called the master equation):

$$
\frac{\partial}{\partial t} p_t(y) = \sum_{y'} p_t(y') Q_t(y', y) \iff \partial_t p_t = Q_t^\top p_t .
\tag{94}
$$

Analogous to the Stein score in continuous spaces, one can define the *concrete score* in discrete spaces as the vector of marginal density ratios

$$
S_t(y)_{y'} := \frac{p_t(y')}{p_t(y)} .
$$

In terms of this score, the log-density dynamics admit the compact form

$$
\frac{\partial}{\partial t} \log p_t(y) = \frac{1}{p_t(y)} \sum_{y'} p_t(y') Q_t(y', y) = \sum_{y'} S_t(y)_{y'} Q_t(y', y) .
\tag{95}
$$

Finally, under mild regularity conditions, the *time-reversed process* $(Y_t)_t$ is again a CTMC with generator $(\tilde{Q}_t)_t$ (Kelly, 2011) given by

$$Y_t \sim \text{CTMC}(\tilde{Q}_t), \quad \text{where} \quad \tilde{Q}_t(y, y') = \begin{cases} Q_t(y', y) \frac{p_t(y')}{p_t(y)}, & y \neq y', \\ -\sum_{y' \neq y} \tilde{Q}_t(y, y'), & y = y'. \end{cases} \tag{96}$$

Though the marginals $(p_t)_t$ are intractable, one could train time-dependent neural networks $s_t^\theta(y)$ to minimize the following *Score Entropy* (SE) loss

$$\mathcal{L}_{\text{SE}}(\theta; t) = \mathbb{E}_{p_t} \left[ \sum_{y \neq Y_t \in \mathbb{Y}} Q_t(Y_t, y) \left( S_t(Y_t)_y \log \frac{S_t(Y_t)_y}{s_t^\theta(Y_t)_y} - S_t(Y_t)_y + s_t^\theta(Y_t)_y \right) \right], \tag{97}$$

$$\tag{98}$$

by optimizing the following *Conditional Score Entropy* loss analogous to DSM

$$\mathcal{L}_{\text{CSE}}(\theta; t) = \mathbb{E}_{p_0} \mathbb{E}_{p_{t|0}} \left[ \sum_{y \neq Y_t \in \mathbb{Y}} Q_t(Y_t, y) \left( -S_t(Y_t|Y_0)_y \log s_t^\theta(Y_t)_y + s_t^\theta(Y_t)_y \right) \right], \tag{99}$$

where $S_t(Y_t|Y_0)$ is the *conditional concrete score*, $\nabla_\theta \mathcal{L}_{\text{CSE}}(\theta; t) = \nabla_\theta \mathcal{L}_{\text{SE}}(\theta; t)$, and $p_{t|0}$ is the conditional distribution obtained by solving the following ODE

$$\frac{\partial}{\partial t} p_{t|0}(y|y_0) = \sum_{y'} p_{t|0}(y'|y_0) Q_t(y', y) \quad \text{with } p_{0|0}(y|y_0) = \delta_{y_0}(y). \tag{100}$$

**Energy-based training.** Similar to Equation (2), one could define a family of EBMs on $\mathbb{Y}$ as follows

$$p_t^\theta(y_t) = \exp(-\mathrm{U}_t^\theta(y_t))/\mathcal{Z}_t^\theta, \quad \mathcal{Z}_t^\theta = \exp(\mathrm{F}_t^\theta) = \sum_{y_t \in \mathbb{Y}} \exp(-\mathrm{U}_t^\theta(y_t)). \tag{101}$$

Simply plugging $p_t^\theta$ into Equation (99) we have

$$\mathcal{L}_{\text{CSE}}(\theta; t) = \mathbb{E}_{p_0} \mathbb{E}_{p_{t|0}} \left[ \sum_{y \neq Y_t \in \mathbb{Y}} Q_t(Y_t, y) \left( -S_t(Y_t)_y \log \frac{p_t^\theta(y)}{p_t^\theta(Y_t)} + \frac{p_t^\theta(y)}{p_t^\theta(Y_t)} \right) \right] \tag{102}$$

$$= \mathbb{E}_{p_0} \mathbb{E}_{p_{t|0}} \left[ \sum_{y \neq Y_t \in \mathbb{Y}} Q_t(Y_t, y) \left\{ S_t(Y_t)_y (\mathrm{U}_t^\theta(y) - \mathrm{U}_t^\theta(Y_t)) + \exp(\mathrm{U}_t^\theta(Y_t) - \mathrm{U}_t^\theta(y)) \right\} \right] \tag{103}$$

**Combined with classification loss.** Therefore, it is straightforward to combine the classification loss 10 with Equation (99) to train an energy-based discrete Diffusion model.

## G. Pseudo codes for DiffCLF

In this section, we include pseudo codes for training energy-based generative models with `DiffCLF`, for both the binary case (Algorithm 1) and multi-class case (Algorithm 2). It is noticeable that we will always use a half batch size compared to DSM-only training, to ensure the same number of updates regarding to the DSM objective. In such a case, training with binary `DiffCLF` is only using 50% more computes, compared to DSM-only energy-based training.

---

**Algorithm 1** Training energy-based generative models with binary `DiffCLF`

---

**Input:** $U_t^\theta(y) = U_\theta(t, y) : \mathbb{R}^d \to \mathbb{R}$ the energy network, $F_t^\theta := F_\theta(t) : [0, 1] \to \mathbb{R}$ the free energy network, noising schedules $\alpha_t, \beta_t, \gamma_t, w : [0, 1] \to \mathbb{R}^+$ the weighting function for DSM loss, and a TimeSampler that draws a pair of times.

**while** not converged **do**

  ▷ Draw clean samples

  $x_0 \sim p_0$ for DMs, or $(x_0, x_1) \sim \pi$ for SIs

  ▷ Draw time pairs

  $(t, t') \sim$ TimeSampler

  ▷ Get noisy samples

  $y_t \leftarrow x_t + \gamma_t z, \quad z \sim \mathcal{N}(0, 1), \quad \text{and} \begin{cases} x_t \leftarrow \alpha_t x_0, \text{ for DMs} \\ x_t \leftarrow \alpha_t x_0 + \beta_t x_1, \text{ for SIs} \end{cases}$

  $y_{t'} \leftarrow x_{t'} + \gamma_{t'} z', \quad z' \sim \mathcal{N}(0, 1), \quad \text{and} \begin{cases} x_{t'} \leftarrow \alpha_{t'} x_0, \text{ for DMs} \\ x_{t'} \leftarrow \alpha_{t'} x_0 + \beta_{t'} x_1, \text{ for SIs} \end{cases}$

  ▷ Compute the DSM loss

  $\mathcal{L}_{\text{dsm}}(\theta) \leftarrow \frac{1}{2} w(t) \left\| -\nabla U_t^\theta(y_t) + z/\gamma_t^2 \right\|^2 + \frac{1}{2} w(t') \left\| -\nabla U_{t'}^\theta(y_{t'}) + z'/\gamma_{t'}^2 \right\|^2$

  ▷ Compute the `DiffCLF` loss

  $\text{Softplus}(x) := \log(1 + e^x)$

  $\mathcal{L}_{\text{clf}}(\theta) \leftarrow \frac{1}{2} \text{Softplus}\left( -U_t^\theta(y_t) + F_t^\theta + U_{t'}^\theta(y_{t'}) - F_{t'}^\theta \right) + \frac{1}{2} \text{Softplus}\left( -U_{t'}^\theta(y_{t'}) + F_{t'}^\theta + U_t^\theta(y_{t'}) - F_t^\theta \right)$

  ▷ Optimize parameters $\theta$ by GD

  $\theta \leftarrow \theta - \eta \nabla_\theta (\mathcal{L}_{\text{dsm}} + \mathcal{L}_{\text{clf}})$

**end while**

---

**Algorithm 2** Training energy-based generative models with multi-class `DiffCLF`

---

**Input:** $U_t^\theta(y) = U_\theta(t, y) : \mathbb{R}^d \to \mathbb{R}$ the energy network, $F_t^\theta := F_\theta(t) : [0, 1] \to \mathbb{R}$ the free energy network, noising schedules $\alpha_t, \beta_t, \gamma_t, w : [0, 1] \to \mathbb{R}^+$ the weighting function for DSM loss, and a TimeSampler that draws a pair of times.

**while** not converged **do**

  ▷ Draw clean samples

  $x_0 \sim p_0$ for DMs, or $(x_0, x_1) \sim \pi$ for SIs

  ▷ Draw time pairs

  $t_{1:N} \sim$ TimeSampler

  ▷ Get noisy samples

  **for** $i = 1, ..., N$ **do**

    $y_{t_i} \leftarrow x_{t_i} + \gamma_{t_i} z_i, \quad z_i \sim \mathcal{N}(0, 1), \quad \text{and} \begin{cases} x_{t_i} \leftarrow \alpha_{t_i} x_0, \text{ for DMs} \\ x_{t_i} \leftarrow \alpha_{t_i} x_0 + \beta_{t_i} x_1, \text{ for SIs} \end{cases}$

  **end for**

  ▷ Compute the DSM loss

  $\mathcal{L}_{\text{dsm}}(\theta) \leftarrow \frac{1}{N} \sum_{i=1}^N w(t_i) \left\| -\nabla U_{t_i}^\theta(y_{t_i}) + z_i/\gamma_{t_i}^2 \right\|^2$

  ▷ Compute the `DiffCLF` loss

  $\text{Softmax}(z_1, ..., z_N)[i] = \left( \frac{e^{z_1}}{\sum_{j=1}^N e^{z_j}}, \ldots, \frac{e^{x_N}}{\sum_{j=1}^N e^{z_j}} \right)[i] = \frac{e^{z_i}}{\sum_{j=1}^N e^{z_j}}$

  $\mathcal{L}_{\text{clf}}(\theta) \leftarrow \frac{1}{N} \sum_{i=1}^N \text{Softmax}\left( -U_{t_1}^\theta(y_{t_i}) + F_{t_1}^\theta, \ldots, -U_{t_N}^\theta(y_{t_i}) + F_{t_N}^\theta \right)[i]$

  ▷ Optimize parameters $\theta$ by GD

  $\theta \leftarrow \theta - \eta \nabla_\theta (\mathcal{L}_{\text{dsm}} + \mathcal{L}_{\text{clf}})$

**end while**

---

# H. Experimental details

## H.1. Preconditioning in the DMs cases

Throughout the experiments under the setting of Diffusion Models, we adapt the preconditioning scheme proposed by (Thornton et al., 2025), which recovers the preconditioning proposed by (Karras et al., 2022) in standard score-based DMs through the Tweedie's formula. In particular, for a general noising process $x_t = S(t)x_0 + \gamma(t)z$ and an any arbitrary scalar network $\text{scalar}_t^\theta(y)$, we parameterize the EBM as follows:

$$U_t^\theta(y) = \frac{\|y\|^2}{2\alpha_{\text{in}}^2(t)} - \frac{\sigma_{\text{data}}}{\sigma(t)} \text{scalar}_t^\theta \left( \frac{y - \beta_{\text{in}}(t)}{\alpha_{\text{in}}(t)} \right) - \alpha_t \frac{\sigma^2(t)\mu_{\text{data}}^T y}{\sigma^2(t) + \sigma_{\text{data}}^2} + \frac{1}{2} \frac{\alpha_t^2}{\alpha_{\text{in}}^2(t)} \|\mu_{\text{data}}\|^2 + \frac{d}{2} \log(2\pi\alpha_{\text{in}}^2), \qquad (104)$$

where

$$\lambda(t) = \gamma_t^2 \frac{\sigma^2(t) + \sigma_{\text{data}}^2}{\sigma_{\text{data}}^2}, \quad \alpha_{\text{skip}}(t) = \frac{\sigma_{\text{data}}^2}{s(t)\left(\sigma_{\text{data}}^2 + \sigma^2(t)\right)} \tag{105}$$

$$\alpha_{\text{in}}(t) = \sqrt{\alpha_t^2 \sigma_{\text{data}}^2 + \gamma_t^2}, \quad \beta_{\text{in}}(t) = \alpha_t \mu_{\text{data}} \tag{106}$$

$$\alpha_{\text{out}}(t) = \frac{\sigma(t)\sigma_{\text{data}}}{\sqrt{\sigma^2(t) + \sigma_{\text{data}}^2}}, \quad \beta_{\text{out}}(t) = \left(1 - \frac{\sigma_{\text{data}}^2}{\sigma^2(t) + \sigma_{\text{data}}^2}\right)\mu_{\text{data}}. \tag{107}$$

In most cases, we centerized the training data and therefore $\mu_{\text{data}} \equiv \beta_{\text{in}}(t) \equiv \beta_{\text{out}}(t) \equiv 0$. And specially, this parameterization ensures that if $\text{NN}_t^\theta = 0$, we have $U_t^\theta(y) = -\log\mathcal{N}(y; \beta_{\text{in}}(t), \alpha_{\text{in}}^2(t)I)$. Through the Tweedie's formula:

$$\mathbb{E}[Y_0|Y_t = y] = S(t)^{-1}\left(y + \gamma(t)^2 \nabla \log p_t(y)\right), \tag{108}$$

we recover the denoiser parameterization proposed by (Karras et al., 2022):

$$D_t^\theta(y) = \frac{-\gamma(t)^2 \nabla U_t^\theta(y) + y}{S(t)} = \alpha_{\text{skip}}(t)y + \alpha_{\text{out}}(t)\nabla\text{scalar}_t^\theta\left(\frac{y - \beta_{\text{in}}(t)}{\alpha_{\text{in}}(t)}\right) + \beta_{\text{out}}(t). \tag{109}$$

In addition, to exploit the achitectures that are well known to be effective in standard score-based DMs, we follow (Salimans & Ho, 2021; Guth et al., 2025) to parameterize $\text{scalar}_t^\theta$ by taking inner product of the input $y$ and the output of a vector-valued network $\text{vector}_t^\theta$, i.e.

$$\text{scalar}_t^\theta(y) = y^\top \text{vector}_t^\theta(y). \tag{110}$$

## H.2. Gaussian mixtures and closed form expressions for DMs and SIs

Mixture of Gaussians (MOG) is distribution having the following density

$$\pi(x) = \sum_{n=1}^N w_n \mathcal{N}(x; \mu_n, \Sigma_n).$$

**Diffusion Models case.** In DMs, we require the exact marginal density, which is a convolution of the noising kernel and the target distribution. Assume the noising kernel is $p_{t|0}(y_t|y_0) = \mathcal{N}(y_t; S(t)y_0, \gamma(t)^2 I_d)$, we have

$$p_t(y_t) = \int p_0(y_0)p_{t|0}(y_t|y_0)dy_0 = \sum_{n=1}^N w_n \mathcal{N}(y_t; S(t)\mu_n, S(t)^2\Sigma_n + \gamma(t)^2 I_d), \tag{111}$$

again a MOG. Therefore the marginal density and score are tractable.

**Stochastic Interpolant case.** In SIs, we consider $p_0(y) = \sum_{n=1}^N w_n \mathcal{N}(y; \mu_n, \Sigma_n)$ and $p_1(y) = \sum_{m=1}^M \tilde{w}_m \mathcal{N}(y; \tilde{\mu}_m, \Sigma_m)$ are both MOGs with independent coupling and a linear interpolation $I_t(y_0, y_1) = (1-t)y_0 + ty_1$, therefore

$$p_t(y_t) = \iint p_0(y_0)p_1(y_1)\mathcal{N}(y_t; I_t(y_0, y_1), \gamma(t)^2 I_d)dy_0 dy_1 \tag{112}$$

$$= \sum_n \sum_m w_n \tilde{w}_m \mathcal{N}(y_t; (1-t)\mu_n + t\tilde{\mu}_m, (1-t)^2\Sigma_n + t^2\tilde{\Sigma}_m + \gamma(t)^2 I_d), \tag{113}$$

allowing exact marginal density and score calculation. Moreover, we require the velocity in SIs, which is $\mathbb{E}[\partial_t I_t(Y_0, Y_1)|Y_t = y_t] = \mathbb{E}[Y_1 - Y_0|Y_t = y_t]$. To get the analytical velocity, notice that

$$p(y_0, y_1 \mid y_t) = \sum_{n=1}^N \sum_{m=1}^M \pi_{n,m}(y_t) \mathcal{N}\left(\begin{bmatrix} y_0 \\ y_1 \end{bmatrix}; \begin{bmatrix} \mu_{0|t}^{(n,m)} \\ \mu_{1|t}^{(n,m)} \end{bmatrix}, \Sigma_{|t}^{(n,m)}\right), \tag{114}$$

$$\pi_{n,m}(y_t) = \frac{w_n \tilde{w}_m \mathcal{N}(y_t; \bar{\mu}_{n,m}, S_{n,m})}{\sum_{n'=1}^N \sum_{m'=1}^M w_{n'} \tilde{w}_{m'} \mathcal{N}(y_t; \bar{\mu}_{n',m'}, S_{n',m'})}, \tag{115}$$

with

$$\bar{\mu}_{n,m} = (1-t)\mu_n + t\tilde{\mu}_m \ , \tag{116}$$

$$S_{n,m} = (1-t)^2\Sigma_n + t^2\tilde{\Sigma}_m + \gamma(t)^2 I_d \ , \tag{117}$$

$$\mu_{0|t}^{(n,m)} = \mu_n + (1-t)\Sigma_n S_{n,m}^{-1}(y_t - \bar{\mu}_{n,m}) \ , \tag{118}$$

$$\mu_{1|t}^{(n,m)} = \tilde{\mu}_m + t\tilde{\Sigma}_m S_{n,m}^{-1}(y_t - \bar{\mu}_{n,m}) \ . \tag{119}$$

Therefore, the exact velocity in this case is given by

$$v_t(y_t) = \mathbb{E}[Y_1 - Y_0 \mid Y_t = y_t] = \sum_{n=1}^{N}\sum_{m=1}^{M} \pi_{n,m}(y_t) \left( \mu_{1|t}^{(n,m)} - \mu_{0|t}^{(n,m)} \right) \ . \tag{120}$$

## H.3. Analytical comparison with DSM on MOG

In this section, we provide the experimental setup for Gaussian mixture experiments along with additional results.

### H.3.1. GAUSSIAN MIXTURE DESIGN

We study two types of Gaussian mixtures. The first, introduced in (Midgley et al., 2023), is a widely used benchmark consisting of 40 Gaussians with uniform weights (MOG-40). The means are sampled from $\mathrm{U}([-40, 40]^d)$, and all components share the same covariance $\log(1 + e)\mathrm{I}_d$. The second, taken from (Grenioux et al., 2024; Noble et al., 2025), is a two-component mixture with modes at $-5 \times \mathbf{1}_d$ and $+5 \times \mathbf{1}_d$ (where $\mathbf{1}_d$ denotes the $d$-dimensional vector of ones), covariance $5 \times 10^{-2}, \mathrm{I}_d$, and imbalanced weights $2/3$ and $1/3$. For training, we standardize these distributions (subtracting the mean and dividing by the standard deviation).

### H.3.2. ARCHITECTURE, TRAINING AND EVALUATION DETAILS

We train log-densities using three settings: (i) $\mathcal{L}_{\mathrm{DSM}}$ alone, (ii) a convex combination of $\mathcal{L}_{\mathrm{DSM}}$ with either $\mathcal{L}_{\mathrm{clf}}$ or $\mathcal{L}_{\mathrm{C}t\mathrm{SM}}$. In the latter case, simply summing the two losses generally worked best.

**Diffusion Model.** For DMs, we adopt the Variance Preserving (VP) schedule (Song et al., 2021b) with a linear $\beta$-schedule ending at $\beta_{\max} = 20$. Time is discretized linearly into 512 steps between $10^{-4}$ and $1 - 10^{-4}$. We follow the energy parameterization of (Thornton et al., 2025), use the DSM weighting from (Karras et al., 2022), and implement a 4-layer MLP of width 128 with sinusoidal time embeddings (Song et al., 2021b). The conditional t-SM loss is reweighted by $\gamma^2/\dot{\gamma}^2$ as recommended by (Yu et al., 2025). Models are trained on 60k samples for 500 epochs with DSM only, followed by 500 epochs with the chosen loss combination. We use a batch size of 2048, Adam optimizer with learning rate $10^{-3}$. We average results over two random training seeds. Metrics for sample quality and log-density estimation are computed on 4096 samples. The Fisher devergence and the classification objective are computed on the full time-grid. Sampling is performed using the DDIM denoising kernel (Song et al., 2021a).

**Stochastic Interpolant.** For SIs, we use the linear interpolant $I_t(x_0, x_1) = (1-t)x_0 + tx_1$ and $\gamma : t \mapsto \sqrt{t(1-t)}$ bridging the 40-mode and 2-mode Gaussian mixtures described earlier. Time is discretized into 512 steps between $10^{-3}$ and $1 - 10^{-3}$. The potential is parameterized as

$$\mathrm{U}_{(\theta_1,\theta_2)}(t, x) = x^\top \mathrm{NN}^{\theta_1}(t, x) + \mathrm{NN}^{\theta_2}(t, x) \ ,$$

where $\mathrm{NN}^{\theta_1} : [0, T] \times \mathbb{R}^d \to \mathbb{R}^d$ and $\mathrm{NN}^{\theta_2} : [0, T] \times \mathbb{R}^d \to \mathbb{R}$ are MLPs with depth 4 and width 64 (if $d \leq 32$) or 256 otherwise. Time embeddings follow (Song et al., 2021b). Training proceeds for 10k steps with DSM only, then 50k steps with the chosen objective. We use a batch size of 1024 and a learning rate of $5 \times 10^{-4}$, sampling endpoint distributions at each step. To reduce variance in $\mathcal{L}_{\mathrm{DSM}}$ and $\mathcal{L}_{\mathrm{C}t\mathrm{SM}}$, we apply the antithetic trick (Albergo et al., 2025, Appendix 6.1). Results are averaged over two seeds, and evaluation metrics are computed on 4096 samples. The Fisher devergence and the classification objective are computed on the full time-grid.

*Table 4.* **Comparison of classification and score matching on synthetic Gaussian mixtures**. Mixtures with two modes are trained using the same architecture under DSM as well as conditional time-score matching and our classification objective, averaged over seeds and number of classification levels $N \in 2, 4, 8, 16$ (DSM uses the same number of score evaluations for every $N$). We report the classification loss (10), Fisher divergence, and MMD ($\times 100$) from the denoising SDE (all on 512 time-steps). The classification approach matches DSM in Fisher divergence and MMD, while yielding markedly better consistency in classification loss.

| | $\mathcal{L}_{\text{clf}} + \mathcal{L}_{\text{DSM}}$ | | | $\mathcal{L}_{\text{C}t\text{SM}} + \mathcal{L}_{\text{DSM}}$ | | | $\mathcal{L}_{\text{DSM}}$ | | |
| Dim | $\mathcal{L}_{\text{clf}}$ | FD | MMD | $\mathcal{L}_{\text{clf}}$ | FD | MMD | $\mathcal{L}_{\text{clf}}$ | FD | MMD |
|---|---|---|---|---|---|---|---|---|---|
| 8 | $4.14_{\pm 0.02}$ | $2.48_{\pm 2.34}$ | $6.94_{\pm 0.59}$ | $5.55_{\pm 1.27}$ | $6.78_{\pm 3.29}$ | $20.45_{\pm 8.43}$ | $17.88_{\pm 4.13}$ | $1.21_{\pm 1.14}$ | $5.91_{\pm 0.68}$ |
| 16 | $3.95_{\pm 0.04}$ | $3.47_{\pm 3.15}$ | $8.57_{\pm 1.83}$ | $17.97_{\pm 9.77}$ | $9.15_{\pm 2.82}$ | $22.50_{\pm 6.13}$ | $191.78_{\pm 51.04}$ | $0.83_{\pm 0.74}$ | $7.13_{\pm 0.83}$ |
| 32 | $3.84_{\pm 0.15}$ | $4.86_{\pm 3.87}$ | $11.91_{\pm 1.00}$ | $27.05_{\pm 13.99}$ | $10.54_{\pm 2.99}$ | $28.59_{\pm 2.31}$ | $194.54_{\pm 23.85}$ | $1.04_{\pm 0.89}$ | $8.62_{\pm 1.15}$ |
| 64 | $3.83_{\pm 0.52}$ | $4.39_{\pm 1.77}$ | $15.30_{\pm 2.08}$ | $47.65_{\pm 22.47}$ | $11.57_{\pm 3.48}$ | $27.49_{\pm 1.93}$ | $208.32_{\pm 14.77}$ | $1.16_{\pm 0.89}$ | $10.73_{\pm 1.04}$ |
| 128 | $3.85_{\pm 0.51}$ | $6.86_{\pm 2.26}$ | $17.61_{\pm 2.09}$ | $151.48_{\pm 51.21}$ | $17.25_{\pm 9.12}$ | $30.69_{\pm 3.09}$ | $1521.54_{\pm 538.48}$ | $3.01_{\pm 2.30}$ | $15.48_{\pm 0.57}$ |

### H.3.3. ADDITIONAL RESULTS

In this section, we complete the results of Section 5 with additional metrics and problems.

**Diffusion Model** While Table 4 provides the same comparison as Table 1 but for the bi-modal case, Tables 5 and 7 examine how the number of classification levels affects log-density estimation, whereas Tables 6 and 8 focus on generation quality.

**Stochastic Interpolant** Like Figure 3, Figures 6a to 6d we visualize the determination coefficient $R^2$ between learned log-densities and the ground truth ones across time for each modal, as well as the log-density log-density scatter plots at two ends which should be a perfect diagonal line in optimality. In Tables 10 and 9, we report the quality of the estimated log-densities using the stochastic interpolant model.

### H.4. Composition

In this section, we give the experimental details for the toy composition example from Section 5.

#### H.4.1. DISTRIBUTIONS DETAILS

**Gaussian mixtures** The Gaussian mixtures $p_A$ and $p_B$ in the left part of Figure 5 (top row) are both bi-modal. For $p_A$, the modes are centered at $\mu_1 = (-a, +a)$ and $\mu_2 = (+a, +a)$ with weights 0.3 and 0.7, respectively. For $p_B$, the modes are at $\mu_1 = (-a, -a)$ and $\mu_2 = (+a, -a)$ with weights 0.7 and 0.3. We chose $a = 0.5$. Both mixtures share identical covariances $\Sigma_1 = \Sigma_2 = 0.01 \times \text{I}_2$.

**Rings mixture** The ring distribution is constructed as the product of a uniform distribution on $[0, 2\pi]$ and a Gaussian distribution on the radius with mean $r$ and variance $\sigma^2 = 10^{-2}$ (with $r \gg \sigma$). Applying the polar transformation maps this distribution on $[0, 2\pi] \times \mathbb{R}^+$ into a ring shape in $\mathbb{R}^2$. In the left part of Figure 5 (bottom row), $p_A$ is defined as a mixture of four such rings with radii $r \in \{1, 3, 5\}$, where each ring is assigned a weight proportional to its radius. This weighting makes the rings appear visually balanced in the mixture.

**Uniforms mixture** The distribution used as $p_B$ in the bottom row of Figure 5 is an equilibrated mixture of 4 uniform distributions : $\mathcal{U}([-6.0, -1.6] \times [-1.4, 1.4]), \mathcal{U}([-1.4, 1.4] \times [1.6, 6.0]), \mathcal{U}([1.6, 6.0] \times [-1.4, 1.4])$ and $\mathcal{U}([-1.4, 1.4] \times [-6.0, -1.6])$.

#### H.4.2. MODELS TRAINING

The models follow the energy parameterization of (Thornton et al., 2025), implemented with a 3-layer MLP of width 128 and trained under the variance-preserving noising scheme. The DSM baseline was trained for 500 epochs with a batch size of 4096, while `DiffCLF` was trained for the same number of epochs using the $N = 4$ version of the objective (10) with a batch size of 1024. Both models used a learning rate of $10^{-4}$ and were trained on a dataset of 100k samples. Figures 7 and 8 show samples generated via their respective denoising SDEs, demonstrating that both approaches capture all distributions.

*Table 5.* **Log-density estimation on synthetic 40-mode Gaussian mixtures.** We report classification loss, Fisher divergence, and average Effective Sample Size (ESS). The ESS is computed between the learned and exact log-densities using exact samples, averaged across time levels. Unlike Table 1, this table shows results for varying numbers of classification levels $N$. For fairness, each setting of $N$ uses the same total number of score evaluations as $\mathcal{L}_{DSM}$.

| | | $\mathcal{L}_{clf} + \mathcal{L}_{DSM}$ | | | $\mathcal{L}_{CtSM} + \mathcal{L}_{DSM}$ | | | $\mathcal{L}_{DSM}$ | | |
|---|---|---|---|---|---|---|---|---|---|---|
| Dim | $N$ | $\mathcal{L}_{clf}$ | ESS | FD | $\mathcal{L}_{clf}$ | ESS | FD | $\mathcal{L}_{clf}$ | ESS | FD |
| 8 | 2 | $4.61_{\pm0.04}$ | $89.9\%_{\pm0.9\%}$ | $3.61_{\pm0.25}$ | $5.14_{\pm0.08}$ | $88.2\%_{\pm0.4\%}$ | $5.27_{\pm0.08}$ | $8.80_{\pm0.13}$ | $90.1\%_{\pm1.2\%}$ | $5.45_{\pm0.83}$ |
| 8 | 4 | $4.40_{\pm0.03}$ | $93.2\%_{\pm0.6\%}$ | $2.50_{\pm0.00}$ | $5.48_{\pm0.19}$ | $90.6\%_{\pm0.4\%}$ | $4.69_{\pm0.01}$ | $9.06_{\pm0.05}$ | $90.4\%_{\pm0.1\%}$ | $4.18_{\pm0.20}$ |
| 8 | 8 | $4.32_{\pm0.00}$ | $94.2\%_{\pm0.3\%}$ | $1.20_{\pm0.03}$ | $6.77_{\pm0.48}$ | $91.9\%_{\pm0.4\%}$ | $4.09_{\pm0.00}$ | $9.20_{\pm0.05}$ | $91.6\%_{\pm1.1\%}$ | $3.94_{\pm0.11}$ |
| 8 | 16 | $4.31_{\pm0.00}$ | $96.2\%_{\pm0.3\%}$ | $0.69_{\pm0.01}$ | $7.83_{\pm0.16}$ | $92.8\%_{\pm0.2\%}$ | $3.92_{\pm0.04}$ | $9.68_{\pm0.08}$ | $91.5\%_{\pm0.1\%}$ | $2.78_{\pm0.08}$ |
| 16 | 2 | $4.40_{\pm0.05}$ | $85.8\%_{\pm0.7\%}$ | $4.15_{\pm0.11}$ | $5.17_{\pm0.42}$ | $77.7\%_{\pm0.4\%}$ | $8.77_{\pm0.26}$ | $22.36_{\pm0.55}$ | $84.7\%_{\pm1.1\%}$ | $6.83_{\pm0.71}$ |
| 16 | 4 | $4.22_{\pm0.04}$ | $86.8\%_{\pm1.6\%}$ | $3.37_{\pm0.04}$ | $11.40_{\pm2.85}$ | $78.4\%_{\pm0.4\%}$ | $8.33_{\pm0.17}$ | $21.45_{\pm0.36}$ | $85.8\%_{\pm0.3\%}$ | $7.13_{\pm1.34}$ |
| 16 | 8 | $4.09_{\pm0.00}$ | $88.8\%_{\pm0.1\%}$ | $2.26_{\pm0.12}$ | $10.04_{\pm1.60}$ | $80.8\%_{\pm0.6\%}$ | $7.52_{\pm0.29}$ | $22.93_{\pm0.46}$ | $87.7\%_{\pm0.9\%}$ | $4.64_{\pm0.89}$ |
| 16 | 16 | $4.05_{\pm0.00}$ | $91.5\%_{\pm0.4\%}$ | $1.48_{\pm0.01}$ | $5.48_{\pm0.18}$ | $81.6\%_{\pm0.7\%}$ | $7.11_{\pm0.03}$ | $22.69_{\pm0.61}$ | $88.9\%_{\pm0.0\%}$ | $3.38_{\pm0.14}$ |
| 32 | 2 | $4.41_{\pm0.04}$ | $76.8\%_{\pm1.1\%}$ | $4.70_{\pm0.11}$ | $5.66_{\pm0.11}$ | $67.8\%_{\pm0.1\%}$ | $10.19_{\pm0.10}$ | $73.34_{\pm0.94}$ | $76.4\%_{\pm0.4\%}$ | $4.88_{\pm0.38}$ |
| 32 | 4 | $4.03_{\pm0.07}$ | $77.1\%_{\pm0.4\%}$ | $4.04_{\pm0.03}$ | $6.63_{\pm0.38}$ | $70.7\%_{\pm0.1\%}$ | $9.46_{\pm0.35}$ | $81.87_{\pm3.99}$ | $81.9\%_{\pm0.3\%}$ | $4.43_{\pm0.10}$ |
| 32 | 8 | $3.89_{\pm0.00}$ | $79.7\%_{\pm0.4\%}$ | $3.33_{\pm0.06}$ | $6.46_{\pm0.26}$ | $71.4\%_{\pm0.4\%}$ | $9.64_{\pm0.43}$ | $86.24_{\pm2.31}$ | $80.4\%_{\pm1.3\%}$ | $3.49_{\pm0.24}$ |
| 32 | 16 | $3.83_{\pm0.01}$ | $84.7\%_{\pm0.9\%}$ | $2.65_{\pm0.03}$ | $13.92_{\pm2.81}$ | $72.7\%_{\pm0.4\%}$ | $9.32_{\pm0.06}$ | $98.82_{\pm1.74}$ | $82.9\%_{\pm0.4\%}$ | $2.72_{\pm0.01}$ |
| 64 | 2 | $4.68_{\pm0.34}$ | $66.0\%_{\pm2.9\%}$ | $5.88_{\pm0.13}$ | $21.25_{\pm2.64}$ | $55.9\%_{\pm1.2\%}$ | $10.10_{\pm0.23}$ | $121.14_{\pm12.99}$ | $69.5\%_{\pm2.2\%}$ | $4.86_{\pm0.68}$ |
| 64 | 4 | $4.04_{\pm0.10}$ | $69.0\%_{\pm0.5\%}$ | $5.16_{\pm0.03}$ | $62.11_{\pm47.56}$ | $55.8\%_{\pm0.1\%}$ | $10.42_{\pm0.02}$ | $121.76_{\pm3.21}$ | $71.3\%_{\pm1.1\%}$ | $4.00_{\pm0.09}$ |
| 64 | 8 | $3.70_{\pm0.04}$ | $72.8\%_{\pm0.1\%}$ | $4.42_{\pm0.10}$ | $34.83_{\pm5.43}$ | $56.5\%_{\pm0.1\%}$ | $10.17_{\pm0.03}$ | $152.74_{\pm1.00}$ | $72.6\%_{\pm0.6\%}$ | $3.57_{\pm0.03}$ |
| 64 | 16 | $3.61_{\pm0.01}$ | $72.3\%_{\pm0.4\%}$ | $4.05_{\pm0.03}$ | $85.59_{\pm68.72}$ | $57.8\%_{\pm1.4\%}$ | $10.47_{\pm0.27}$ | $202.18_{\pm4.46}$ | $75.1\%_{\pm1.7\%}$ | $3.28_{\pm0.18}$ |
| 128 | 2 | $5.98_{\pm0.05}$ | $54.1\%_{\pm0.1\%}$ | $7.30_{\pm0.06}$ | $11.80_{\pm2.10}$ | $44.7\%_{\pm0.0\%}$ | $9.33_{\pm0.00}$ | $427.77_{\pm12.04}$ | $60.0\%_{\pm0.3\%}$ | $8.34_{\pm0.43}$ |
| 128 | 4 | $4.52_{\pm0.11}$ | $54.6\%_{\pm0.4\%}$ | $7.39_{\pm0.04}$ | $17.12_{\pm1.23}$ | $44.8\%_{\pm0.0\%}$ | $9.33_{\pm0.01}$ | $383.60_{\pm15.45}$ | $62.6\%_{\pm0.2\%}$ | $6.88_{\pm0.13}$ |
| 128 | 8 | $3.63_{\pm0.07}$ | $55.8\%_{\pm0.3\%}$ | $6.79_{\pm0.01}$ | $54.49_{\pm28.78}$ | $45.0\%_{\pm0.1\%}$ | $9.40_{\pm0.01}$ | $356.38_{\pm39.38}$ | $61.9\%_{\pm0.7\%}$ | $5.83_{\pm0.33}$ |
| 128 | 16 | $3.47_{\pm0.01}$ | $55.0\%_{\pm0.1\%}$ | $6.17_{\pm0.01}$ | $24.84_{\pm9.45}$ | $45.1\%_{\pm0.1\%}$ | $9.51_{\pm0.10}$ | $366.36_{\pm16.04}$ | $62.6\%_{\pm0.4\%}$ | $6.09_{\pm0.05}$ |

### H.4.3. COMPOSITION ALGORITHM DETAILS

A variety of training-free strategies have been proposed to compose diffusion models. Here, we focus on the "AND" operator, defined as the product distribution $p_{A,B} = p_A \times p_B$. The "OR" operator covered in Section 5 is given by $p_{A,B} = (1/2)p_A + (1/2)p_B$. As emphasized in (Du et al., 2023), if $p_A(t, \cdot)$ and $p_B(t, \cdot)$ denote the time-dependent marginals obtained by noising $p_A$ and $p_B$, then their product $p_A(t, \cdot) \times p_B(t, \cdot)$ is not equal to the marginal of the noised operator $p_{A,B}$. Thus, we do not obtain the correct sequence of marginals "for free". Nevertheless, this construction defines a valid interpolation: it recovers $p_{A,B}$ at $t = 0$ and approaches a Gaussian at $t = T$.

Based on this observation, (Du et al., 2023) proposed annealed Langevin sampling over this sequence, i.e., running an MCMC chain at each noise level, with improved performance when denoising steps (via the discretized SDE equation 8 using $\nabla \log p_A(t, \cdot) + \nabla \log p_B(t, \cdot)$) are inserted between MCMC transitions. Building on this, (Thornton et al., 2025) suggested using the discretized denoising dynamics, (lthough formally incorrect, as noted above) as a proposal distribution within a Sequential Monte Carlo (SMC) (Doucet et al., 2001; Del Moral et al., 2006) framework.

In this work, we combine the strengths of both approaches: we apply standard SMC (Algorithm 3) to the sequence $p_k(\cdot) = p_A(t_k, \cdot) \times p_B(t_k, \cdot)$, using the Metropolis-adjusted Langevin algorithm (MALA) (Roberts & Tweedie, 1996) as the transition kernel. This strategy retains the theoretical guarantees of SMC without depending on the incorrect denoising kernel, which we found prone to divergence. Concretely, we run $N = 64$ MALA steps at each level, tuning the step size to maintain a 75% acceptance, and perform adaptive resampling with threshold $\alpha = 30\%$ (Chopin & Papaspiliopoulos, 2020).

### H.5. Boltzmann Generators

In this section, we describe how diffusion models or stochastic interpolants can be integrated into annealed sampling methods to build unbiased estimators. Let $\rho$ denote a simple base distribution (e.g., a Gaussian), and $\pi$ the target distribution, known up to a normalizing constant. Our goal is to compute expectations $\mathbb{E}_\pi[\phi(Y)]$ for a $\pi$-measurable test function $\phi$.

*Table 6.* **Generation quality on 40-mode Gaussian mixtures.** We report Maximum Mean Discrepancy (MMD) (Gretton et al., 2012) (×100), sliced 2-Wasserstein distance (×100), and total variation (TV) distance between mode-weight histograms (as in (Noble et al., 2025)). Results are reported for varying classification levels $N$.

| | | $\mathcal{L}_{\text{clf}} + \mathcal{L}_{\text{DSM}}$ | | | $\mathcal{L}_{\text{C}t\text{SM}} + \mathcal{L}_{\text{DSM}}$ | | | $\mathcal{L}_{\text{DSM}}$ | | |
| Dim | $N$ | MMD | Sliced $W_2$ | TV | MMD | Sliced $W_2$ | TV | MMD | Sliced $W_2$ | TV |
|---|---|---|---|---|---|---|---|---|---|---|
| 8 | 2 | 1.34±0.37 | 6.32±0.28 | 0.09±0.00 | 1.65±0.28 | 7.55±0.61 | 0.15±0.01 | 1.75±0.07 | 7.74±0.42 | 0.13±0.00 |
| 8 | 4 | 0.56±0.56 | 5.88±0.36 | 0.12±0.00 | 1.41±1.09 | 7.12±0.94 | 0.12±0.02 | 0.54±0.54 | 5.70±0.81 | 0.11±0.01 |
| 8 | 8 | 0.87±0.05 | 5.28±0.19 | 0.09±0.02 | 1.07±0.03 | 6.14±0.03 | 0.12±0.01 | 0.46±0.46 | 5.49±0.37 | 0.10±0.01 |
| 8 | 16 | 0.00±0.00 | 4.77±0.68 | 0.09±0.01 | 1.49±0.13 | 6.82±0.54 | 0.11±0.01 | 1.21±0.06 | 6.34±0.02 | 0.11±0.01 |
| 16 | 2 | 0.81±0.31 | 6.15±0.03 | 0.11±0.00 | 2.40±0.21 | 10.74±0.55 | 0.22±0.01 | 1.81±0.18 | 7.50±0.35 | 0.13±0.00 |
| 16 | 4 | 1.15±0.03 | 5.86±0.19 | 0.10±0.01 | 3.18±0.31 | 11.78±0.81 | 0.23±0.02 | 1.33±0.58 | 6.86±0.44 | 0.12±0.00 |
| 16 | 8 | 1.00±0.17 | 5.96±0.41 | 0.10±0.00 | 1.82±0.47 | 8.85±1.12 | 0.21±0.01 | 0.80±0.39 | 5.90±0.68 | 0.12±0.01 |
| 16 | 16 | 0.68±0.41 | 5.93±0.27 | 0.09±0.01 | 2.40±0.36 | 9.39±0.48 | 0.19±0.02 | 1.19±0.49 | 6.30±0.37 | 0.11±0.01 |
| 32 | 2 | 1.67±0.50 | 7.51±0.53 | 0.12±0.01 | 3.11±0.37 | 12.96±0.85 | 0.21±0.05 | 1.66±0.28 | 7.47±0.50 | 0.14±0.01 |
| 32 | 4 | 1.21±0.12 | 6.69±0.29 | 0.11±0.01 | 2.32±0.02 | 10.92±0.18 | 0.17±0.03 | 1.40±0.04 | 6.91±0.09 | 0.12±0.01 |
| 32 | 8 | 0.94±0.22 | 6.41±0.06 | 0.10±0.01 | 2.40±0.04 | 10.90±0.42 | 0.18±0.01 | 1.09±0.22 | 6.22±0.82 | 0.09±0.00 |
| 32 | 16 | 0.97±0.38 | 6.22±0.61 | 0.09±0.01 | 1.86±0.20 | 9.76±0.36 | 0.15±0.01 | 0.67±0.16 | 5.84±0.09 | 0.10±0.01 |
| 64 | 2 | 3.68±0.80 | 10.41±1.11 | 0.15±0.02 | 12.25±5.58 | 30.15±10.58 | 0.52±0.19 | 1.52±0.24 | 7.40±0.09 | 0.14±0.01 |
| 64 | 4 | 2.13±0.18 | 7.88±0.39 | 0.13±0.01 | 10.80±0.89 | 26.86±1.25 | 0.48±0.00 | 1.65±0.02 | 8.11±0.45 | 0.16±0.01 |
| 64 | 8 | 1.69±0.11 | 7.66±0.27 | 0.12±0.01 | 9.12±0.93 | 21.78±1.74 | 0.43±0.04 | 1.41±0.02 | 7.18±0.09 | 0.12±0.00 |
| 64 | 16 | 1.22±0.25 | 6.67±0.38 | 0.10±0.00 | 7.60±2.05 | 18.46±2.06 | 0.34±0.09 | 1.46±0.08 | 6.78±0.59 | 0.13±0.01 |
| 128 | 2 | 5.34±0.27 | 15.18±1.27 | 0.16±0.02 | 5.23±0.47 | 22.31±1.14 | 0.61±0.01 | 2.37±0.19 | 9.41±0.26 | 0.18±0.01 |
| 128 | 4 | 4.27±0.15 | 13.29±0.34 | 0.15±0.01 | 14.73±0.26 | 32.63±0.32 | 0.80±0.05 | 2.02±0.13 | 8.20±0.60 | 0.15±0.00 |
| 128 | 8 | 2.42±0.20 | 10.51±0.07 | 0.12±0.01 | 22.79±1.63 | 47.99±4.90 | 1.11±0.03 | 1.95±0.37 | 7.61±0.72 | 0.14±0.00 |
| 128 | 16 | 2.12±0.25 | 9.72±0.55 | 0.13±0.01 | 22.97±6.80 | 42.97±11.55 | 1.02±0.30 | 1.61±0.01 | 7.25±0.24 | 0.12±0.02 |

**Importance Sampling (IS).** A natural approach is *Importance Sampling (IS)*. Provided $\text{supp}(\rho) \subseteq \text{supp}(\pi)$,

$$\mathbb{E}_\pi[\phi(Y)] = \int \phi(y)\pi(y)\mathrm{d}y = \mathbb{E}_\rho[w(Y)\phi(Y)] \quad w = \pi/\rho \,. \tag{121}$$

This yields the Monte Carlo estimator

$$\mathbb{E}_\pi[\phi(Y)] \approx \frac{1}{N} \sum_{i=1}^N W_i \phi(Y_i), \quad Y_i \sim \rho \,,$$

where $\{Y_i\}_{i=1}^N$ are called particles. When $\pi$ is unnormalized, we use normalized weights $\tilde{W}_i = W_i / \sum_j W_j$, leading to a biased but consistent estimator. The main drawback of IS is variance explosion when $\rho$ poorly overlaps with $\pi$, particularly in high dimensions (Agapiou et al., 2017).

**Annealed Importance Sampling (AIS).** To alleviate mismatch between $\rho$ and $\pi$, (Neal, 2001) introduced *Annealed Importance Sampling* (AIS). AIS extends the problem to a sequence of distributions and defines an augmented target–proposal pair

$$\bar{\pi}(y_{0:K}) = \pi(x_0) \prod_{k=0}^{K-1} q_{k+1|k}(y_{k+1}|y_k), \quad \bar{\rho}(y_{0:K}) = \rho(x_K) \prod_{k=0}^{K-1} q_{k|k+1}(y_k|y_{k+1}) \,, \tag{122}$$

where $\{q_{k+1|k}\}_{k=0}^{K-1}$ and $\{q_{k|k+1}\}_{k=0}^{K-1}$ are forward and backward Markov kernels, often chosen as reversible MCMC kernels (see (Neal, 2001)). Expectations under $\pi$ can then be expressed as expectations under $\bar{\pi}$ and estimated via IS with proposal $\bar{\rho}$

$$\mathbb{E}_\pi[\phi(Y)] = \int \phi(y_0)\bar{\pi}(y_{0:K})\mathrm{d}y_{0:K} \approx \frac{1}{N} \sum_{i=1}^N \bar{W}^i \phi(Y_0^i), \quad (Y_0^i, \ldots, Y_K^i) \sim \bar{\rho} \,. \tag{123}$$

*Table 7.* **Log-density estimation on synthetic 2-mode Gaussian mixtures.** We report classification loss, Fisher divergence, and average Effective Sample Size (ESS). The ESS is computed between the learned and exact log-densities using exact samples, averaged across time levels. Unlike Table 1, this table shows results for varying numbers of classification levels $N$. For fairness, each setting of $N$ uses the same total number of score evaluations as $\mathcal{L}_{\text{DSM}}$.

| | | $\mathcal{L}_{\text{clf}} + \mathcal{L}_{\text{DSM}}$ | | | $\mathcal{L}_{\text{C}t\text{SM}} + \mathcal{L}_{\text{DSM}}$ | | | $\mathcal{L}_{\text{DSM}}$ | | |
|---|---|---|---|---|---|---|---|---|---|---|
| Dim | $N$ | $\mathcal{L}_{\text{clf}}$ | ESS | FD | $\mathcal{L}_{\text{clf}}$ | ESS | FD | $\mathcal{L}_{\text{clf}}$ | ESS | FD |
| 8 | 2 | $4.18_{\pm0.01}$ | $95.7\%_{\pm2.3\%}$ | $2.83_{\pm0.19}$ | $5.05_{\pm0.67}$ | $91.8\%_{\pm0.1\%}$ | $5.02_{\pm0.38}$ | $14.70_{\pm1.88}$ | $91.6\%_{\pm0.9\%}$ | $2.20_{\pm0.73}$ |
| 8 | 4 | $4.13_{\pm0.00}$ | $96.5\%_{\pm0.4\%}$ | $3.70_{\pm0.34}$ | $4.72_{\pm0.49}$ | $83.3\%_{\pm1.5\%}$ | $7.25_{\pm0.15}$ | $14.58_{\pm1.91}$ | $90.9\%_{\pm1.5\%}$ | $1.29_{\pm0.34}$ |
| 8 | 8 | $4.12_{\pm0.00}$ | $98.2\%_{\pm0.0\%}$ | $2.22_{\pm0.18}$ | $6.02_{\pm0.43}$ | $88.0\%_{\pm8.3\%}$ | $5.36_{\pm1.63}$ | $18.21_{\pm1.49}$ | $92.6\%_{\pm1.2\%}$ | $0.91_{\pm0.30}$ |
| 8 | 16 | $4.12_{\pm0.00}$ | $97.5\%_{\pm0.1\%}$ | $1.17_{\pm0.17}$ | $6.39_{\pm1.93}$ | $85.9\%_{\pm1.8\%}$ | $9.50_{\pm2.75}$ | $24.02_{\pm0.36}$ | $92.6\%_{\pm1.0\%}$ | $0.44_{\pm0.00}$ |
| 16 | 2 | $4.00_{\pm0.01}$ | $90.8\%_{\pm4.0\%}$ | $2.93_{\pm0.61}$ | $14.16_{\pm7.75}$ | $66.5\%_{\pm2.3\%}$ | $9.32_{\pm0.11}$ | $157.79_{\pm0.87}$ | $81.3\%_{\pm0.3\%}$ | $1.19_{\pm0.21}$ |
| 16 | 4 | $3.94_{\pm0.03}$ | $95.3\%_{\pm0.5\%}$ | $4.49_{\pm1.22}$ | $12.38_{\pm3.86}$ | $76.9\%_{\pm3.6\%}$ | $8.53_{\pm0.46}$ | $278.78_{\pm1.52}$ | $82.2\%_{\pm1.5\%}$ | $1.08_{\pm0.03}$ |
| 16 | 8 | $3.94_{\pm0.03}$ | $92.7\%_{\pm1.3\%}$ | $3.99_{\pm0.75}$ | $27.12_{\pm5.10}$ | $73.3\%_{\pm9.8\%}$ | $8.74_{\pm1.83}$ | $156.97_{\pm1.03}$ | $82.5\%_{\pm0.5\%}$ | $0.68_{\pm0.08}$ |
| 16 | 16 | $3.91_{\pm0.04}$ | $97.9\%_{\pm0.4\%}$ | $2.44_{\pm0.13}$ | $18.21_{\pm12.30}$ | $59.5\%_{\pm1.5\%}$ | $10.02_{\pm0.01}$ | $173.58_{\pm12.17}$ | $83.7\%_{\pm0.3\%}$ | $0.37_{\pm0.06}$ |
| 32 | 2 | $3.99_{\pm0.17}$ | $83.8\%_{\pm3.9\%}$ | $2.77_{\pm0.45}$ | $45.50_{\pm9.22}$ | $55.3\%_{\pm6.1\%}$ | $10.49_{\pm0.74}$ | $164.35_{\pm12.67}$ | $74.4\%_{\pm0.9\%}$ | $1.81_{\pm0.55}$ |
| 32 | 4 | $3.87_{\pm0.06}$ | $87.2\%_{\pm6.5\%}$ | $4.26_{\pm1.03}$ | $27.16_{\pm10.23}$ | $48.8\%_{\pm0.5\%}$ | $10.84_{\pm0.00}$ | $220.25_{\pm11.54}$ | $75.2\%_{\pm0.2\%}$ | $0.98_{\pm0.14}$ |
| 32 | 8 | $3.70_{\pm0.05}$ | $90.0\%_{\pm1.2\%}$ | $5.10_{\pm0.59}$ | $16.21_{\pm3.15}$ | $53.1\%_{\pm4.5\%}$ | $10.75_{\pm0.63}$ | $185.16_{\pm8.36}$ | $76.3\%_{\pm0.1\%}$ | $0.81_{\pm0.05}$ |
| 32 | 16 | $3.81_{\pm0.08}$ | $87.3\%_{\pm0.8\%}$ | $7.30_{\pm0.60}$ | $19.32_{\pm8.11}$ | $50.5\%_{\pm2.0\%}$ | $10.06_{\pm0.60}$ | $208.39_{\pm7.67}$ | $76.9\%_{\pm0.8\%}$ | $0.54_{\pm0.10}$ |
| 64 | 2 | $3.72_{\pm0.12}$ | $78.8\%_{\pm1.2\%}$ | $4.53_{\pm0.54}$ | $29.66_{\pm11.12}$ | $43.7\%_{\pm0.4\%}$ | $11.07_{\pm0.04}$ | $214.31_{\pm8.38}$ | $67.1\%_{\pm0.5\%}$ | $1.96_{\pm0.31}$ |
| 64 | 4 | $4.29_{\pm0.84}$ | $81.3\%_{\pm0.0\%}$ | $3.83_{\pm0.66}$ | $62.44_{\pm6.10}$ | $45.0\%_{\pm0.9\%}$ | $12.10_{\pm0.42}$ | $210.10_{\pm3.03}$ | $68.2\%_{\pm0.4\%}$ | $1.14_{\pm0.05}$ |
| 64 | 8 | $3.73_{\pm0.21}$ | $80.9\%_{\pm6.9\%}$ | $5.65_{\pm2.41}$ | $43.24_{\pm2.78}$ | $43.7\%_{\pm1.7\%}$ | $11.22_{\pm0.33}$ | $206.46_{\pm25.51}$ | $69.2\%_{\pm0.4\%}$ | $0.84_{\pm0.08}$ |
| 64 | 16 | $3.57_{\pm0.15}$ | $75.6\%_{\pm7.0\%}$ | $3.54_{\pm0.76}$ | $55.28_{\pm35.08}$ | $44.2\%_{\pm0.2\%}$ | $11.87_{\pm0.93}$ | $202.40_{\pm8.05}$ | $70.3\%_{\pm0.6\%}$ | $0.69_{\pm0.02}$ |
| 128 | 2 | $3.79_{\pm0.13}$ | $56.5\%_{\pm0.7\%}$ | $6.99_{\pm1.01}$ | $93.67_{\pm7.28}$ | $37.5\%_{\pm0.1\%}$ | $11.27_{\pm0.25}$ | $2192.68_{\pm397.10}$ | $56.3\%_{\pm0.3\%}$ | $4.27_{\pm0.34}$ |
| 128 | 4 | $4.66_{\pm0.14}$ | $61.5\%_{\pm0.7\%}$ | $5.89_{\pm0.50}$ | $141.52_{\pm40.48}$ | $36.8\%_{\pm0.1\%}$ | $28.36_{\pm13.03}$ | $1637.77_{\pm155.21}$ | $56.5\%_{\pm0.3\%}$ | $3.15_{\pm0.71}$ |
| 128 | 8 | $3.66_{\pm0.14}$ | $57.7\%_{\pm1.6\%}$ | $8.87_{\pm0.54}$ | $195.62_{\pm16.22}$ | $39.6\%_{\pm2.9\%}$ | $14.67_{\pm1.39}$ | $1401.30_{\pm161.64}$ | $56.8\%_{\pm0.1\%}$ | $1.95_{\pm0.06}$ |
| 128 | 16 | $3.30_{\pm0.03}$ | $62.4\%_{\pm0.6\%}$ | $5.71_{\pm0.41}$ | $175.09_{\pm50.85}$ | $37.0\%_{\pm0.7\%}$ | $14.69_{\pm1.52}$ | $854.42_{\pm168.79}$ | $56.8\%_{\pm0.3\%}$ | $2.66_{\pm0.30}$ |

where $\bar{W}^i = \bar{\pi}(Y_{0:K}^i)/\bar{\rho}(Y_{0:K}^i)$. AIS thus interpolates between $\rho$ and $\pi$ by gradually refining proposals through intermediate distributions.

**Sequential Monte Carlo (SMC).** A challenge in AIS is *weight degeneracy*: as $k$ increases, particles sampled from $\bar{\rho}$ may diverge from the marginals of $\bar{\pi}$, especially in high dimensions. *Sequential Monte Carlo* (SMC) (Doucet et al., 2001; Del Moral et al., 2006) addresses this by introducing a sequence of intermediate distributions $p_k$ with tractable unnormalized densities which aim to approximate the marginals of $\bar{\pi}$ (i.e., $p_k(y_k) \approx \int \bar{\pi}(y_{0:K})\mathrm{d}y_{0:k-1}\mathrm{d}y_{k:K}$). SMC proceeds by:

1. running AIS between $\rho$ and $p_k$,

2. resampling particles to match $p_k$ using the previous AIS approximation, and

3. running AIS between $p_k$ and $\pi$.

This resampling step prevents degeneracy and improves stability. While early methods performed resampling at every step, modern implementations use adaptive criteria to trigger resampling only when needed (Chopin & Papaspiliopoulos, 2020). The algorithm is presented in Algorithm 3 in the classic case where the forward and backward kernels are the same $p_k$-reversible MCMC kernel (which simplifies the weights) and the generic one is in Algorithm 4.

**SMC in DMs and SIs.** When dealing with a stochastic process such as Equation (1), a natural construction for annealed methods is to take the conditional distributions of $Y_{t_{k+1}}|Y_{t_k}$ as forward kernels, $Y_{t_k}|Y_{t_{k+1}}$ as backward kernels, and the marginals of $Y_{t_k}$ as intermediate distributions $p_k$, with $Y_{t_0} \sim \pi$ and $Y_{t_K} \sim \rho$. DMs and SIs provide this setup by design, as their dynamics satisfy the endpoint conditions. In the case of DMs, (Zhang et al., 2025) propose using the exact noising kernel as forward kernel and the discretized denoising SDE (8) as the backward kernel in AIS, requiring only the score. Extending this principle to SMC, (Phillips et al., 2024) additionally incorporate approximations of the marginals, precisely the focus of this work. In our experiments (Section 5), we apply the same idea to SIs: the forward and backward kernels

*Table 8.* **Generation quality on 2-mode Gaussian mixtures.** We report Maximum Mean Discrepancy (MMD) (Gretton et al., 2012) (×100), sliced 2-Wasserstein distance (×100), and total variation (TV) distance between mode-weight histograms (as in (Noble et al., 2025)). Results are reported for varying classification levels $N$.

| | | $\mathcal{L}_{\text{clf}} + \mathcal{L}_{\text{DSM}}$ | | | $\mathcal{L}_{\text{C}t\text{SM}} + \mathcal{L}_{\text{DSM}}$ | | | $\mathcal{L}_{\text{DSM}}$ | | |
| Dim | $N$ | MMD | Sliced $W_2$ | TV | MMD | Sliced $W_2$ | TV | MMD | Sliced $W_2$ | TV |
|---|---|---|---|---|---|---|---|---|---|---|
| 8 | 2 | $7.42_{\pm 0.31}$ | $20.82_{\pm 1.74}$ | $0.01_{\pm 0.01}$ | $13.79_{\pm 0.24}$ | $15.87_{\pm 1.78}$ | $0.03_{\pm 0.01}$ | $6.32_{\pm 0.78}$ | $21.37_{\pm 1.77}$ | $0.01_{\pm 0.00}$ |
| 8 | 4 | $7.05_{\pm 0.29}$ | $14.28_{\pm 5.52}$ | $0.03_{\pm 0.00}$ | $21.67_{\pm 5.75}$ | $47.94_{\pm 22.18}$ | $0.11_{\pm 0.10}$ | $6.28_{\pm 0.42}$ | $19.70_{\pm 3.09}$ | $0.02_{\pm 0.01}$ |
| 8 | 8 | $6.31_{\pm 0.60}$ | $19.07_{\pm 2.13}$ | $0.01_{\pm 0.00}$ | $22.03_{\pm 11.45}$ | $48.37_{\pm 30.22}$ | $0.19_{\pm 0.18}$ | $5.84_{\pm 0.49}$ | $25.29_{\pm 6.98}$ | $0.02_{\pm 0.02}$ |
| 8 | 16 | $6.96_{\pm 0.47}$ | $13.43_{\pm 2.67}$ | $0.02_{\pm 0.01}$ | $24.33_{\pm 7.51}$ | $59.01_{\pm 25.17}$ | $0.19_{\pm 0.13}$ | $5.18_{\pm 0.01}$ | $13.89_{\pm 0.77}$ | $0.01_{\pm 0.01}$ |
| 16 | 2 | $9.93_{\pm 2.64}$ | $15.48_{\pm 1.49}$ | $0.02_{\pm 0.00}$ | $21.18_{\pm 7.41}$ | $68.65_{\pm 16.57}$ | $0.18_{\pm 0.06}$ | $7.63_{\pm 0.78}$ | $11.31_{\pm 7.99}$ | $0.00_{\pm 0.00}$ |
| 16 | 4 | $8.56_{\pm 1.09}$ | $22.40_{\pm 0.20}$ | $0.02_{\pm 0.01}$ | $22.40_{\pm 5.68}$ | $46.29_{\pm 5.80}$ | $0.08_{\pm 0.03}$ | $7.69_{\pm 0.43}$ | $22.83_{\pm 6.96}$ | $0.04_{\pm 0.01}$ |
| 16 | 8 | $8.56_{\pm 1.22}$ | $10.92_{\pm 0.28}$ | $0.01_{\pm 0.01}$ | $19.38_{\pm 1.92}$ | $48.88_{\pm 25.33}$ | $0.06_{\pm 0.00}$ | $6.70_{\pm 0.89}$ | $22.80_{\pm 9.18}$ | $0.04_{\pm 0.02}$ |
| 16 | 16 | $7.25_{\pm 0.26}$ | $13.33_{\pm 6.27}$ | $0.01_{\pm 0.00}$ | $27.06_{\pm 5.19}$ | $115.65_{\pm 38.93}$ | $0.31_{\pm 0.15}$ | $6.50_{\pm 0.09}$ | $17.29_{\pm 4.43}$ | $0.03_{\pm 0.01}$ |
| 32 | 2 | $11.98_{\pm 0.45}$ | $18.86_{\pm 4.51}$ | $0.02_{\pm 0.01}$ | $26.50_{\pm 1.32}$ | $76.92_{\pm 6.78}$ | $0.37_{\pm 0.01}$ | $9.99_{\pm 0.06}$ | $28.01_{\pm 0.00}$ | $0.03_{\pm 0.01}$ |
| 32 | 4 | $12.27_{\pm 1.72}$ | $18.98_{\pm 5.11}$ | $0.03_{\pm 0.00}$ | $30.14_{\pm 0.64}$ | $85.84_{\pm 0.70}$ | $0.57_{\pm 0.03}$ | $9.50_{\pm 0.17}$ | $14.84_{\pm 2.00}$ | $0.01_{\pm 0.00}$ |
| 32 | 8 | $11.19_{\pm 0.07}$ | $11.49_{\pm 3.99}$ | $0.03_{\pm 0.01}$ | $28.07_{\pm 3.31}$ | $76.61_{\pm 3.20}$ | $0.54_{\pm 0.09}$ | $7.29_{\pm 0.05}$ | $10.69_{\pm 3.76}$ | $0.00_{\pm 0.00}$ |
| 32 | 16 | $12.22_{\pm 0.30}$ | $9.63_{\pm 3.58}$ | $0.01_{\pm 0.00}$ | $29.68_{\pm 0.20}$ | $85.34_{\pm 2.66}$ | $0.63_{\pm 0.03}$ | $7.71_{\pm 0.15}$ | $26.74_{\pm 4.60}$ | $0.05_{\pm 0.02}$ |
| 64 | 2 | $15.68_{\pm 1.39}$ | $20.10_{\pm 3.14}$ | $0.01_{\pm 0.00}$ | $27.41_{\pm 2.15}$ | $87.37_{\pm 3.11}$ | $0.64_{\pm 0.01}$ | $11.62_{\pm 1.10}$ | $19.19_{\pm 9.18}$ | $0.03_{\pm 0.00}$ |
| 64 | 4 | $15.06_{\pm 1.70}$ | $22.11_{\pm 12.00}$ | $0.03_{\pm 0.03}$ | $27.12_{\pm 1.68}$ | $86.51_{\pm 1.15}$ | $0.51_{\pm 0.03}$ | $11.21_{\pm 0.29}$ | $18.76_{\pm 5.50}$ | $0.00_{\pm 0.00}$ |
| 64 | 8 | $15.39_{\pm 2.67}$ | $21.27_{\pm 3.64}$ | $0.02_{\pm 0.01}$ | $27.09_{\pm 2.34}$ | $91.65_{\pm 4.87}$ | $0.67_{\pm 0.00}$ | $10.44_{\pm 0.86}$ | $18.83_{\pm 0.14}$ | $0.02_{\pm 0.01}$ |
| 64 | 16 | $15.07_{\pm 2.26}$ | $23.33_{\pm 8.53}$ | $0.03_{\pm 0.02}$ | $28.35_{\pm 0.95}$ | $83.80_{\pm 2.62}$ | $0.55_{\pm 0.12}$ | $9.63_{\pm 0.06}$ | $19.02_{\pm 8.67}$ | $0.02_{\pm 0.01}$ |
| 128 | 2 | $17.21_{\pm 0.47}$ | $31.28_{\pm 8.20}$ | $0.01_{\pm 0.00}$ | $29.11_{\pm 0.04}$ | $82.95_{\pm 0.86}$ | $0.67_{\pm 0.00}$ | $15.72_{\pm 0.52}$ | $34.61_{\pm 6.61}$ | $0.04_{\pm 0.02}$ |
| 128 | 4 | $16.78_{\pm 0.69}$ | $32.77_{\pm 0.65}$ | $0.06_{\pm 0.01}$ | $31.45_{\pm 0.19}$ | $82.31_{\pm 6.71}$ | $0.66_{\pm 0.01}$ | $15.84_{\pm 0.57}$ | $15.26_{\pm 2.56}$ | $0.02_{\pm 0.01}$ |
| 128 | 8 | $19.74_{\pm 3.16}$ | $49.72_{\pm 10.95}$ | $0.07_{\pm 0.03}$ | $32.18_{\pm 5.65}$ | $92.22_{\pm 1.62}$ | $0.66_{\pm 0.00}$ | $14.99_{\pm 0.25}$ | $31.94_{\pm 13.94}$ | $0.01_{\pm 0.01}$ |
| 128 | 16 | $16.69_{\pm 0.72}$ | $39.50_{\pm 0.44}$ | $0.05_{\pm 0.01}$ | $30.01_{\pm 0.74}$ | $88.74_{\pm 7.63}$ | $0.64_{\pm 0.02}$ | $15.38_{\pm 0.42}$ | $22.36_{\pm 7.18}$ | $0.02_{\pm 0.02}$ |

are discretizations of SDE (9), both depending only on the score, while the marginal approximations required by SMC are learned either via DSM (7) or `DiffCLF` (10) (with $N = 2$ levels). Precisely, in Algorithm 4, we set $p_k(\cdot) = p_{t_k}^\theta(\cdot)$ and, given $v^\theta$ an approximation of $v$, we denote $b_t^\theta(\cdot) = v_t^\theta(\cdot) - \left[\dot{\gamma}(t)\gamma(t) + g(t)^2/2\right] \nabla \log p_t^\theta(\cdot)$ and set

$$p_{k+1|k}(\cdot|y_k) = \mathcal{N}\left(y_k + (t_{k+1} - t_k)b_{t_k}^\theta(y_k),\ g(t_k)^2(t_{k+1} - t_k)\mathrm{I}_d\right)\ ,$$

$$p_{k|k+1}(\cdot|y_{k+1}) = \mathcal{N}\left(y_{k+1} - (t_{k+1} - t_k)b_{t_{k+1}}^\theta(y_{k+1}),\ g(t_{k+1})^2(t_{k+1} - t_k)\mathrm{I}_d\right)\ ,$$

with $g(\cdot) = 10^{-2}$. We run SMC with $N = 8192$ particles and adaptive resampling.

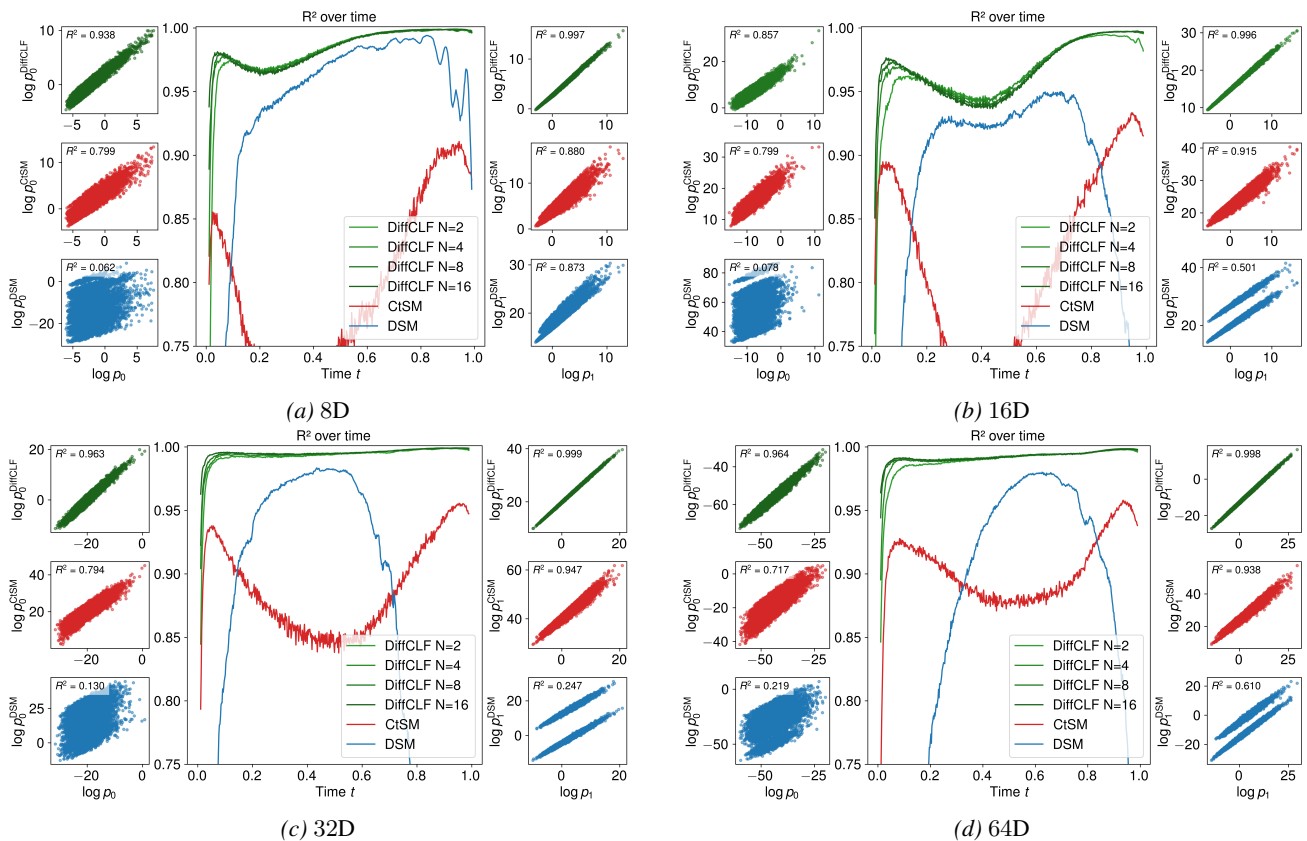

*Figure 6.* $R^2$ **of learned versus exact log-densities for SIs between MoG-40 and MoG-2 across different dimensions.** Complementing Figure 3, which shows detailed 2D scatter plots, this figure demonstrates that `DiffCLF` maintains consistently higher $R^2$ as dimensionality increases.

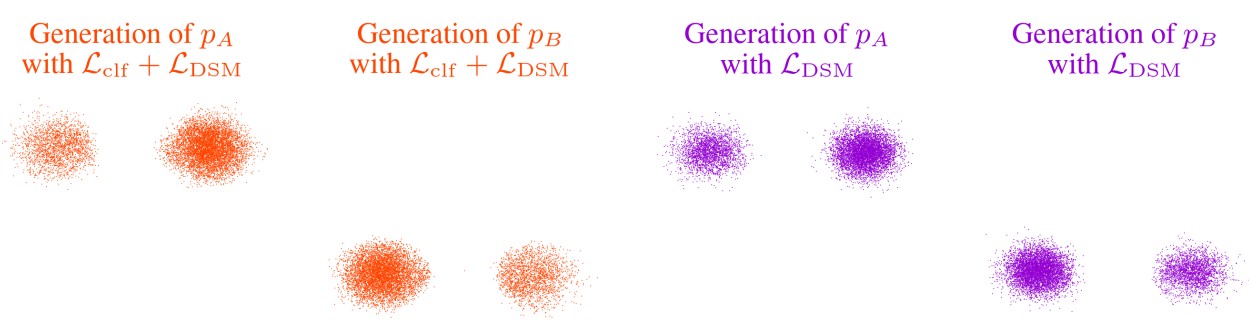

*Figure 7.* **Samples generated by the models trained on the $p_A$ and $p_B$ distribution for the "OR" composition task.** 8192 samples are displayed obtained by discretization of the denoising SDE (8) using the exponential integrator for 512 steps.

### H.5.1. COMPUTATION OF METRICS

The same neural networks used in the analytical comparison with DSM on the MOG benchmarks are employed here, with a single modification: whenever the model is evaluated at $t = 0$, its output is replaced by the exact target density. Regarding the metrics, we report the sliced Wasserstein distance and the sliced Kolmogorov-Smirnov which are defined between two

*Table 9.* **Log-density focused metrics for stochastic interpolants.** We report classification loss, time-average ESS, and Fisher divergence for a SI model learned between MOG-40 and MOG-2. The results present different values of $N$ but the computational budgets remain equal across methods.

| Dim | N | $\mathcal{L}_{\text{clf}} + \mathcal{L}_{\text{DSM}}$ | | | $\mathcal{L}_{\text{C}t\text{SM}} + \mathcal{L}_{\text{DSM}}$ | | | $\mathcal{L}_{\text{DSM}}$ | | |
|---|---|---|---|---|---|---|---|---|---|---|
| | | $\mathcal{L}_{\text{clf}}$ | ESS | FD | $\mathcal{L}_{\text{clf}}$ | ESS | FD | $\mathcal{L}_{\text{clf}}$ | ESS | FD |
| 8 | 2 | 5.26±0.00 | 91.45%±0.02% | 0.16±0.00 | - | - | - | - | - | - |
| 8 | 4 | 5.24±0.00 | 91.78%±0.08% | 0.16±0.00 | - | - | - | - | - | - |
| 8 | 8 | 5.23±0.00 | 91.83%±0.07% | 0.16±0.00 | - | - | - | - | - | - |
| 8 | 16 | 5.23±0.00 | 91.96%±0.20% | 0.16±0.00 | 6.54±0.01 | 5.10%±0.11% | 0.96±0.00 | 6.85±0.01 | 76.91%±0.07% | 0.39±0.00 |
| 16 | 2 | 4.87±0.00 | 68.08%±0.14% | 0.22±0.00 | - | - | - | - | - | - |
| 16 | 4 | 4.86±0.00 | 70.26%±0.10% | 0.23±0.00 | - | - | - | - | - | - |
| 16 | 8 | 4.85±0.00 | 69.73%±0.14% | 0.23±0.00 | - | - | - | - | - | - |
| 16 | 16 | 4.85±0.00 | 69.44%±0.13% | 0.23±0.00 | 8.97±0.01 | 2.74%±0.01% | 1.04±0.00 | 7.33±0.00 | 46.93%±0.01% | 0.46±0.00 |
| 32 | 2 | 4.47±0.00 | 88.76%±0.02% | 0.06±0.00 | - | - | - | - | - | - |
| 32 | 4 | 4.46±0.00 | 90.52%±0.12% | 0.05±0.00 | - | - | - | - | - | - |
| 32 | 8 | 4.46±0.00 | 91.36%±0.08% | 0.05±0.00 | - | - | - | - | - | - |
| 32 | 16 | 4.46±0.00 | 91.87%±0.01% | 0.05±0.00 | 11.62±0.02 | 1.35%±0.16% | 0.61±0.00 | 5.75±0.00 | 48.72%±0.10% | 0.30±0.00 |
| 64 | 2 | 4.12±0.00 | 71.00%±0.04% | 0.08±0.00 | - | - | - | - | - | - |
| 64 | 4 | 4.12±0.00 | 75.26%±0.26% | 0.07±0.00 | - | - | - | - | - | - |
| 64 | 8 | 4.12±0.00 | 74.01%±0.06% | 0.07±0.00 | - | - | - | - | - | - |
| 64 | 16 | 4.12±0.00 | 74.42%±0.34% | 0.07±0.00 | 16.52±0.05 | 0.76%±0.03% | 0.43±0.00 | 5.67±0.02 | 28.38%±0.14% | 0.31±0.00 |
| 128 | 2 | 3.81±0.00 | 54.51%±0.09% | 0.08±0.00 | - | - | - | - | - | - |
| 128 | 4 | 3.79±0.00 | 47.37%±0.02% | 0.07±0.00 | - | - | - | - | - | - |
| 128 | 8 | 3.78±0.00 | 50.34%±0.07% | 0.07±0.00 | - | - | - | - | - | - |
| 128 | 16 | 3.78±0.00 | 52.65%±0.22% | 0.06±0.00 | 21.33±0.00 | 0.23%±0.00% | 0.38±0.00 | 67.03±0.12 | 12.35%±0.07% | 0.32±0.00 |

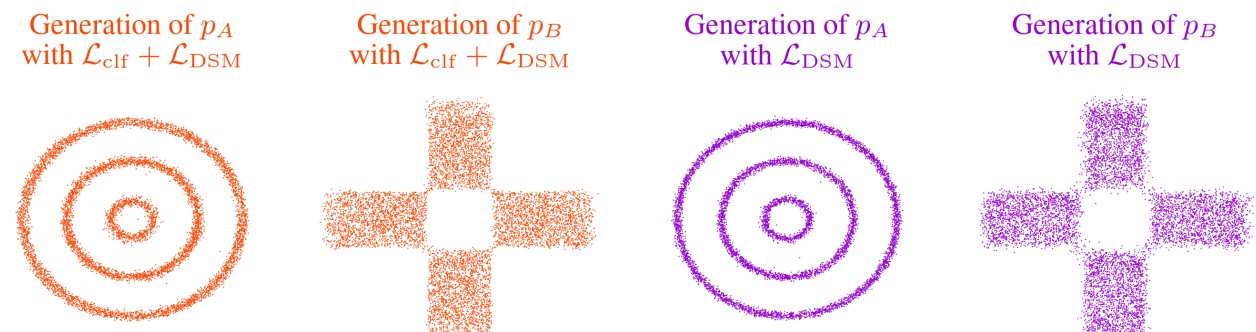

Generation of $p_A$ with $\mathcal{L}_{\text{clf}} + \mathcal{L}_{\text{DSM}}$ — Generation of $p_B$ with $\mathcal{L}_{\text{clf}} + \mathcal{L}_{\text{DSM}}$ — Generation of $p_A$ with $\mathcal{L}_{\text{DSM}}$ — Generation of $p_B$ with $\mathcal{L}_{\text{DSM}}$

*Figure 8.* **Samples generated by the models trained on the $p_A$ and $p_B$ distribution for the "AND" composition task.** 8192 samples are displayed obtained by discretization of the denoising SDE (8) using the exponential integrator for 512 steps.

distributions $p$ and $q$ as

$$\text{s-}W_2(p, q) = \sqrt{\int_{\mathbb{S}^{d-1}} \int_0^1 \left| F_{p_\theta}^{-1}(s) - F_{q_\theta}^{-1}(s) \right| \mathrm{d}s \mathrm{d}\theta} \, ,$$

$$\text{s-KS}(p, q) = \int_{\mathbb{S}^{d-1}} \sup_{s \in \mathbb{R}} \left| F_{p_\theta}(s) - F_{q_\theta}(s) \right| \mathrm{d}s \mathrm{d}\theta \, ,$$

where $p_\theta$ (respectively $q_\theta$) is the push-forward of $p$ (respectively $q$) through $x \mapsto x^T \theta$ and $F.$ indicates the cumulative distribution function. Those distances can be seen as functions of expectations with respect to $p$ and $q$ as by definition, for any distribution $\mu$, $F_\mu(x) = \mathbb{E}_\mu[\mathbf{1}_{Y \leq x}]$.

In the BG experiments, we approximate the distances between a target distribution $\pi$ and itself by setting $F_p = F_\pi$ and

*Table 10.* **Averaged log-density metrics for stochastic interpolants.** We report classification loss and Fisher divergence across different dimensions. The different computation budgets of Table 9 are here averaged.

| Dim | $\mathcal{L}_{\mathrm{clf}} + \mathcal{L}_{\mathrm{DSM}}$ | | $\mathcal{L}_{\mathrm{C}t\mathrm{SM}} + \mathcal{L}_{\mathrm{DSM}}$ | | $\mathcal{L}_{\mathrm{DSM}}$ | |
|---|---|---|---|---|---|---|
| | Classif | Fisher | Classif | Fisher | Classif | Fisher |
| 8 | $5.35_{\pm 0.07}$ | $4.38_{\pm 0.77}$ | $8.59_{\pm 0.16}$ | $40.69_{\pm 1.88}$ | $8.03_{\pm 0.44}$ | $0.37_{\pm 0.06}$ |
| 16 | $4.88_{\pm 0.02}$ | $5.21_{\pm 0.93}$ | $16.10_{\pm 0.46}$ | $42.04_{\pm 2.17}$ | $10.15_{\pm 0.82}$ | $0.52_{\pm 0.09}$ |
| 32 | $4.67_{\pm 0.10}$ | $5.63_{\pm 0.97}$ | $30.88_{\pm 2.49}$ | $40.60_{\pm 2.58}$ | $17.69_{\pm 2.56}$ | $0.53_{\pm 0.08}$ |
| 64 | $4.17_{\pm 0.02}$ | $2.71_{\pm 0.51}$ | $56.54_{\pm 2.91}$ | $35.63_{\pm 2.57}$ | $9.36_{\pm 2.82}$ | $0.23_{\pm 0.04}$ |
| 128 | $3.84_{\pm 0.03}$ | $2.66_{\pm 0.44}$ | $128.28_{\pm 8.23}$ | $279.37_{\pm 3.91}$ | $4.54_{\pm 0.28}$ | $0.24_{\pm 0.04}$ |

---

**Algorithm 3** Sequential Monte Carlo (SMC) algorithm with adaptive resampling

---

**Input:** Sequence of target distributions $\{p_k\}_{k=0}^{K}$, number of MCMC steps per level $L \geq 1$, number of particles $N \geq 1$, ESS resampling threshold $\alpha \in (0, 1]$

▷ Initialization

Sample $Y_K^{1:N} \overset{\mathrm{iid}}{\sim} p_K$

Set $\tilde{W}_K^{1:N} \leftarrow 1/N$

**for** $k = K - 1$ **to** $0$ **do**

  ▷ 1. Compute the SMC importance weights (in parallel, for $n = 1, \ldots, N$)

  $\tilde{W}_k^n \leftarrow (p_k(Y_{k+1}^n)/p_{k+1}(Y_{k+1}^n))\, \tilde{W}_{k+1}^n$

  ▷ 2. Compute the accumulated ESS

  $\mathrm{ESS}_k \leftarrow \left(\sum_{n=1}^{N} \tilde{W}_k^n\right)^2 / \left(\sum_{n=1}^{N} (\tilde{W}_k^n)^2\right)$

  ▷ 3. Resample $X_k^{1:N}$ if the ESS is too low

  **if** $\mathrm{ESS}_{k+1} < \alpha N$ **then**

    ▷ Normalize the importance weights

    $W_k^n \leftarrow \tilde{W}_k^n / \sum_{j=1}^{N} \tilde{W}_k^j$

    ▷ Resample the particles

    Sample $I^{1:N} \sim \mathcal{M}(W_k^1, \ldots, W_k^N)$

    Set $X_k^{1:N} \leftarrow Y_{k+1}^{I^{1:N}}$

    Set $\tilde{W}_{k+1}^{1:N} \leftarrow 1/N$

  **else**

    Set $X_k^n \leftarrow Y_{k+1}^n$

  **end if**

  ▷ 4. Sample from $p_k$ by starting from sample $X_k^n$ (in parallel, for $n = 1, \ldots, N$)

  $\tilde{Y}_k^{1:L} \leftarrow \mathrm{MCMC}(p_k, L, \delta_{X_k^n})$

  $Y_k^n \leftarrow \tilde{Y}_k^L$

**end for**

**Output:** Sequence of samples $Y_{0:K}^{1:N}$ such that $Y_k^{1:N} \overset{\mathrm{iid}}{\sim} p_k$ (approximately)

---

replacing $\mathrm{F}_q$ by a Monte-Carlo approximation of $\mathrm{F}_\pi$.

## H.6. Free energy estimation

In this section, we provide addtional background on free energy difference estimation and *Thermodynamics Integration* (TI) based on (Máté et al., 2025), and the experimental details for free energy estimation.

### H.6.1. FREE ENERGY DIFFERENCE ESTIMATION WITH THERMODYNAMICS INTEGRATION

As illustrated in Appendix B.2, we aim to estimate the free energy difference between two states A and B, which are associated with energy/potential functions $U_A, U_B$ respectively. The free energy difference between these two states is the difference between the log partition functions

$$\Delta \mathrm{F}_{AB} = \mathrm{F}_A - \mathrm{F}_B = \log \frac{Z_B}{Z_A}.$$

---

**Algorithm 4** Extension of the Sequential Monte Carlo algorithm for DMs and SIs

---

**Input:** Sequence of target distributions $\{p_k\}_{k=0}^{K}$, forward kernels $\{q_{k+1|k}\}_{k=0}^{K-1}$, backward kernels $\{q_{k|k+1}\}_{k=0}^{K-1}$, number of MCMC steps per level $L \geq 1$, number of particles $N \geq 1$, ESS resampling threshold $\alpha \in (0, 1]$

▷ Initialization

Sample $Y_K^{1:N} \overset{\text{iid}}{\sim} p_K$

Set $\tilde{W}_K^{1:N} \leftarrow 1/N$

**for** $k = K - 1$ **to** 0 **do**

  ▷ 1. Move to the next distribution with backward kernel (in parallel for $n = 1, \ldots, N$)

  Sample $\tilde{X}_k^n \sim q_{k|k+1}(\cdot \mid Y_{k+1}^n)$

  ▷ 2. Compute the SMC importance weights (in parallel for $n = 1, \ldots, N$)

  $\tilde{W}_k^n \leftarrow \dfrac{p_k(\tilde{X}_k^n)\, p_{k+1|k}(Y_{k+1}^n \mid \tilde{X}_k^n)}{p_{k+1}(Y_{k+1}^n)\, q_{k|k+1}(\tilde{X}_k^n \mid Y_{k+1}^n)} \tilde{W}_{k+1}^n$

  ▷ 3. Compute the accumulated ESS

  $\text{ESS}_k \leftarrow \left(\sum_{n=1}^{N} \tilde{W}_k^n\right)^2 / \left(\sum_{n=1}^{N} (\tilde{W}_k^n)^2\right)$

  ▷ 4. Resample if the ESS is too low

  **if** $\text{ESS}_{k+1} < \alpha N$ **then**

    ▷ Normalize the importance weights

    $W_k^n \leftarrow \tilde{W}_k^n / \sum_{j=1}^{N} \tilde{W}_k^j$

    ▷ Resample the particles

    Sample $I^{1:N} \sim \mathcal{M}(W_k^1, \ldots, W_k^N)$

    Set $X_k^{1:N} \leftarrow Y_{k+1}^{I^{1:N}}$

    Set $\tilde{W}_{k+1}^{1:N} \leftarrow 1/N$

  **else**

    Set $X_k^n \leftarrow Y_{k+1}^n$

  **end if**

  ▷ 5. Sample from $p_k$ using MCMC (in parallel for $n = 1, \ldots, N$)

  $\tilde{Y}_k^{1:L} \leftarrow \text{MCMC}(p_k, L, \delta_{X_k^n})$

  $Y_k^n \leftarrow \tilde{Y}_k^L$

**end for**

**Output:** Sequence of samples $Y_{0:K}^{1:N}$ such that $X_k^{1:N} \overset{\text{iid}}{\sim} p_k$ (approximately)

---

TI leverages a sequence of intermediate distributions $(p_t)_{t \in [0,1]}$ where $p_0 = p_A$ and $p_1 = p_B$ to estimate the free energy difference via

$$
\begin{aligned}
\Delta F_{AB} &= \int_0^1 \partial_t \log Z_t \, dt \\
&= \int_0^1 \frac{\partial_t Z_t}{Z_t} \, dt \\
&= \int_0^1 \frac{\partial_t \int \exp(-U_t(x)) dx}{Z_t} \, dt \\
&= \int_0^1 \frac{-\int \exp(-U_t(x)) \partial_t U_t(x) dx}{Z_t} \, dt \\
&= -\int_0^1 \mathbb{E}_{p_t} \left[ \partial_t U_t(x) \right] dt.
\end{aligned}
\tag{124}
$$

Furthermore, (Máté et al., 2025) proposes the define the intermediate distributions via SIs and using $U_t^\theta$. In such case, the estimation becomes

$$
\Delta F_{AB} = -\int_0^1 \mathbb{E}_{p_t^\theta} \left[ \partial_t U_t^\theta(x) \right] dt,
$$

which has two issues:

(1) Difficult to draw samples from $p_t^\theta$,

(2) the boundaries are not match, i.e. $U_0^\theta \neq U_A$ and $U_1^\theta \neq U_B$.

To alleviate the first issue, one could approximate $\mathbb{E}_{p_t^\theta}$ with $\mathbb{E}_{p_t}$. To fix the second issue, one could estimate the free energy difference between $U_A$ and $U_0^\theta$ ($\Delta F_{U_A,U_0^\theta}$) and $U_B$ and $U_1^\theta$ ($\Delta F_{U_1^\theta,U_B}$) via FEP (Equation (45)) described in Appendix B.2. Hence, the final estimation of $\Delta F_{AB}$ is given by:

$$\Delta F_{AB} = \Delta F_{U_A,U_0^\theta} \underbrace{- \int_0^1 \mathbb{E}_{p_t^\theta}\left[\partial_t U_t^\theta(x)\right] dt}_{\Delta F_{U_0^\theta,U_1^\theta}} + \Delta F_{U_1^\theta,U_B}. \tag{125}$$

### H.6.2. Experimental details

**Alanine dipeptide details.** Both ALDP-imp and ALDP-vac are defined with AMBER ff96 classical force field, under temperature $300K$. For ALDP-imp, we use the samples generated from (Midgley et al., 2023), which contains $1,000,000$ samples and we subsample $250,000$ of them as dataset. To gather equilibrium samples from ALDP-vac, we simulate MD with `openmmtools` (Chodera et al., 2025) to generate $250,000$ samples, where we follow hyperparameter choices as (Du et al., 2025, Appendix D.1.3).

**Stochastic Interpolant.** For SIs, we bridge ALDP-imp ($p_0$) and ALDP-vac ($p_1$) with the linear interpolant $I_t(x_0, x_1) = (1-t)x_0 + tx_1$ and $\gamma : t \mapsto \sqrt{0.01t(1-t)}$. Time is discretized into 512 steps between $10^{-3}$ and $1 - 10^{-3}$. The potential is parameterized as

$$U_\theta(t,x) = x^\top \text{NN}_\theta(t,x),$$

where $\text{NN}_\theta : [0,T] \times \mathbb{R}^d \to \mathbb{R}^d$ is an EGNN with depth 5, width 128, sinusoidal time embeddings, learnable bond embeddings, and learnable atom-type embeddings. Time embeddings follow (Song et al., 2021b). Training proceeds for 500 epochs with DSM only, then 500 epochs with the chosen objective. We use a batch size of 512 and a learning rate of $5 \times 10^{-4}$, sampling endpoint distributions at each step. To reduce variance in $\mathcal{L}_{\text{DSM}}$, we apply the antithetic trick (Albergo et al., 2025).

**Architecture, training and evaluation details.** We use exactly the same network architecture as illustrated in (Du et al., 2025). To enhance the energy learning, we follow (Máté et al., 2025; Du et al., 2025) to apply *Target Score Matching* (TSM) (Bortoli et al., 2024),

$$\mathcal{L}_{\text{TSM}}^{\text{imp}}(\theta) = \mathbb{E}_{t \sim U(0,0.5)} \mathbb{E}_{y_0,y_1} \mathbb{E}_{y_t|y_0,y_1} \left[\left\|\nabla \log p_t^\theta(y_t) - U_{\text{imp}}(y_0)\right\|^2\right], \tag{126}$$

$$\mathcal{L}_{\text{TSM}}^{\text{vac}}(\theta) = \mathbb{E}_{t \sim U(0.5,1)} \mathbb{E}_{y_0,y_1} \mathbb{E}_{y_t|y_0,y_1} \left[\left\|\nabla \log p_t^\theta(y_t) - U_{\text{vac}}(y_1)\right\|^2\right], \tag{127}$$

where $U_{\text{imp}}$ and $U_{\text{vac}}$ are the energy function for the implicit-solvent and vacuum defined with AMBER ff96 classical force field respectively. (Máté et al., 2025) uses

$$\mathcal{L}_{\text{base}} := \mathcal{L}_{\text{dsm}} + \mathcal{L}_{\text{TSM}}^{\text{imp}} + \mathcal{L}_{\text{TSM}}^{\text{vac}}$$

to train energy-based models for TI. To evaluate our method, we simply add the diffusive classification loss to the aforementioned base loss, i.e.

$$\mathcal{L}_{\text{base}} + \mathcal{L}_{\text{clf}},$$

where we use $N = 4$ classes in $\mathcal{L}_{\text{clf}}$

**Estimation details.** We use Equation (125) to estimate the free energy difference. In particular, we draw 5000 samples from ALDP-imp and 5000 samples from ALDP-vac, and use FEP (Equation (45)) to estimate $\Delta F_{U_{\text{imp}},U_0^\theta}$ and $\Delta F_{U_1^\theta,U_{\text{vac}}}$. To estimate $\Delta F_{U_0^\theta,U_1^\theta}$, we uniformly discretize the time from 0 to 1 with 1000 steps and draw 5000 samples from each marginal distributions $p_t$.

### H.7. Additional results on molecular systems

In this section, we present additional experiments on training energy-based diffusion models $(p_t^\theta)_t$ for molecular systems, including the Müller-Brown potential, Alanine Dipeptide (ALDP), and Chignolin, by training only with samples from the equilibrium distribution and without any access to their energies or forces. We simply compare classic DSM training and our method, i.e. jointly training with DSM and the diffusive classification loss. To evaluate the trained model, we simulate molecular dynamics using only $p_{t=0}^\theta$ and compare the simulated samples.

*Table 11.* Quantitative results for ALDP and Chignolin systems. Both FPE and `DiffCLF` refer to jointly training with DSM. Results of DSM and FPE are from (Plainer et al., 2025). We report the metric with mean ± std under 3 random seeds.

| System | Method | IID JS | Langevin JS | IID PMF | Langevin PMF | Train time (GPU hrs) |
|---|---|---|---|---|---|---|
| ALDP | DSM | $0.0081_{\pm 0.0003}$ | $0.0695_{\pm 0.0517}$ | $0.095_{\pm 0.003}$ | $1.047_{\pm 0.924}$ | 3.3 |
| | FPE | $0.0082_{\pm 0.0002}$ | $0.0090_{\pm 0.0006}$ | $0.098_{\pm 0.003}$ | $0.104_{\pm 0.004}$ | 8.1 |
| | `DiffCLF` | $0.0068_{\pm 0.0001}$ | $0.0092_{\pm 0.0002}$ | $0.070_{\pm 0.002}$ | $0.094_{\pm 0.001}$ | 5.6 |
| Chignolin | DSM | $0.0036_{\pm 0.0001}$ | $0.4351_{\pm 0.0141}$ | $0.027_{\pm 0.000}$ | $63.804_{\pm 0.372}$ | 8.7 |
| | FPE | $0.0048_{\pm 0.0001}$ | $0.0050_{\pm 0.0001}$ | $0.037_{\pm 0.000}$ | $0.039_{\pm 0.001}$ | 49.6 |
| | `DiffCLF` | $0.0073_{\pm 0.0019}$ | $0.0181_{\pm 0.0055}$ | $0.060_{\pm 0.015}$ | $0.162_{\pm 0.053}$ | 18.9 |

**Setup.** We follow exactly the same settings of (Plainer et al., 2025), for both dataset processing, network architecture, and hyperparameters. Details of the three different systems could be found in the Appendix B in (Plainer et al., 2025). For simulations based on learn models $p_{t=0}^\theta$, we follow (Plainer et al., 2025) to use Langevin dynamics and use exactly the same hyperparameters. In particular, both ALDP and Chigolin in this setup are coarse-grained, resulting in 5 and 10 residuals respectively (*i.e.* $d = 15$ and $d = 30$).

**Results.** We train energy-based Diffusion models for the aforementioned three different molecular systems, with DSM only ($\mathcal{L}_{dsm}$) and `DiffCLF` ($\mathcal{L}_{dsm} + \mathcal{L}_{clf}$), where we use $N = 2$ for $\mathcal{L}_{clf}$. Figure 9 demonstrates the effectiveness of the proposed diffusive classification loss, which help learning better energy at $t = 0$ while not bringing degeneracy to diffusion sampling without degenarating the scores. Table 11 shows the quantitative results, where the Jensen-Shannon divergence (JS), and the potential of mean force (PMF) error. We additionally report the training time in GPU hours, estimated from a partial training in a single `NVIDIA A100` GPU. The results demonstrate that `DiffCLF` can achieve comparable results to the FPE regularization, while being much faster to train.

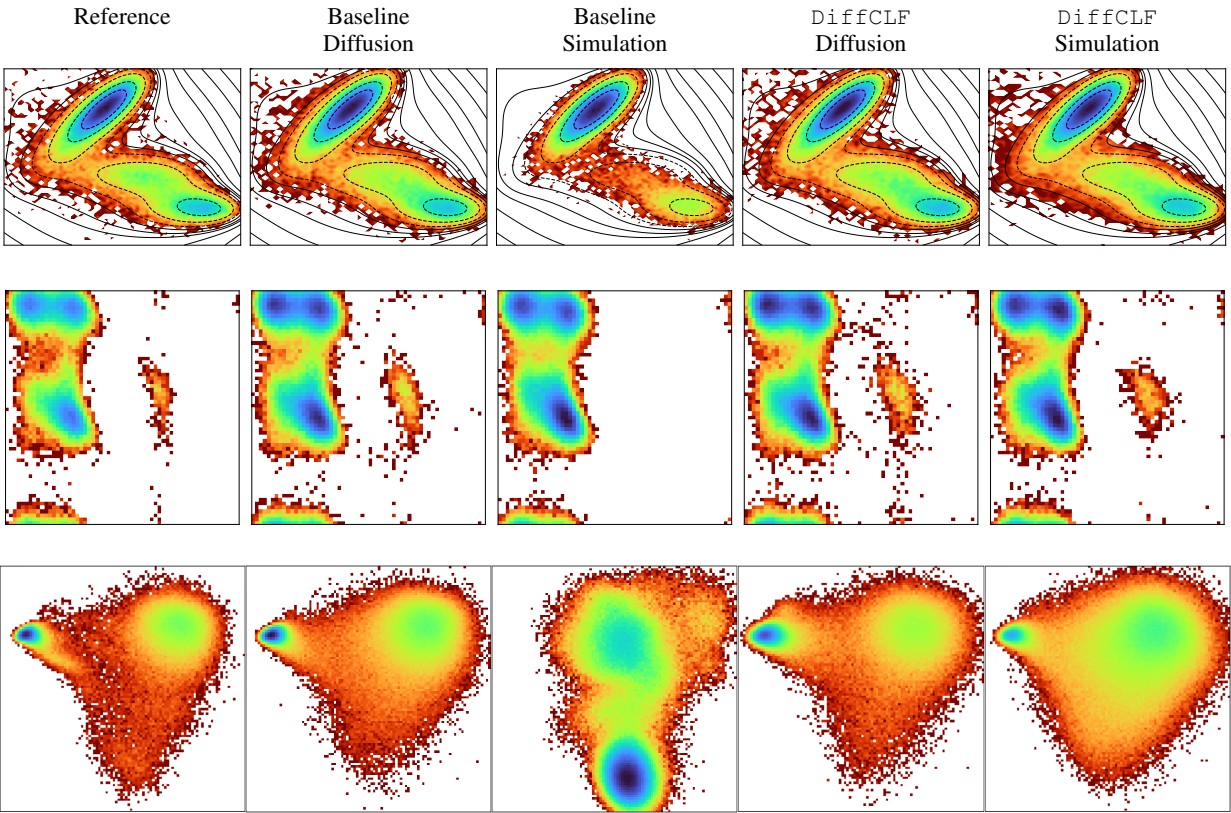

*Figure 9.* Comparison of samples drawn via *Diffusion sampling* (Diffusion) and *Langevin dynamics* (Simulation) by solely using the trained model $\mathring{U}^\theta_{t=0}$. We compare DSM training only (Baseline) and joint training with the diffusive classification loss with 4 classes (DiffCLF). **(Top)**: Müller-Brown potential, visualization of the 2D histogram of the samples; **(Middle)**: Alanine Dipeptide, visualization of the 2D histogram of the torsion angles (x-axis: $\phi$, y-axis: $\psi$); **(Bottom)**: Chignolin, visualization of the 2D histogram of the first two TIC coordinates.

