# OpenReview forum: "A Diffusive Classification Loss for Learning Energy-based Generative Models"
_ICML.cc/2026/Conference — ICML 2026 regular_

### Official Review · Reviewer_WkeT · 2026-03-02

**Soundness:** 3
**Presentation:** 3
**Significance:** 2
**Originality:** 2
**Overall Recommendation:** 4
**Confidence:** 3

**Summary:**

Classical  denoising score matching  (DSM) is effective for sample generation (e.g.  images) by learning the score function $\nabla_x \log p_t(x)$ and sampling via the reverse-time SDE/ODE. However, DSM does not naturally provide accurate  log-densities/energies  $\log p_t(x)$, which helps differentiate distinct modes and can be useful for downstream applications that require density values (e.g. certain forms of model composition or SMC-style weighting).

This paper proposes Diffusive Classification Loss (DiffCLF), an additional training objective that can be combined with DSM to learn energy/log-density values $\log p_t(x)$ (not only score gradients). The key idea reframes log-density estimation as a supervised classification problem over different time levels: by classifying which time level a sample was drawn from, the method effectively learns density ratios between time marginals, which can be converted into consistent log-density/energy information. The paper provides theoretical results on consistency at optimality, along with experiments suggesting improved energy estimation accuracy and potential benefits for downstream uses such as model composition and Boltzmann-generator/SMC-style pipelines. Conceptually, this is a classification-based  density-ratio estimation approach adapted to distinguish samples from different time levels and thereby recover consistent log-densities/energies.

**Compliance With Llm Reviewing Policy:**

Affirmed.

**Final Justification:**

While I still have some reservations about how strongly the controlled-setting improvements will translate to larger-scale applications, the rebuttal clarified the scope of the claims and helped me better understand the intended use cases. I think the paper has genuine potential and could prove practically valuable. For that reason, I am comfortable giving it a chance and raising my score to 4.

**Key Questions For Authors:**

Question 1: How exactly are the log-density $\log p_t$   obtained for DSM  in the log-density estimation comparisons and when used inside SMC/composition? Would a stronger “DSM + a clever score to energy reconstruction (if exists)”  algorithm change the conclusions?

Question 2:  What is the additional training cost (e.g. running time)  of DiffCLF relative to DSM for the reported experiments, and how sensitive are results to the number of classes N ?

Question 3: A key question for me is whether this approach can be validated at larger scale (e.g., image datasets). You mention this as future work. Do you have a concrete plan or initial thoughts on how to test DiffCLF on image-scale diffusion models (even at a rough level), and what metrics you would use to demonstrate benefit?

**Limitations:**

Yes. The paper acknowledges  at the end that experiments are limited in scale and points to larger-scale image modeling as future work.

**Strengths And Weaknesses:**

Strengths

1. Interesting, simple idea: Adding a simple classification-style (density-ratio formulation) loss to DSM to better recover energies/log-densities is conceptually interesting and easy to integrate into existing training.

2. Theoretical verification: The paper provides analysis and claims consistency/identifiability at optimality, positioning DiffCLF as not “mode-blind” in contrast to score-only approaches.

3. Experimental validation: On synthetic multimodal examples (e.g., mixtures), the paper demonstrates improved agreement with true log-densities/energies while maintaining comparable sampling/score metrics.

Weaknesses

1. Limited realism (This is my main concern): Many of the clearest demonstrations of “energy helps” rely on synthetic or toy setups (mixture benchmarks, toy OR/AND composition). While these are useful evidence, the practical impact on realistic large-scale generative modeling tasks remains unclear. The authors also note that experiments are limited in scale and leave large-scale image modeling to future work.

2. Baseline clarity: Since DSM does not directly output the log-density $\log p_t$, it would help to clarify exactly how energies/log-densities are obtained for DSM when computing “energy fidelity” metrics and when used inside SMC/composition pipelines.

Overall, I find the approach conceptually interesting and plausibly useful for energy-based downstream workflows, but the current empirical evidence is not yet convincing enough  that the benefits extend beyond controlled toy settings if we are looking for something significant for  diffusion model.

---

> ### Author Rebuttal · Authors · 2026-03-30
>
> We thank the reviewer for their thoughtful and balanced assessment. We especially appreciate the recognition that our approach is conceptually simple and easy to integrate, supported by theoretical guarantees, and validated through carefully designed experiments demonstrating improved energy recovery. We address the remaining questions below.
>
> ---
>
> > Baseline clarity: [...] How exactly are the log-density $\log p_t$ obtained for DSM [...]
>
> Thank you for raising this important point. In our setup (and in prior work), DSM-based models recover energies by parameterizing the score as the gradient of a scalar network, i.e., $g^\theta_t(x) = \nabla_x f^\theta_t(x)$. Training via DSM yields $\nabla f^{\theta^\star}_t \approx \nabla \log p_t$, which implies (up to an additive constant) $f^{\theta^\star}_t \approx \log p_t$. This is the standard way energies are obtained from DSM, and is exactly how we compute energy-based metrics and integrate DSM into SMC/composition pipelines.
>
> ---
>
> > Limited realism (main concern): [...] empirical evidence [...] beyond controlled toy settings [...]
>
> We appreciate this important concern. While part of our evaluation uses controlled settings (which are already challenging as competitors fail on them), the paper also includes molecular experiments, which are real-world, high-impact applications where accurate energies are essential.
>
> In these settings, DiffCLF demonstrates clear improvements, supporting its relevance beyond toy problems. More broadly, we note that if competing methods fail in controlled multimodal settings, they are unlikely to succeed in more complex real-world scenarios where such issues are exacerbated.
>
> We agree that larger-scale validation is valuable and view this work as a foundational step in that direction.
>
> ---
>
> > What is the additional training cost [...] and sensitivity to the number of classes ( N )?
>
> As described at the end of Section 3, DiffCLF introduces a cost proportional to pairwise comparisons across time levels, i.e., $\mathcal{O}(N(N-1))$. In practice, this remains moderate, as $N$ is small (typically 2, 4 or 8). Importantly, throughout the paper we ensure fair comparisons by matching the total number of network evaluations across methods.
>
> Empirically, we observe that performance is not highly sensitive to $N$ within this range, and small values already provide strong improvements.
>
> ---
>
> > [...] validation at larger scale (e.g., image datasets) [...] concrete plan [...]
>
> This is an excellent question. A natural next step would be to replicate image-scale composition experiments such as Thornton et al. (2025). However, a key challenge in this domain is the lack of reliable evaluation metrics: assessing whether composed image models behave as intended is often ill-defined. While we conducted preliminary experiments, we chose not to include them due to this ambiguity within the ICML timeline. Looking forward, our concrete plan is to focus on molecular modeling, where:
> * accurate energies are critical,
> * evaluation is well-defined, and
> * established benchmarks (e.g., Skreta et al., 2025b) exist.
> We believe this direction is particularly well-suited for DiffCLF and intend to pursue it in future work, potentially in a more specialized venue.
>
> ---
>
> We thank the reviewer again for highlighting the simplicity, theoretical grounding, and practical promise of our approach. We hope the above clarifications address the concerns regarding baselines, realism, and scalability.
>
> We would greatly appreciate if the reviewer would consider raising their score in light of these clarifications.

---

> > ### Author Rebuttal · Reviewer_WkeT · 2026-04-03
> >
> > The authors clarified several points helpfully, especially regarding the DSM baseline and the role of $N$. However, I still remain somewhat unconvinced on my main concern: while the toy experiments are useful for isolating the intended failure mode, it is not yet clear to me that improved performance in these controlled settings should translate to stronger performance at larger scale, where additional factors may intervene. I am also still uncertain about the intended scope of the claimed benefit, in particular whether the method is expected to improve practical downstream tasks such as image composition, or whether its main value lies primarily in settings where the learned energy function itself is more directly tied to the final objective.

---

> > > ### Author Response · Authors · 2026-04-04
> > >
> > > We thank the reviewer for the follow-up and are glad that our previous clarifications were helpful.
> > >
> > > > Scale of experiments
> > >
> > > We appreciate the concern regarding scalability. To better contextualize our experimental choices, we provide a brief overview of the **evaluation protocols used in closely related works** across the literature:
> > >
> > > * **Boltzmann Generators**: prior works typically evaluate on 2D Gaussian mixtures and toy particle systems [a,b,c], with some recent works including molecular systems such as Alanine Dipeptide [c].
> > > * **Model composition**: most works rely on 2D toy compositions alongside image experiments evaluated primarily via visual inspection or FID, which do not reliably capture issues such as mode miscalibration [e,f].
> > > * **Force field estimation / free energy**: experiments are conducted on molecular or particle systems, similar to [g,h,k].
> > > * **Solely EBM training**: evaluations are commonly performed on Gaussian mixtures and MNIST-scale tasks to check generation [i,j,k].
> > >
> > > From this perspective, our work aims to unify and cover the range of evaluation settings used across the past several years, including high-dimensional Gaussian mixtures (up to 128D), controlled composition tasks, and realistic molecular applications. Importantly, these are standard and accepted benchmarks in the literature, and many of the cited works were published at major venues such as ICML/NeurIPS/ICLR.
> > >
> > > We agree that large-scale image experiments would be valuable, but we note that they remain difficult to evaluate rigorously (especially for composition). We therefore view this as a natural direction for future work, and will state this limitation more explicitly.
> > >
> > > That said, our key point is that DiffCLF improves over baselines in controlled settings where the failure mode is clearly isolated. Since this limitation is intrinsic to the objective, improvements in these settings are expected to translate to at least comparable behavior in more complex regimes.
> > >
> > > > Scope of the benefit
> > >
> > > We thank the reviewer for raising this point and agree that the scope should be clarified. Our primary target is indeed "settings where the learned energy is directly tied to the final objective". This includes model composition [e,f], Boltzmann generators / sampling [a-d], force field learning [g,k] and free energy estimation [h,k]. In these settings, inaccuracies in the learned energy lead to direct and measurable failures:
> > > * *model composition*: miscalibrated energies result in incorrect mode weights (e.g., over/under-represented classes).
> > > * *Molecular simulations / Boltzmann generators*: energy errors lead to unstable or physically incorrect dynamics.
> > > * *Free energy estimation*: inaccuracies produce systematically biased estimates.
> > >
> > > Thus, improving energy estimation is not only an intermediate goal, but directly impacts downstream performance in these applications.
> > >
> > > More broadly, we emphasize that DiffCLF:
> > >
> > > * addresses a **long-standing limitation** of score-based training in EBMs,
> > > * introduces a **simple and computationally efficient solution**,
> > > * is supported by **theoretical guarantees**,
> > > * is thoroughly compared to prior work,
> > > * and is validated across **multiple challenging and representative settings**.
> > >
> > > ---
> > >
> > > Taken together, we believe this combination of novelty, theory, and empirical validation places the contribution well within the scope of ICML. We hope these clarifications address the remaining concerns and would greatly appreciate reconsideration of the score.
> > >
> > > ---
> > >
> > > [a] Phillips et al., Particle Denoising Diffusion Sampler, 2024, ICML.
> > > [b] Zhang et al., Efficient and Unbiased Sampling of Boltzmann Distributions via Consistency Models, 2024.
> > > [c] Zhang et al., Efficient and Unbiased Sampling from Boltzmann Distributions via Variance-Tuned Diffusion Models, 2025, TMLR.
> > > [d] Zhang et al., Accelerated Parallel Tempering via Neural Transports, 2026, ICLR.
> > > [e] Du et al., Reduce, Reuse, Recycle: Compositional Generation with Energy-Based Diffusion Models and MCMC, 2023, ICML.
> > > [f] Thornton et al., Composition and Control with Distilled Energy Diffusion Models and Sequential Monte Carlo, 2025, AISTATS.
> > > [g] Plainer et al., Consistent Sampling and Simulation: Molecular Dynamics with Energy-Based Diffusion Models, 2025, NeurIPS.
> > > [h] Máté et al., Neural Thermodynamic Integration: Free Energies from Energy-Based Diffusion Models, 2024, J. Phys. Chem. Lett.
> > > [i] Choi et al., Density Ratio Estimation via Infinitesimal Classification, 2022, AISTATS.
> > > [j] Guth et al., Learning Normalized Image Densities via Dual Score Matching, 2025, NeurIPS.
> > > [k] He et al., RNE: Plug-and-Play Diffusion Inference-Time Control and Energy-Based Training, 2026, ICLR.

---

### Official Review · Reviewer_wwtW · 2026-03-05

**Soundness:** 3
**Presentation:** 3
**Significance:** 2
**Originality:** 3
**Overall Recommendation:** 4
**Confidence:** 3

**Summary:**

Standard diffusion/score models are widely used but can be mode-blind. The paper proposes Diffusive Classification (DiffCLF): a cross-entropy objective, which reframes log-density estimation as a supervised multi-classification problem.

In terms of theoretical guarantees, DiffCLF alone can have non-unique minimizers, but adding DSM resolves this and supports consistency as well as asymptotic normality for the learned model.  Some experimental results were also conducted.

**Compliance With Llm Reviewing Policy:**

Affirmed.

**Final Justification:**

The rebuttal clarified some of my questions and importantly rigorously demonstrated the $\approx$ used in the proof. Therefore, I remain my score 4.

**Key Questions For Authors:**

1. Can you specify meaning of DiffCLF being unbiased training principle ( around line 417) ?
2. You have used approximate equalities $\approx$ several places in the appendix ( line 907, 909, 980, 985, 1657, 1993 et. al.). Can you be more specific about what it means?
3. Can you provide illustration on how DiffCLF could deal with distributions with disjoint supports, since you mentioned that as a caveat of other methods?
3. Can you give more introduction/math preliminaries about backward SDE for better readability?

**Limitations:**

How to deal with the fact that the learned energies are not guaranteed to be properly normalized densities might be a future work.

**Strengths And Weaknesses:**

Strengths:

1.This method deals with the mode-blindness weakness of the traditional scoring matching method.

2.The new proposed idea is conceptually useful but simple --- it considers a standard multi-class classification

3.The authors provide clear connection between this new work and existing works.

4.Theoretical justification is provided with weak limits obtained.

Weakness:

1.DiffCLF alone is not identifiable with possibly multiple global minima.

2.As the other side of the coin, the learned energies are not guaranteed to be properly normalized densities, which might cause trouble for downstream tasks.

---

> ### Author Rebuttal · Authors · 2026-03-30
>
> We thank the reviewer for their positive and encouraging assessment of our work. We especially appreciate the recognition that (i) DiffCLF addresses an important limitation of score matching, (ii) the proposed idea is conceptually simple yet effective, (iii) the connections to prior work are clearly established, and (iv) the paper provides meaningful theoretical justification. We address the remaining questions and suggestions below.
>
> ---
>
> > As the other side of the coin, the learned energies are not guaranteed to be properly normalized densities, which might cause trouble for downstream tasks.
>
> We would like to clarify that none of the downstream tasks we consider require knowledge of the normalizing constant. In particular, applications such as Boltzmann generators, model composition, and free energy estimation only rely on relative energies, not normalized densities. We will make this point clearer in the revision to avoid confusion.
>
> ---
>
> > Can you specify meaning of DiffCLF being unbiased training principle (around line 417)?
>
> Thank you for pointing out this ambiguity, we agree that the wording is unclear. What we meant is the following: the Monte Carlo approximation of the DiffCLF objective admits a unique correct global minimizer, whereas standard objectives (e.g., DSM) are mode-blind almost surely, introducing additional spurious minimizers and thus effectively biasing training. We will revise the wording to make this precise.
>
> ---
>
> > You have used approximate equalities $\simeq$ several places in the appendix [...]
>
> Thank you for this careful reading. The approximate equalities arise from different standard sources, including:
>
> * SDE discretization (e.g., lines 907–909)
>
>     * We have added details regarding the approximation error. Specifically, this error is quantified by the difference in the expectation of a test function evaluated under the true SDE transition kernel versus the Euler-Maruyama discretized kernel. We demonstrate that this error is bounded by $O(\Delta t^2)$, where $\Delta t$ denotes the discretization step size.
> * First-order Taylor expansions (e.g., lines 980, 985, and others),
>     * We added the error terms for all these Taylor expansions, which are bounded by $O(\Delta t^2)$
> * Monte Carlo approximations of expectations,
> * L1657 is the Central limit theorem argument (for large sample sizes).
>
> We agree that this is not sufficiently explicit in the current version and will clarify each occurrence individually in the revision for improved readability.
>
> ---
>
> > Can you provide illustration on how DiffCLF could deal with distributions with disjoint supports [...]?
>
> This behavior is illustrated in Fig. 2. In that example, distribution $p_1$ (red samples) clearly exhibits disjoint supports. Standard DSM learns from these samples alone. In contrast, DiffCLF leverages samples at multiple noise levels (e.g., green and red), and the classification objective compares these distributions directly. This enables relative calibration across supports, even when individual distributions are disconnected. We will highlight this intuition more explicitly in the revision.
>
> ---
>
> > Can you give more introduction/math preliminaries about backward SDE for better readability?
>
> The full background is currently provided in Appendix A.
>
> ---
>
> > How to deal with the fact that the learned energies are not guaranteed to be properly normalized densities might be a future work.
>
> We respectfully note that this is not a limitation for the tasks considered in this paper, as discussed above. Nevertheless, we agree that clarifying this point earlier would be helpful, and we will emphasize this more clearly in the revision.
>
> ---
>
> We thank the reviewer again for their positive evaluation and for highlighting the simplicity, originality, and theoretical grounding of our approach. We hope the above clarifications address the remaining questions and improve the overall clarity of the paper.
>
> We would greatly appreciate if the reviewer would consider raising their score in light of these clarifications.

---

> > ### Author Rebuttal · Reviewer_wwtW · 2026-03-31
> >
> > Thanks for the detailed reply. I am generally satisfied with the response and have the following questions/remark.
> > The only concern that I have is about the $\approx$ in the proofs. Besides that, I appreciate your replies and I think they have adequately addressed my questions.
> >
> > It seems that the $\approx$ comes from different sources as suggested by your response. Some might indicate a negligible high-order term, some might indicate a multiplicative factor that is universal. Using $\approx$ can be quite misleading especially when there are different type of convergences.
> >
> > One common technique people use is to use $x \lesssim y $ to represent that $x\leq Cy$ for some universal constant $C$ and $x \gtrsim y $ to represent that $x\geq Cy$ for some universal constant $C$. Is it possible to formulate your inequalities in this way? Or any other ways that could make the inequality rigorously justified would be good.
> >
> > Besides that, I also want to mention the $\approx$ appeared in L1043, L1561, L1569, L1964, L1969, L1973, L1984. Please also clarify what they mean rigorously.

---

> > > ### Author Response · Authors · 2026-04-01
> > >
> > > We thank the reviewer for acknowledging our rebuttal. We now address the remaining concern of the reviewer detailedly, regarding approximation errors, which were briefly discussed in the previous response due to space constraints.
> > >
> > > ---
> > >
> > > > L907 and L909
> > >
> > > These are Euler-Maruyama discretized kernel that approximate the transition kernel in SDEs. The errors are quantified by taking expectation of a test function evaluated under the the SDE transition kernel versus the approximated one. In particular, for any test function $h$ with polynomial growth, let $q_{t|s}(\cdot| y_s), q_{s|t}(\cdot | y_t)$ denote the Gaussian approximated kernels respectively, we have
> > >
> > > $$ E_{p_{t|s}(\cdot|y_s)}[h(Y_t)] = E_{q_{t|s}(\cdot|y_s)}[h({Y}_t)] + O(\delta^2) $$
> > >
> > > $$ E_{p_{s|t}(\cdot|y_t)}[h(Y_s)] = E_{q_{s|t}(\cdot|y_t)}[h({Y}_s)] + O(\delta^2) $$
> > >
> > > Hence, these are high order errors w.r.t. $\delta$.
> > >
> > > ---
> > >
> > > > L980 and L985
> > >
> > > Both approximations introduce a second order error, $\mathcal{O}(\delta^2)$, due to the footnote stated in L988-989.
> > >
> > > ---
> > >
> > > > L1043
> > >
> > > This is the MC estimation of the expectation $\mathbb{E}_{p(n)}\mathbb{E}_{p_n}$, which is standard.
> > >
> > > ---
> > >
> > > > L1561
> > >
> > > This is the first-order Taylor expansion, where the high order term $\mathcal{O}(\|\theta-\theta^{\star}\|^2)$ is negligible.
> > >
> > > ---
> > >
> > > > L1657
> > >
> > > This is the Central limit theorem argument, which is standard.
> > >
> > > ---
> > >
> > > > L1964, L1969, L1973, L1984, and L1993
> > >
> > > We agree that these approximations require further clarification. Due to space constraints, we have provided a revised, more rigorous proof (updating L1961–2008) with expanded details here: https://anonymous.4open.science/r/diffclf-rebuttal/revised_proof.pdf
> > >
> > > ---
> > >
> > > We thank the reviewer for their helpful and constructive feedback regarding the mathematical rigors and details. We will include the aforementioned clarifications to the revised manuscript.
> > > We hope these clarifications address the remaining concerns, and we would appreciate the reviewer considering raising their score.

---

### Official Review · Reviewer_zqPZ · 2026-03-10

**Soundness:** 2
**Presentation:** 2
**Significance:** 2
**Originality:** 3
**Overall Recommendation:** 3
**Confidence:** 4

**Summary:**

​​This paper addresses the challenge of training time-conditioned Energy-Based Models (EBMs) that can directly output accurate scalar energy values (defined as the negative unnormalized log-densities) without requiring continuous ODE/SDE integration.

The authors identify that when EBMs are trained using standard Denoising Score Matching (DSM), they suffer from "mode blindness". Because DSM only penalizes the spatial gradients of the energy landscape, the model successfully learns the local shapes of isolated data modes but fails to learn the relative probability masses between disjoint regions.

To solve this, the central contribution of the work is the introduction of a Diffusive Classification Loss (DiffCLF). This method acts as a generalization of Noise Contrastive Estimation (NCE) applied to time-conditioned EBMs over Gaussian-diffused data. During training, the model evaluates the scalar energy of a data point at $N$ randomly sampled time levels and is trained via categorical cross-entropy to classify which time step the noisy sample actually originated from.

Crucially, as established in Propositions 3.1 and 3.2, minimizing DiffCLF alone is not sufficient for optimality; it must be optimized jointly with the DSM loss. While the DSM loss accurately learns the local vector fields (gradients), the DiffCLF loss directly evaluates the scalar energies to learn the correct global constants (relative weights) between disconnected modes, effectively curing the mode blindness of pure score matching.

Computationally, DiffCLF avoids the need for double-backpropagation, requiring $N$ forward network evaluations per training step alongside the standard DSM loss. By doing so, this method effectively offloads the computational cost from inference—where standard diffusion models require iterative ODE/SDE integration to preserve mode weights—to the training phase.

**Compliance With Llm Reviewing Policy:**

Affirmed.

**Final Justification:**

The authors did not address my primary concern, so I decided to maintain my initial assessment.

**Key Questions For Authors:**

1. Clarification on Table 1: What exactly is being reported regarding the different values of $N$ in this table? Are these the "best" results across the different $N$ values for each dimension? If so, by what specific measure was the "best" model selected?
2. Computation of Fisher Distance: How exactly is the Fisher distance computed in the evaluations? Is it evaluated only on the score at $t=0$, or is it integrated over the entire time range?
3. Time-Dependent Bias Vector: In Lines 208-209 (left column), the text refers to a $t$-dependent bias vector. How is this implemented in practice? If there is a single normalizing constant for all time levels, it cannot be $t$-dependent. Could the authors clarify the exact parameterization used here?
4. Training Recipe and Loss Evolution: Appendix G.2.2 describes a training recipe that involves a DSM warmup followed by the combined loss (L_DSM​+L_clf​). What is the justification for this specific schedule? It would be highly beneficial to provide an ablation study on this warmup phase, as well as a visualization of how the two loss terms evolve during training to understand their interaction.

**Limitations:**

yes

**Strengths And Weaknesses:**

**Soundness**

Mismatch Between Core Claim and Evaluation Metrics:
The central theoretical claim of DiffCLF is that it cures "mode blindness" by learning the correct global scalar energies (the correct relative normalization constants or free energy differences between disconnected modes). However, the primary quantitative metric used is Maximum Mean Discrepancy (MMD), which measures sample quality, not energy accuracy. If the samples are generated via diffusion (ODE/SDE integration), the continuous integration process natively routes the probability mass correctly, meaning the baseline L_DSM​ will produce good samples even if its underlying scalar energy values $U_\theta​(x)$ are completely wrong.
To prove that DiffCLF actually learns better energies, the authors must evaluate the energy directly, e.g., by reporting Negative Log-Likelihood (NLL) and comparing with the baselines.

Incomplete and Tautological Evaluation Metrics:

- Evaluating on the Training Loss: In Table 1 (and similar tables), evaluating the model based on L_clf​ is not informative, as this is the exact objective function being optimized.
- Missing Distributional Metrics: Given that the authors have direct access to the score, they should evaluate using Kernelized Stein Discrepancy (KSD) rather than just MMD. KSD is the gold standard for evaluating unnormalized density models. Additionally, reporting the Hellinger distance would provide a much more robust measure of distribution match.
- Inconsistencies in Reporting: It is unclear why Table 5 does not report MMD when it is the primary metric used elsewhere in the paper.

Handicapped Baselines (Architecture Mismatch in CtSM):
The authors compare their method against Conditional Time Score Matching (CtSM). However, they force the CtSM baseline to use an EBM architecture (a scalar output requiring autograd to compute the score). The primary advantage of methods mathematically equivalent to CtSM (like Dual Score Matching) is their architectural design: they use a standard vector-output network and an inner product to define the energy, completely avoiding the costly double-backpropagation. Forcing the baseline into a scalar-autograd architecture strips it of its main advantage.

Unfair Compute Matching (Missing Hyperparameter Tuning):
To match the compute (NFEs) of the more expensive DiffCLF, the authors likely increased the batch size or training steps for the L_DSM​ baseline. However, there is no mention of tuning hyperparameters (specifically applying the linear scaling rule to the learning rate) to account for this. If the learning rate was kept fixed, the baselines were likely under-optimized and artificially handicapped.

Inaccurate Claim regarding Baseline:
Line 273 (right column), the authors seem to mischaracterize Yu et al. (2025). That work does not use a standard DSM objective, and their model architecture is not compatible with it.

**Presentation**

The "Mode Blindness" Narrative:
The paper frames "mode blindness" as a fundamental flaw of score matching. However, the narrative obscures the fact that standard Diffusion Models and Flow Matching models—which parameterize a vector field and sample(evaluate density) by solving (augmented) ODE from(to) a unimodal prior—do not suffer from mode blindness as nicely shown in Figure 9. The paper needs to explicitly clarify early on that they are solving a problem that is entirely specific to pure Energy-Based Models (EBMs) that evaluate scalar densities without temporal integration.

Redundancy in Mathematical Background:
Repeating the equations for "Example 2: Stochastic Interpolants" immediately after "Example 1: Diffusion Models" is unnecessary. Since the underlying mathematical framework is functionally equivalent, repeating the equations disrupts the flow without adding new information.

Notation:
In Lines 202-204 (left column), the convention of using $i$ as a shorthand for $t_i$​ is highly counterintuitive and causes unnecessary confusion when reading the derivations. Please stick to $t_i$​.

Missing Citations and Formatting:

- Line 051 (right column): The claim "In contrast, direct access to the energy often leads to improved performance" requires a citation to substantiate it.
- Multiple references in the bibliography are missing publication venues/proceedings information.

**Significance**

The significance of this work is moderate. While correcting the "mode blindness" of pure EBMs is a theoretically interesting endeavor, the problem setting is somewhat narrow in scope. As noted earlier, mainstream diffusion and flow matching models naturally bypass this issue via ODE/SDE integration from a unimodal prior. Furthermore, the empirical results suggest that the benefits of DiffCLF are largely confined to lower-dimensional spaces. As the dimensionality of the data increases, the performance gap between the proposed method and standard baselines effectively disappears. Consequently, while the paper successfully resolves a specific theoretical quirk of EBMs, its broader practical impact on large-scale, high-dimensional generative modeling appears limited.

**Originality**

To the best of my knowledge, the proposed DiffCLF method is original. Adapting the principles of Noise Contrastive Estimation (NCE) into a time-conditioned classification task across varying diffusion noise levels is a clever approach to calibrating global energy constants. While the theoretical pitfalls of standard score matching in EBMs have been discussed in prior literature, the specific mechanism of using categorical cross-entropy across discrete time steps to explicitly align the relative weights of disconnected modes offers a fresh and creative contribution to the field.

---

> ### Author Rebuttal · Authors · 2026-03-30
>
> We thank the reviewer for their careful reading and thoughtful feedback, and for recognizing the originality and positioning of our work. Below we address the main points.
>
> > Computationally, DiffCLF avoids the need for double-backpropagation [...] effectively offloads the computational cost from inference [...] to the training phase.
>
> We believe this point stems from a misunderstanding and would like to clarify. First, DiffCLF is fully simulation-free, just like DSM. The only extra cost during training comes from evaluating pairs of time indices which we explicitly control (with $N$), making the overhead moderate in practice.
>
> More importantly, for the class of tasks we consider (see Sec. 2.2), it is intrinsically necessary to access both energies and scores at inference time. As a result, there is no meaningful notion of “offloading” computation from inference to training or vice versa in our setting (both quantities must be available regardless of the training objective).
>
> > the primary quantitative metric used is MMD [...] To prove that DiffCLF actually learns better energies, the authors must evaluate the energy directly
>
> We believe this is a misunderstanding. MMD is used only once as a sanity check to verify that improved energy modeling does not harm generation quality, while energy quality is primarily evaluated via ESS, Fisher divergence, and explicit energy comparisons (Fig. 3 and Appendix).
>
> > Evaluating on the Training Loss: [...] evaluating the model based on $\mathcal{L}_{clf}$ is not informative
>
> We respectfully disagree. Reporting $\mathcal{L}_{clf}$ is informative as:
> 1. it is theoretically robust to mode blindness, making it a relevant diagnostic;
> 2. it reveals that baselines (e.g., CtSM) do not optimize it effectively despite theoretical links (Prop. 4.1);
> 3. jointly achieving low $\mathcal{L}_{clf}$ and Fisher divergence empirically confirms the compatibility of the two objectives.
>
> > Handicapped Baselines: [...]
>
> We follow the formulation in the Dual Score Matching paper, and use exactly the architecture described in Sec. 2.3 of that work. Moreover, this choice still requires backpropagation to recover the score from the energy. Furthermore, Yu et al. 2025 propose a method to learn the time score only, but not EBMs. Hence, their proposed vectorization trick is not applicable in our EBM case.
>
> > Unfair Compute Matching: [...]
>
> We respectfully disagree. Compute is matched by ensuring that all methods perform the same number of network evaluations and see the same number of data points. Applying learning-rate scaling rules from large-batch literature is not appropriate here given the small $N$. Such adjustments would be disproportionate and unnecessary. We believe our comparison is fair and controlled.
>
> > The "Mode Blindness" Narrative: [...] The significance of this work is moderate [...] problem setting is somewhat narrow
>
> We believe this stems from a misunderstanding of our objective. As explained in Sec. 2.2, our goal is not standard sample generation, but tasks such as Boltzmann generators, model composition or free energy estimation. These require accurate energies, not just accurate scores, and are therefore sensitive to mode blindness.
>
> Regarding Fig. 9: “Simulation” evaluates the $t=0$ force field via Langevin dynamics (incorrect for DSM, correct for DiffCLF), while “Diffusion” reflects denoising across times and can appear correct despite inaccurate energies.
>
> > Line 051: [...] requires a citation
>
> We respectfully note that this claim is supported by multiple works already cited in the paper, including results in:
> * Boltzmann generators (e.g., Phillips et al., 2024 vs. Zhang et al., 2025a)
> * Model composition (e.g., Thornton et al., 2025 vs. Du et al., 2023)
>
> These consistently show that access to energies improves performance over score-only approaches.
>
> > [...] benefits [...] confined to lower-dimensional spaces
>
> We respectfully disagree. Across all tables and figures, the results consistently show improvements that persist in higher dimensions, without evidence of the gap vanishing.
>
> > Missing Distributional Metrics: [...] KSD [...] Hellinger distance
>
> MMD is not our primary metric; we also report ESS (i.e., upper bound on the $\chi^2$) extensively. KSD is redundant with the Fisher divergence. Direct energy comparisons are shown in Fig. 3 (right) and Appendix.
>
> > Time-Dependent Bias Vector: [...]
>
> We use a discrete time grid of size $K$, the bias is a vector of size $K$. A continuous parameterization gives similar results. We will clarify.
>
> We thank the reviewer for their helpful and constructive feedback and will incorporate all suggested clarifications and corrections in the revision (improving clarity, fixing notation/typos, and completing missing details).
> We hope these clarifications address the concerns and better convey the contribution, and we would appreciate the reviewer considering raising their score.

---

> > ### Author Rebuttal · Reviewer_zqPZ · 2026-04-02
> >
> > Thank you for the rebuttal and the clarifications. I believe some parts of my review may have been interpreted differently than I intended, so I would like to restate my main points more clearly.
> >
> > Score-matching, diffusion, and flow-matching models support exact density evaluation [1], which does not suffer from mode blindness. In my original reading, I mistakenly understood Figure 9 under the “Baseline Diffusion” column as showing density landscapes computed in that way. I now realize these are histograms of samples. However, this does not change my main conclusion. If the diffusion model produces such a landscape in the generative direction, then in principle it can also induce the corresponding landscape in the reverse or normalizing direction. The main challenge is that computing log-density requires solving an augmented ODE, with cost comparable to sample generation, together with Jacobian-related computations at each step. These Jacobian computations are challenging both computationally and in memory. In practice, this is typically mitigated using Hutchinson’s trace estimator [2], which is often the only feasible approach in high dimensions. I agree that reducing this cost is an interesting and important direction, and I acknowledged this positively in my review. However, I still believe the empirical evaluation would be stronger if it included comparisons with a standard score-matching / flow-matching / diffusion baseline across all experiments.
> >
> > Specific comments and questions:
> > 1. Author’s comment on the last paragraph of the review’s “Summary” section
> >
> > I believe there was a misunderstanding here. That paragraph was intended as a positive comment about the paper, not a criticism. At the same time, I would like to clarify two points:
> > - I did not claim that DiffCLF training is not simulation-free.
> > - Since the parameter $N$ scales computational complexity linearly, describing it as a “moderate overhead” does not seem fully justified.
> > 2. MMD, ESS, and other evaluation metrics
> >
> > - MMD is reported in Table 1 in the main text, alongside $\mathcal{L}_{\mathrm{clf}}$​ and Fisher divergence, so I view it as one of the paper’s primary metrics.
> > - ESS is not reported in the main text, and the main text does not point the reader to the appendix for ESS results. For that reason, I did not treat it as a primary metric in my review.
> > - My question about Fisher divergence (Q2) does not appear to have been addressed.
> > - I agree that $\mathcal{L}_{\mathrm{clf}}$ may be informative, but caution is warranted when comparing methods using the training objective of one of those methods.
> > 3. Advantages in higher dimensions
> >
> > Across Tables 1 (FD, MMD), 4 (FD, MMD), 5 (ESS, FD), 6 (MMD, $sW_2$​, TV), 7 (FD), 8 (MMD, $sW_2$, TV), and 10 (FD), the reported results suggest that the baseline DSM model can match or sometimes outperform the proposed method at higher dimensionalities, in some cases by a significant margin. This does not invalidate the contribution, but I believe the manuscript should present this aspect more explicitly and discuss it more carefully.
> >
> > 4. Comment on the "Mode Blindness" Narrative
> >
> > I understand and appreciate the motivation of the method. My concern is narrower: as noted above, just as diffusion models can denoise across time to generate accurate samples, they can also be used in the reverse direction to evaluate accurate densities. For that reason, I do not find the mode-blindness framing fully convincing as stated.
> >
> > 5. Architecture used in the paper
> >
> > I appreciate the clarification that the Dual Score Matching architecture is used. I believe this should be stated explicitly in the manuscript.
> >
> > 6. Learning-rate adjustment
> >
> > I agree that common batch-size scaling heuristics can fail in the small-batch regime. In my view, this reinforces the need for proper learning-rate tuning whenever batch size is changed. Since the rebuttal indicates that compute matching involved changing batch size, I believe this tuning is important for a fair comparison.
> >
> > 7. Claim in L051
> >
> > While I appreciate that the relevant references appear elsewhere in the manuscript, they should be cited where this claim is made. As written, the statement is insufficiently supported at that location.
> >
> > More generally, only one of my four original questions has been addressed in the rebuttal. At this stage, I therefore maintain my initial assessment, while remaining open to further clarification.
> >
> > [1] Lipman, Yaron, et al. "Flow Matching for Generative Modeling." The Eleventh International Conference on Learning Representations.
> >
> > [2] Hutchinson, M. F. (1989). A Stochastic Estimator of the Trace of the Influence Matrix for Laplacian Smoothing Splines. Communications in Statistics - Simulation and Computation, 18(3), 1059–1076. https://doi.org/10.1080/03610918908812806

---

> > > ### Author Response · Authors · 2026-04-03
> > >
> > > We thank the reviewer for taking the time to respond to our rebuttal. We also note that our original rebuttal was significantly longer, but due to the character limits, we had to shorten several explanations.
> > >
> > > > [...] diffusion / flow-matching models support exact density evaluation [...]
> > >
> > > We thank the reviewer for clarifying; we apologize for initially misunderstanding the concern as evaluating energies directly. Regarding the redundancy of learning EBMs when PF-ODE integration can recover densities: while theoretically valid, this is computationally prohibitive. For a time grid of size $K$, computing the likelihood at $p_{t_k}$​​ requires $k$ reverse PF-ODE steps, needing $2k$ network evaluations. In sequential procedures like SMC, this yields an $O(K^2)$ cost. This bound is unrealistically optimistic, as adaptive solvers [1] further inflate costs. Consequently, this strategy is impractical (see [3] which mentions it) and skipped over by most works.
> > >
> > > DiffCLF instead amortizes this cost during training. We do not frame our work as replacing PF-ODE methods, but as advancing EBM estimation by mitigating mode blindness. We will clarify this positioning.
> > >
> > > > [...] I did not claim that DiffCLF training is not simulation-free [...] “moderate overhead”
> > >
> > > Thank you for clarifying. We intended to emphasize that DiffCLF operates within the same paradigm, rather than offloading computation between training and inference (which aligns with our previous point).
> > >
> > > Regarding N: while complexity scales with N, we use small, fixed values (typically N=2 or 4) regardless of dimensionality or task complexity. Consequently, the added cost is consistently low, justifying our description of a “moderate overhead”. We will clarify this.
> > >
> > > > MMD is reported in Table 1 [...] I viewed it as a primary metric [...]
> > >
> > > We agree this was unclear. MMD is included merely as a sanity check to ensure that improving energy modeling doesn't degrade generation quality. Our primary focus remains on energy-related metrics: $\mathcal{L}_{clf}$, time-integrated Fisher divergence, ESS, and direct energy comparisons (e.g., Fig. 3, Appendix). We will clarify this distinction and explicitly reference the Appendix.
> > >
> > > > about Fisher divergence (Q2) [...]
> > >
> > > We apologize for not addressing this point previously due to space constraints. KSD fundamentally compares scores, and is therefore redundant with (and less precise than) the time-integrated Fisher divergence we report. We will clarify this explicitly.
> > >
> > > > [...] caution when comparing methods using the training objective
> > >
> > > We appreciate this remark and are glad this point is acknowledged. To clarify, the reported $\mathcal{L}_{clf}$ is not the time-subsampled training objective, but the full objective evaluated over the complete time grid. Therefore, it is not the exact quantity optimized during training. We will make this explicit.
> > >
> > > > DSM can match or outperform [...] at higher dimensionalities
> > >
> > > We respectfully disagree with this interpretation. Across the tables:
> > > * Table 1 / 4 / 7 show comparable score/generation performance but orders-of-magnitude better energy estimation.
> > > * Table 9 (not mentioned) shows strong improvements even in high dimensions.
> > > We do acknowledge that generation metrics (Tables 6 and 8) degrade slightly in high dimensions, and we will present this more explicitly and discuss it more carefully in the revision.
> > >
> > > Finally, Fig. 6 provides a clear qualitative illustration of the persistent energy modeling advantage in high dimensions.
> > >
> > > > mode-blindness framing [...]
> > >
> > > We believe this point is addressed above.
> > >
> > > > Dual Score Matching architecture [...]
> > >
> > > While this is described in Appendix G, we agree it should be stated explicitly in the main text.
> > >
> > > > learning-rate tuning [...]
> > >
> > > We agree with this point and will include it as an explicit limitation.
> > >
> > > > citation placement [...]
> > >
> > > The relevant citations appear a few lines above, as the statement directly follows from them. We agree this could be clearer and will improve the presentation.
> > >
> > > > [...] only one question addressed [...]
> > >
> > > We respectfully believe that we addressed the full set of concerns within the space constraints. In particular, we clarified:
> > > * the computational infeasibility of PF-ODE likelihood evaluation in our setting,
> > > * the role and interpretation of evaluation metrics,
> > > * the use and meaning of $\mathcal{L}_{clf}$,
> > > * the computational cost and architectural choices,
> > > * and the limitations regarding generation performance and tuning.
> > >
> > > We will further improve clarity in the revision to ensure these points are fully transparent.
> > >
> > > We thank the reviewer again for their detailed and constructive feedback. We hope these clarifications better convey the motivation, scope, and empirical findings of our work, and we would greatly appreciate reconsideration of the score in light of these clarifications.
> > >
> > > [3] Skreta et al. "The Superposition of Diffusion Models Using the Itô Density Estimator." (2024).

---

### Official Review · Reviewer_3vLW · 2026-03-12

**Soundness:** 2
**Presentation:** 2
**Significance:** 3
**Originality:** 3
**Overall Recommendation:** 4
**Confidence:** 3

**Summary:**

The authors present DiffCLF, a method to address mode blindness in score matching trained energy based models. The authors start by pointing out that score matching is relatively insensitive to mixture weights. Based on that observation, the authors propose a self normalized classification loss, which is to be added to the denoising score matching objective. The authors provide a theoretical justification and a thorough analysis of how their work compared to others. They provide support for their method in the form of experiments on gaussian mixtures and other toy distributions, as well as learning the energy landscapes of various molecules.

**Compliance With Llm Reviewing Policy:**

Affirmed.

**Final Justification:**

The authors clarified a lot of the setup in the rebuttal, but I do still find the experiments a weak point of the manuscript. Overall I lean positive.

**Key Questions For Authors:**

See Strenghts / Weaknesses, in particular I'd like to see a better explanation of the key technical details presented around Eqn 10.

**Limitations:**

yes

**Strengths And Weaknesses:**

- This is an interesting problem that I had not considered very much and the authors convinced me of the existence of the problem quite effectively.
- The authors clearly put effort into writing a good paper
- The paper has a considerable amount of background and related work

But:
- I find the exposition sometimes difficult to follow, and generalizations sometimes seem to get in the way of clarity, e.g. with stochastic interpolants, they do not seem to be used after they are introduced. I think the paper might end up clearer if it was focused just on diffusion models.
- The meat of the paper is centered around eqn. 10, but reading that section leaves me with plenty of questions, and not all notation is explained carefully. It is not clear for example how N is chosen or what it even refers to. Is it the number of modes, or the number of timesteps, or something else? The t_i notation seems to indicate the N are a set of timesteps at which the classification is done, but this seems contrary to the text, and the next paragraph which sets i = c, a class label.
- If F^theta_t is implemented as a bias in the neural net, does it still depend on t? Or should this be simply F^theta?
- Overall, paper spends a lot of time comparing to other methods. That time would be better spent focussing more on the method the authors introduce themselves, Section 3 is only about a page long, while Sections 2 and 4 span about 3 pages. I think the paper would be better if more time was spent on the method itself in Section 3.
- Experiments are either on simple, synthetic distributions (MoG's, and and/or problem), or qualitative only (molecule exmples)
Out of these, my biggest concern is that eqn 10 is just not quite explained well enough to understand what the objective actually is, I am not sure I could implement the algorithm the authors described from the explanation in the text

---

> ### Author Rebuttal · Authors · 2026-03-30
>
> We thank the reviewer for their thoughtful and constructive feedback. We especially appreciate their recognition that the paper addresses an interesting and underexplored problem, and that the motivation was conveyed effectively. We are also grateful for the acknowledgment of the effort put into writing and contextualizing the work within prior literature. Below, we address the concerns and clarify the points raised.
>
> ---
>
> > I find the exposition sometimes difficult to follow [...] stochastic interpolants [...] do not seem to be used after they are introduced [...]
>
> Thank you for this helpful suggestion. We would like to clarify that energy-based stochastic interpolants (SIs) are actively used in the paper, not only introduced for generality. In particular:
> * Figure 3 visualizes the performance of energy-based SIs on multimodal distributions,
> * Table 2 uses them for estimating the solvation free energy of Alanine Dipeptide.
>
> That said, we agree that this may not be sufficiently emphasized, and we will revise the presentation.
>
> ---
>
> > The meat of the paper is centered around eqn. 10 [...] not all notation is explained carefully [...] what is $N$ [...]
>
> We appreciate this feedback and agree that this section can be clarified.Here, $N$ refers to the number of classes. Concretely, at each training step:
> * we sample a clean batch $x_0$,
> * sample $N$ time steps $t_1, \dots, t_N$,
> * generate $N$ noisy versions $y_{t_1}, \dots, y_{t_N}$,
> * and assign class $c = i$ when a sample comes from $p_{t_i}$.
> Thus, the classification task is over time-indexed distributions, and the term “class” is used for conceptual clarity. In practice, we use $N = 4$ in most experiments (Figs. 3-5, Tables 1-2), and observe improved performance when increasing $N$, as shown in Fig. 3.
>
> We agree that this was not sufficiently clear and will revise the explanation around Eq. (10) to make the procedure and notation fully explicit.
>
> ---
>
> > If $F^\theta_t$ is implemented as a bias [...] does it still depend on $t$ ? [...]
>
> Yes, $F^\theta_t$ is explicitly time-dependent. It plays the role of a time-dependent log-normalizing constant for the EBM $p_t^\theta$. Given that we discretize time in $K$ steps, $F^\theta_{t_{\cdot}}$ is simply represented as a $K$ dimensional vector. We will clarify this implementation detail in the revision.
>
> ---
>
> > [...] more time comparing to other methods [...] Section 3 is only about a page long [...]
>
> We appreciate this suggestion. The brevity of Section 3 reflects the fact that the proposed method is intentionally simple and clean. However, we agree that the current balance may give the impression that the method is under-explained. We will revise the paper to expand Section 3, providing additional intuition and pseudoalgorithms.
>
> ---
>
> > Experiments are either on simple, synthetic distributions (MoG's, and and/or problem), or qualitative only (molecule examples) [...] I am not sure I could implement the algorithm [...] from the explanation in the text
>
> We thank the reviewer for this important point and agree that clarity of presentation is crucial. We would like to clarify two aspects.
>
> First, while mixture-of-Gaussians may appear simple, they are in fact a challenging benchmark for energy-based models, especially in high dimensions (up to $d = 128$) and when evaluating accurate energy recovery (not just sample quality) which was not achieved before.
>
> Second, regarding implementation: the core idea of DiffCLF is to enforce correct density ratios across time levels, i.e., in the simplest (binary) case, $\log p_{t_i}^\theta(y) - \log p_{t_j}^\theta(y) = \log p_{t_i}(y) - \log p_{t_j}(y)$, which we extend to the multi-class setting. This complements DSM, which enforces matching of score gradients, i.e., $\lim_{h \to 0} \log p_{t_i}^\theta(y) - \log p_{t_i}^\theta(y + h) = \lim_{h \to 0} \log p_{t_i}(y) - \log p_{t_i}(y + h)$.
>
> The implementation itself is directly specified by Eq. (10), which defines the classification loss over time-indexed samples. We acknowledge, however, that the current exposition may not make the training procedure sufficiently explicit. To address this, we will revise the paper to include clear pseudocode for both the binary and multi-class cases, and expand the explanation around Eq. (10), so that the method is straightforward to implement from the main text alone.
>
> We provide pseudo codes in https://anonymous.4open.science/r/diffclf-rebuttal/diffclf-rebuttal.pdf for the reviewer to better understand the proposed method and practical implementation of the diffusive classification loss.
>
> We thank the reviewer again for their constructive feedback and for recognizing the importance of the problem and the effort put into the paper. We will revise the manuscript to improve clarity and strengthen the presentation of Eq. (10).
>
> ---
>
> We hope these clarifications address the concerns, and we would greatly appreciate if the reviewer would consider raising their score in light of these improvements.

---

> > ### Author Rebuttal · Reviewer_3vLW · 2026-04-03
> >
> > I think the authors have mostly addressed my concerns, in particular, I found the anonymous pdf with the alogorithm extremely helpful, I do hope the authors make this part of the appendix if there ends up being a camera ready version, as well as the updates to the writing they promise.
> >
> > On the experimental part I still find I am somewhat unsatisfied. It is certainly true that a high dimensional mixture of gaussians is a very challenging distribution to model, but it still falls in the category where it is possible to tune the experiment to the method at hand, hence, limiting the proof of "broad applicability". Could the authors make arguments about the other datasets they target? And perhaps compare to other benchmarks on their molecule experiments?

---

> > > ### Author Response · Authors · 2026-04-04
> > >
> > > We sincerely thank the reviewer for acknowledging the algorithm pseudo codes are helpful. We would ensure that they, as well as the other updates of writing, will be included in the camera ready version. We hope the following clarifications could help solve the remaining concerns.
> > >
> > > > Regarding the MoG experiments
> > >
> > > We respectively disagree that “it is possible to tune the experiment to the method at hand”. For the Mixture-of-Gaussian experiments,
> > >
> > > * *Figure 1 in [1]* visualizes the failure mode of using DSM-only to learn EBMs in a 2-mixture of Gaussians case, where the mode weights are non-equal. This matches our setting of Stochastic Interpolants where one of the endpoint ($t=1$) is a 2-mixture of Gaussian with imbalance modes. **Our Figure-3-LEFT (bottom)** and the **right most column of Figure-3-RIGHT** justify this observation in $d=2$ and $d=128$ respectively;
> > > * *Figure 4 in [2]* visualizes the learned energy for a 40-mixture of Gaussians in 2-dimensional spaces, which matches **our Figure-3-LEFT (top)**; that said, **our Figure-3-LEFT** effectively reflects the fact delivered by both [1] and [2]
> > > * [2] also conducts experiments on high dimensional MoGs with $d=100$, which considers Diffusion Models from a standard Gaussian to a 40-mixture of Gaussians, where they are visualized in Figure 5. In such a case, DSM only suffers to recover good energy estimations with a low $R^2$ score, which matches our observations in **our Figure-3-RIGHT**. On the other hand, our method obtains a better performance in an even slightly higher dimensional space ($d=128$), and hence further justify our statement.
> > >
> > > > Regarding the molecule experiments
> > >
> > > The molecule experiments we included are considered non-toyish, which are conducted in [1]. Here are the details of them:
> > > * Alanine Dipeptide (ALDP): this is a small molecule system containing 2 amino acids, with 22 atoms in the Cartesian space.
> > > * Chignolin: this is a small protein containing 10 amino acids with 166 atoms
> > >
> > > Therefore, they are realistic problems beyond the synthetic Mixture-of-Gaussians, where our experimental results justify the effectiveness of our method.
> > >
> > > To elaborate more on the effectiveness and efficiency of our method, we will include the following table for the Chignolin task in the camera-ready version:
> > > 1. JS divergence of generated samples
> > > * Langevin: simulation using learned $U^\theta_{t=0}$
> > > * Diffusion: denoising sampling with the learned EBMs, serving as a sanity check that the regularizations (FPE or DiffCLF) don’t hurt.
> > >
> > > | Method                | Langevin | Diffusion |
> > > |----------------------|----------|-----------|
> > > | DSM [1]                 | 0.4351   | 0.0036    |
> > > | DSM (rerun)          | 0.58681  | 0.0079933 |
> > > | DSM+FPE [1]             | 0.0050   | 0.0048    |
> > > | DSM+DiffCLF (ours)   | 0.012812 | 0.0094168 |
> > >
> > > Noting that the both *rerun DSM* and *our method* are slightly worse than the metrics in [1], and that’s because that we don’t have access to the ground truth dataset provided by D.E.Shaw. Instead, we use the pseudo data provided by [1], which has a smaller size and a little bit bias.
> > >
> > > 2. Training time
> > >
> > > More importantly, the training time comparison justifies the efficiency of our proposed method:
> > > | DSM [1] | DSM+FPE [1] | DSM+DiffCLF (ours) |
> > > |-----|--------|--------------------|
> > > | 5 hrs  | 22.5 hrs  | 8.5 hrs               |
> > >
> > > * Noting that, our experiment is conducted with **one** A100 80GB GPU; while DSM+FPE [1] is conducted with **two** A100 80GB GPUs (see their B.6)
> > >
> > > Due to time constraints in the rebuttal period, we only conduct the comparison in the largest scale of experiment, Chignolin, which should be the most representative and where the smaller scale ALDP should behave similar. We will include all these comparisons in the camera-ready version, and we believe that this is not considered a major change as there will be only two additional tables (one for metric and another for training time) in the appendix without changing any existed conclusions.
> > >
> > > ---
> > >
> > > Overall, we believe the experiments conducted in this work is challenging, representative, non-biased to our method, and also realistic. We hope these clarifications address the remaining concerns and would greatly appreciate reconsideration of the score.
> > >
> > > ---
> > >
> > > [1] Plainer, Michael, et al. "Consistent sampling and simulation: Molecular dynamics with energy-based diffusion models." arXiv preprint arXiv:2506.17139 (2025).
> > >
> > > [2] He, Jiajun, et al. "RNE: plug-and-play diffusion inference-time control and energy-based training." arXiv preprint arXiv:2506.05668 (2025).

---

### Decision · Program_Chairs · 2026-04-30

**Decision:**

Accept (regular)

**Comment:**

This paper introduces a conceptually elegant Diffusive Classification (DiffCLF) loss to mitigate the mode coverage issue in diffusion training. Reviewers commended the paper's solid theoretical foundation and its simple, effective formulation. While concerns were raised regarding the reliance on synthetic or small-scale molecular benchmarks, and one reviewer argued for standard PF-ODE density evaluation as a baseline, the authors' rebuttals effectively addressed these points. As PF-ODE integration is computationally infeasible for the targeted downstream applications (like Boltzmann generators or SMC-style pipelines), the proposed method's approach to amortize this cost during training is well justified. Despite one reviewer maintaining a weak reject due to the lack of explicit PF-ODE baseline comparisons, the broader consensus is that the method is technically sound, original, and highly useful to the sub-community working on EBMs. Therefore, I recommend acceptance.